# Toward Stable Brain-Computer Interfaces: Revealing and Addressing Prediction Fluctuations in EEG-Based BCIs

## Abstract

Brain-Computer Interfaces (BCIs) are increasingly used in areas such as neurofeedback and mental healthcare, where reliable real-time feedback is essential. While deep learning (DL) has greatly improved Electroencephalography (EEG)-based BCIs by boosting accuracy in tasks like emotion recognition, attention detection, and workload assessment, current models often suffer from *temporal instability*. Predictions fluctuate erratically across consecutive windows, contradicting the slow-changing nature of cognitive states and producing inconsistent feedback that undermines user engagement. Existing metrics and postprocessing methods fail to capture or resolve this issue effectively. We address this gap through three contributions: (1) a systematic study of prediction fluctuations across datasets, tasks, and representative models; (2) two new stability metrics, Frequency-weighted Spectral Entropy (FSE) and First-Order Difference Standard Deviation (FDS), that directly measure temporal irregularities; and (3) TRin (Temporal Robustness integrated BCI), a fluctuation-aware training framework combining stability-driven losses with curriculum learning. Experiments on three public datasets show that TRin consistently reduces fluctuations while improving accuracy. By introducing stability as a core evaluation dimension, this work provide a new way for more robust and effective real-time BCIs.

## 1 Introduction

Deep learning (DL) has substantially advanced Electroencephalography (EEG)-based Brain-Computer Interfaces (BCIs), achieving strong performance in tasks such as emotion recognition (Wang et al., 2024b), motor imagery (Wang et al., 2024a), attention detection (Hjortkjær et al., 2025), and workload assessment (Ding et al., 2025a). Compared to traditional machine learning methods such as Support Vector Machines (SVM) and Linear Discriminant Analysis (LDA), DL models learn richer temporal-spatial representations and consistently deliver higher accuracy. Representative architectures illustrate this progress: EEGNet (Lawhern et al., 2018) demonstrates the efficiency of compact convolutional networks, TSception (Ding et al., 2023b) leverages multi-scale temporal-spatial convolutions, and Deformer (Ding et al., 2025a) applies transformer-based attention for long-range temporal modeling. These backbones cover diverse inductive biases and underpin much of the current state-of-the-art in EEG decoding. Consequently, evaluation in DL-based BCIs has been dominated by *classification correctness (e.g., accuracy, F1-score)*.

Yet beyond benchmark performance, BCIs are increasingly being deployed in real-world scenarios where reliability is just as critical as accuracy. One of the most prominent domains is neurofeedback, which leverages BCI systems to support mental health interventions. Neurofeedback-based BCIs have been applied in conditions such as Generalized Anxiety Disorder (GAD) (Spielberger, 2013) and Attention-Deficit/Hyperactivity Disorder (ADHD) (Wender et al., 2001), with clinical studies showing effectiveness for emotion regulation (Huang et al., 2021) and attention training (Lim et al., 2019), particularly under personalized protocols (Tan et al., 2024). In these systems, a trained classifier operates in **real time**: EEG signals are segmented into short windows (e.g., 2-4 seconds) and processed with a small sliding step (e.g., every 200 ms) to generate a **continuous stream of predictions**. These predictions are transformed into real-time feedback cues (e.g., visual or auditory) that are presented directly to the user, who learns to regulate their own cognitive or affective states

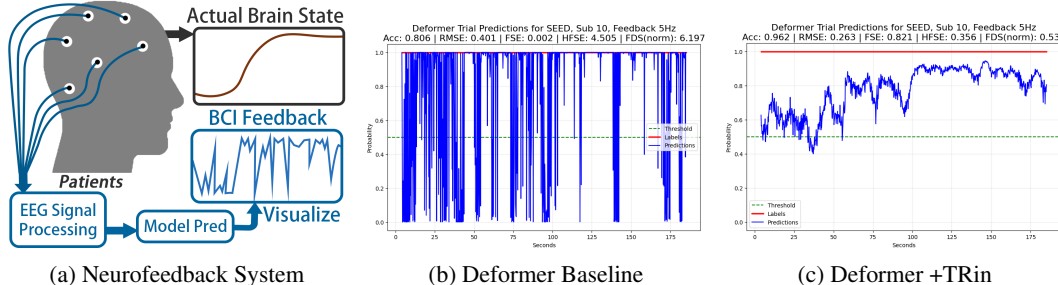

(a) Neurofeedback System     (b) Deformer Baseline     (c) Deformer +TRin

Figure 1: Graph (a) shows how accurate but inconsistant prediction could ruin real-world applications. (b) and (c) show actual examples from model Deformer on SEED dataset. The prediction (blue line) in (b) shows the baseline model with erratic variations in consecutive predictions despite a high accuracy of 0.806, while (c) shows the model with our proposed TRin exhibiting improved accuracy of 0.962 and gradual mental state transitions.

through this closed loop. A critical requirement in this setting is that predictions remain **temporally stable**, as inconsistent counterintuitive feedback degrades the users' engagement with the training process and ultimately reduces neurofeedback efficacy. However, state-of-the-art DL-based BCIs often exhibit **erratic fluctuations between consecutive predictions** (Fig. 1), even while reporting high accuracy. Direct approaches such as post-processing (e.g., moving-average smoothing) can reduce fluctuations, but they introduce response latency and distort accuracy due to outliers. While more principled approaches, such as temporal-aware architectures (e.g., transformers) or regression-based objectives (e.g., mean squared error), may boost classification performance, their benefits to temporal stability are marginal, sometimes even exacerbating fluctuations. From a neuroscience perspective, this is particularly problematic since affective and cognitive states **evolve continuously** and unlikely to exhibit abrupt transitions on the millisecond scale. Emotion is correlated with continuous organism's internal states (Damasio et al., 2000), and lasts for seconds to hours (Verduyn et al., 2015); cognitive attentional mechanisms exhibit slow-changing dynamics (Seeburger et al., 2024) and periodically unfold about 14 seconds (Kasten et al., 2024). Yet DL-based BCIs may output drastically different predictions for consecutive input windows with 95% overlap, suggesting that current models fail to maintain consistent temporal representations.

Despite its significance, **prediction stability has received little systematic attention in BCI research**. While RMSE (Root Mean Square Error) and correlation-based metrics like PCC (Pearson Correlation Coefficient) are metrics potentially reflecting the stability of the prediction, they are more sensitive to the overall classification correctness. Furthermore, because most EEG datasets (Koelstra et al., 2012; Zheng & Lu, 2015; Chen et al., 2023; Shin et al., 2018; Zyma et al., 2019) for cognitive and affective tasks use constant per-trial ground-truth labels, correlation-based metrics degenerate and cannot capture fluctuations. We prove this in Appendix F.2.2.

The above analysis highlights a critical gap: **while DL-based BCIs achieve high accuracy, they lack temporal robustness, and existing evaluation metrics cannot capture this instability**. To address these challenges, we propose a principled framework for both evaluating and mitigating prediction fluctuations in DL-based BCIs. Our main contributions are:

- **Systematic characterization of temporal instability.** To the best of our knowledge, we provide the first systematic study of prediction fluctuations in DL-based BCIs, showing that instability is widespread across datasets, tasks, and architectures, and demonstrating how it undermines neurofeedback effectiveness.

- **New metrics for temporal stability.** We introduce two complementary measures, *Frequency-weighted Spectral Entropy (FSE)* for frequency-domain irregularity and *First-Order Difference Standard Deviation (FDS)* for time-domain smoothness, that directly quantify instability beyond conventional accuracy-based metrics.

- **TRin: a fluctuation-aware training framework.** We present **TRin (Temporal Robustness integrated BCI)**, which combines a fluctuation-aware loss with a curriculum training strategy to jointly optimize accuracy and stability. We further provide theoretical analysis linking our loss design to stability guarantees.

We validate TRin on three public datasets spanning emotion (Zheng & Lu, 2015), attention (Shin et al., 2018), and workload (Zyma et al., 2019), using four representative models, SVM (Zheng & Lu, 2015), EEGNet (Lawhern et al., 2018), TSception (Ding et al., 2023b), Deformer (Ding et al., 2025a). TRin consistently reduces prediction fluctuations while improving classification accuracy. Code is available at: https://anonymous.4open.science/r/TRin.

## 2 METHOD

### 2.1 FLUCTUATION EVALUATION METRICS

We propose two complementary metrics to quantify prediction fluctuations in the continuous per-trial predictions $\hat{Y} = \{\hat{y}_1, \ldots, \hat{y}_T\}$ with ground truth $Y = \{y_1, \ldots, y_T\}$, where there are a total of $T$ samples in an exact trial duration of $S$ seconds. In the following definitions, we donate $y_t$ as the ground truth, $\hat{y}_t$ as the prediction, $e_t = y_t - \hat{y}_t$ as the error at time $t$, and $E = \{e_1, \ldots, e_T\}$ as the error signal. Although for a constant label $Y$, $E$ and $\hat{Y}$ show no fluctuation difference, we aim to generalize our metrics for cases with label changes. In such scenarios, $E$ will remain stable if $\hat{Y}$ changes correspondingly with $Y$. The overview of our proposed evaluation metrics is shown in Figure 2.

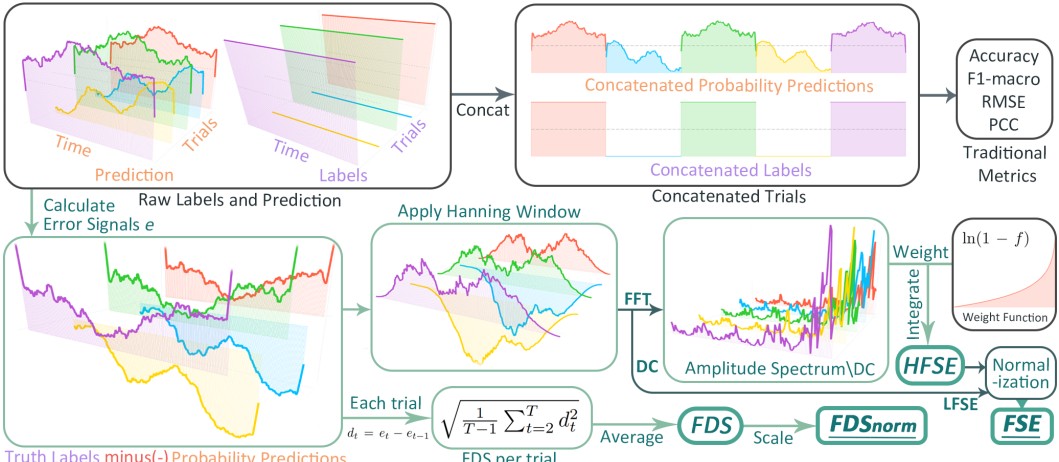

Figure 2: The overview of our proposed evaluation metrics (green) comparing to traditional metrics (grey). FFT donates to Fast Fourier Transform and DC donates to Direct Current Component. FDS and FDS$_\text{norm}$ are defined in Section 2.1.1, while LFSE, HFSE, FSE are defined in Section 2.1.2.

### 2.1.1 FIRST-ORDER DIFFERENCE STANDARD DEVIATION (FDS)

**Definition 1.** *(First-Order Difference Standard deviation).*

$$FDS = \sqrt{\frac{1}{T-1} \sum_{t=2}^{T} d_t^2}, \quad where \ d_t = e_t - e_{t-1} \tag{1}$$

The First-Order Difference Standard deviation (FDS) metric directly quantifies temporal variations by measuring the variance of the difference between two consecutive predictions. Therefore, it maybe affected by overlap or refresh rate of signals, so we introduce normalized FDS defined as FDS$_\text{norm} = FDS \cdot T/S$ to measure instability in absolute time.

### 2.1.2 FREQUENCY-WEIGHTED SPECTRAL ENTROPY (FSE)

Based on the fact that higher frequency components indicate greater fluctuation, the FSE metric offers a unified assessment of both prediction accuracy and temporal stability by analyzing the spectral characteristics in the frequency domain. Similar to FDS, the input signal is the error signal $e_t$.

Before the formal definition of FSE, we introduce the spectral analysis procedure first. The spectral analysis procedure is as follows:

*1. Apply Window and Discrete Fourier Transform (DFT):*

$$X(f) = \mathcal{F}\{E_t \cdot w(t)\} = \sum_{t=1}^{T} e_t \cdot w(t) \cdot e^{-i2\pi ft} \quad \text{where } w(t) = \frac{1}{2}\left(1 - \cos\left(\frac{2\pi t}{T-1}\right)\right) \quad (2)$$

The Hanning window function is applied to reduce spectral leakage. Since for real-valued signals, the DFT output $X(f)$ exhibits conjugate symmetry about the Nyquist frequency $f = \frac{1}{2}$, we could discard $X(f)$ if $f \in (\frac{1}{2}, 1)$ to obtain amplitude spectrum.

*2. Compute Normalized Amplitude Spectrum:*

$$A(f) = \frac{|X(f/2)|}{T \cdot U}, \quad \text{where } U = \frac{1}{T}\sum_{t=1}^{T} w(t), \quad f \in (0, 1] \quad (3)$$

We exclude direct current (DC) component (i.e. $f = 0$) since only alternating current components (i.e. $f \in (0, 1]$) contribute to fluctuation. In addition, note that the DC component is the average of the error signal $e_t$ reflecting classification performance. Therefore, we separate DC component from the amplitude spectrum for further processing and finish the spectral analysis.

*3. Separate Direct Current Component:*

$$\text{DC} = \left|\frac{1}{T}\sum_{t=1}^{T}(y_t - \hat{y}_t)\right| = \left|\frac{1}{T}\sum_{t=1}^{T} e_t\right| \quad (4)$$

The second step yields $A(f)$ which is the normalized amplitude spectrum over the frequency range $f \in (0, 1]$, representing all possible frequencies contributing to fluctuation. Therfore, we wish to apply a frequency weighted integration as the Higher-Frequency weighted Spectral Entropy (HFSE) to quantify the overall fluctuation level. The weight function is chosen with three properties: minimal weight for low frequencies, strong weight for high frequencies, and normalized expectation. Therefore, we propose to use the weight function $-\ln(1-f)$ for its desirable properties:

$$\lim_{f \to 0^+} -\ln(1-f) = 0, \quad \lim_{f \to 1^-} -\ln(1-f) = +\infty, \quad \int_0^1 -\ln(1-f)\,df = 1 \quad (5)$$

**Definition 2.** *Higher-Frequency weighted Spectral Entropy (HFSE).*

$$HFSE = -\int_0^1 \ln(1-f) \cdot A(f)\,df \quad (6)$$

**Remark 1.** *In practice, we scope the frequency range to $(0, 1-\epsilon]$ to avoid numerical instability. We set $\epsilon = 10^{-5}$ in our implementation. Here, we can also understand why a tapered window is necessary during the DFT, since tiny noise caused by spectral leakage in higher frequency components can be amplified by the weighted integration.*

Before combine both classification indicator (DC) and temporal instability (HFSE) together, we wish to avoid misleading performance from incorrect but stable predictions. Therefore, we reversely scale DC from $[0, 1]$ to $[-1, 1]$ to further penalize unacceptable errors. The scaling function we employ is smooth and decreasing derived from the logit function. This function maintains the same scale between DC and HFSE, approximating $\frac{1}{U}$ when near 0 and $-\frac{1}{U}$ when near 1.

**Definition 3.** *Lower-Frequency Weighted Spectral Entropy (LFSE).*

$$LFSE = \ln\left(\frac{3 - 2DC}{1 + 2DC}\right)/U, \quad \text{where } U = \frac{1}{T}\sum_{t=1}^{T} w(t) \quad (7)$$

Finally, we can apply the softmax function to combine LFSE and HFSE to get the Frequency-weighted Spectral Entropy (FSE). The softmax function is chosen to normalize the two components and ensure that FSE is dominated by the worse-performing component for a conservative evaluation.

**Definition 4.** *Frequency-weighted Spectral Entropy (FSE).*

$$FSE = \frac{\exp(LFSE)}{\exp(LFSE) + \exp(HFSE)}, \quad where \ \exp(x) = e^{(\alpha x)}, \quad \alpha \ is \ a \ constant \tag{8}$$

**Remark 2.** *In our implementation, we set softmax temperature $\alpha = 2.0$ to consider more influence of LFSE over HFSE, since models usually have less difference in classification performance than temporal stability (Section 3).*

## 2.2 MODEL TRAINING

During training, we apply temporal sequences preserving data processing (Section 2.2.1), the Temporal Regularization loss function (Section 2.2.2), and dynamic training strategy (Section 2.2.3) discussed in the following subsections. Comprehensive evaluation will be discussed in Section 3. Hyperparameters are tuned on FSE performance on validation set, since it is the only metric considering classification performance and temporal stability all-in-one (Appendix G.1).

### 2.2.1 DATA PROCESSING

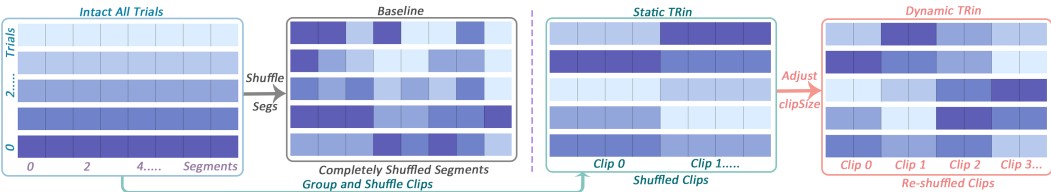

Figure 3: Data shuffling of the Baseline, static TRin, and dynamic TRin system, represensted by gray, green, and pink colors respectively. The same shade of blue indicates that they originate from the same trial.

Fixed training data sequences may cause models to overfit specific sequence patterns rather than capturing the overall distribution. To address this, traditional DL models employ complete random shuffling of training data segments to promote general distribution fitting under cross entropy loss. However, such shuffling entirely destroys the temporal coherence within the data. To preserve essential temporal dependencies while retaining the benefits of randomness, we propose a shuffle clip strategy. Specifically, data segments are grouped into clips, and shuffling is performed at the clip level rather than segment level. For practical implementation, the clip size is chosen as a factor of the batch size. Since shuffle clip is inherently less random than complete shuffling, one might expect potential degradation in classification performance. To mitigate this, we further introduce a curriculum learning approach, named TRin dynamic training strategy that adaptively adjusts clip size over the course of training (Section 2.2.3). Evaluation (Section 3) shows that the impact is negligible for long trials; time-consistant predictions even improves classification performances. Figure 3 illustrates the data processing workflows of the Baseline and our proposed TRin systems.

### 2.2.2 TEMPORAL REGULARIZATION LOSS FUNCTION

Our Temporal Regularization loss function is designed to directly address temporal instability by augmenting the standard cross-entropy loss $\mathcal{L}_{CE}$ with a temporal regularization term $\mathcal{L}_R$. The combined loss function is defined as:

$$\mathcal{L} = \mathcal{L}_{CE} + \alpha \mathcal{L}_R \tag{9}$$

where:

$$\mathcal{L}_{CE} = -\frac{1}{T} \sum_{t=1}^{T} \sum_{c=1}^{C} y_{t,c} \log(\hat{y}_{t,c}) \tag{10}$$

$$\mathcal{L}_R = \frac{1}{(T-1)(C-1)} \sum_{t=2}^{T} \sum_{c=2}^{C} (\hat{y}_{t,c} - \hat{y}_{t-1,c})^2 \tag{11}$$

This loss term takes as input the model's inferred probability sequence $\hat{y}$ with shape $T \times C$, where $T$ denotes the number of time steps and $C$ the number of classes. For each $\hat{y}_t$, regarded as a probability distribution, knowing $\hat{y}_{t,c}$ for $c = 2, \ldots, C$ determines $\hat{y}_{t,1}$. Therefore, the regularization term $\mathcal{L}_R$ only involves components from $c = 2$ to $C$. This term is applied to sequences with preserved temporal order and is thus computed within clips. The batch loss is obtained by averaging losses across all clips. The parameter $\alpha$ controls the trade-off between temporal stability and accuracy and can vary dynamically during training (Section 2.2.3). A larger $\alpha$ enforces stronger temporal stability, whereas a smaller one prioritizes accuracy.

Theoretical analysis of our loss function provides guarantees for temporal stability. According to Theorem 1, the Temporal Regularization loss is convex, so Corollary 1 demonstrates that the loss ensures training to stable and accurate outputs when gradient descent converges. Theorem 2 proves that $\mathcal{L}_R$ mathematically embodies a Gaussian random walk prior. The regularizer imposes $\sigma$-scale local smoothness constraints between adjacent timesteps, statistically bounding $|\hat{y}_{t,c} - \hat{y}_{t-1,c}|$ by a factor of $\sigma$. This aligns with our empirical observations in Figure 1 and 4, and is consistent with neuroscience principles, which suggest that mental states change over time but at a slow rate.

### 2.2.3 CURRICULUM LEARNING TRAINING STRATEGY

We first describe the **static TRin** training strategy. As outlined in Section 2.2.1, the training data are grouped into clips and shuffled at the clip level. In the static version, the clip size is fixed by the hyperparameter clipRate throughout training, where clipSize = clipRate × batchSize. Similarly, the trade-off coefficient $\alpha$ in the loss term (Equation 9) is kept fixed by the hyperparameter $\alpha_s$.

While temporal regularization encourages stable predictions over time, it may also hinder the model from escaping regularization barrier. To address this, we propose the **dynamic TRin** training strategy (algorithm in Appendix D.1 and ablation in Appendix H.1). The dynamic TRin training strategy is a curriculum learning approach derived from its static counterpart. It begins with a weaker regularization to allow the model to learn freely in the early stages, then gradually strengthens regularization to enhance temporal stability as training progresses. To maintain learning flexibility near the end, the method progressively increases randomness by decreasing the clip size in the training data, which helps improve performance on cross-entropy optimization. Other training practices such as batch normalization and dropout are applied as usual.

### 2.3 REAL-TIME INFERENCE

During real-time inference, models under the TRin framework operate identically to baselines, requiring no extra computational cost. Raw EEG signals are preprocessed (e.g., segmented) and fed into the model one segment at a time. Each segment yields a single prediction and then presented to users. Importantly, the TRin framework introduces no additional regularization or dependencies during inference. Unlike post hoc moving average methods, TRin models do not rely on previous signal segments nor past predictions. Latency simulation analysis is provided in Appendix I.

### 2.4 COMPARISON WITH EXISTING METHODS

To the best of our knowledge, TRin is the first application of temporal regularization in BCI models. Empirically, EEG regression tasks are less affected by temporal fluctuations, and typically employ Mean Squared Error (MSE) loss (Ding et al., 2025b), which inherently provides a temporal constraint (Appendix D.2). Thus, we include MSE-based models as baselines.

In other domains with sequential data, regularization methods often minimize differences of predictions (Dileo et al., 2023), graph embeddings (Xu et al., 2021), mutual conditional probabilities (Varghese et al., 2021), and employ recurrent architectures (García-Durán et al., 2018). However, direct adaptation of learning objectives can cause overfitting to specific temporal patterns (Appendix H.2), while recurrent methods depend on previous predictions or segments, introducing feedback latency, which is an issue TRin avoids.

Curriculum learning approach by Dileo et al. (2023) gradually expand the training time coverage. However, in BCI tasks, the time range is limited by the length of the trial. Our dynamic training strategy does not suffer from this limitation and further balances the regularization strength and learning flexibility.

## 3 EXPERIMENTS

### 3.1 EXPERIMENTAL SETUP

#### 3.1.1 DATASET PREPROCESSING AND CROSS-VALIDATION

We evaluate TRin on three widely used BCI datasets **Mental Workload** (Zyma et al., 2019), **SEED** (Zheng & Lu, 2015), and **Attention** (Shin et al., 2018) representing three distinct categories: cognitive load, emotion, and attention, respectively. All datasets undergo the same preprocessing pipeline, including bandpass filtering, artifact removal, downsampling, and segmentation. Detailed dataset descriptions are provided in Appendix B. Following preprocessing, we adopt a standard cross-validation procedure to evaluate model performance and select the optimal model based on validation accuracy. To prevent cross-subject data leakage, we employ **leave-one-subject-out** (LOSO) cross-validation for all datasets. In each LOSO iteration, the data from one subject is held out as the test set. To simulate real-time feedback conditions (Teo et al., 2021; Jochumsen et al., 2019), we further segment the test trials using a 200 ms sliding window (i.e. 95% overlap). For all deep learning models, training and validation data are split with an 80/20 ratio.

#### 3.1.2 MODEL IMPLEMENTATION

We evaluate TRin on four representative deep learning BCI models: **EEGNet** (Lawhern et al., 2018), **TSception** (Ding et al., 2023b), **Deformer** (Ding et al., 2025a), **CBraMod** (Wang et al., 2025), representing compact convolutional networks, temporal-aware convolutional networks, transformer-based architectures, and foundation models, respectively. Traditional **SVM** (Zheng & Lu, 2015) is also included as a reference for acceptable fluctuations, given its widely recognized usability in real-time applications. Details of these models are provided in Appendix C.

For each deep learning (DL) model, we first build a Baseline version following its original publication. This baseline uses a completely random shuffle at the segment level and optimizes the standard cross-entropy loss, without any temporal regularization. As label smoothing is an effective method of restrict overfitting, we also implement a LS version, which is identical to the Baseline version but applies label smoothing ($\epsilon = 0.1$) to the ground truth labels. We then construct the full TRin version by integrating the Temporal Regularization loss (Section 2.2.2) and the dynamic training strategy (Section 2.2.3). To further assess the specific benefit of TRin's regularization, we also implement an MSE loss version, (Details in Appendix D.2). Aside from replacing this loss term, the MSE loss version remains identical to the full TRin one ensuring a fair comparison. Foundation models MSE loss version were not implemented since the MSE loss becomes instable for too large models.

#### 3.1.3 EVALUATION METRICS

We evaluate model performance using both traditional classification metrics and our proposed fluctuation metrics. Traditional metrics include **Accuracy (ACC)**, **F1-score (macro)**, **Rooted Mean Squared Error (RMSE)**, and **Pearson Correlation Coefficient (PCC)**. Definitions are listed in Appendix F.2. For PCC, we concatenate all trials of each subject to avoid degenerate zero values caused by constant trial labels across datasets, as noted in Theorem 3. Although Theorem 4 shows that PCC is not an appropriate indicator of temporal stability, we still report PCC results to illustrate the behavior of conventional metrics in our evaluation. Our fluctuation metrics consist of **Normalized First-Order Difference Standard Deviation (FDS$_{norm}$)** and **Higher-Frequency Spectral Entropy (HFSE)** for temporal stability, and **Frequency-weighted Spectral Entropy (FSE)** for joint assessment of classification accuracy and stability (Section 2.1).

### 3.2 QUANTITATIVE PERFORMANCE ANALYSIS

The experiment results for Mental Workload and SEED datasets are shown in Table 1 and Table 2 respectively. Further analysis and results for Attention dataset are in Appendix E and Table 4.

Results in Table 1 and Table 2 highlight the fluctuation problem inherent in current DL-based BCI systems. Compared against SVMs as a benchmark for acceptable fluctuation, baseline DL models demonstrate significantly poorer temporal stability, despite achieving higher classification accuracy. Specifically, baseline DL models exhibit an average relative increase of more than 265% in HFSE

Table 1: Performance comparison on Mental Workload dataset. Results show mean ± standard deviation over cross-validation folds. The best and performance is highlighted in bold while the second best performance is highlighted in underline.

| Model | ACC($\uparrow$) | F1-macro($\uparrow$) | RMSE($\downarrow$) | PCC($\uparrow$) | FSE($\uparrow$) | HFSE($\downarrow$) | FDS$_{norm}$($\downarrow$) |
|---|---|---|---|---|---|---|---|
| SVM | $0.6795_{\pm 0.1577}$ | $0.6250_{\pm 0.2132}$ | $0.4906_{\pm 0.1671}$ | $0.4544_{\pm 0.3521}$ | $0.6264_{\pm 0.1756}$ | $0.2795_{\pm 0.1564}$ | $0.2669_{\pm 0.1338}$ |
| EEGNet | $0.7111_{\pm 0.1612}$ | $0.6788_{\pm 0.1984}$ | $0.4774_{\pm 0.1679}$ | $0.4862_{\pm 0.3270}$ | $0.3964_{\pm 0.2278}$ | $1.4317_{\pm 0.6873}$ | $1.0126_{\pm 0.4664}$ |
| + *MSEloss* | $0.6869_{\pm 0.1255}$ | $0.6523_{\pm 0.1613}$ | $0.5059_{\pm 0.1219}$ | $0.4673_{\pm 0.2822}$ | $0.3782_{\pm 0.1578}$ | $1.4473_{\pm 0.5076}$ | $1.0856_{\pm 0.3412}$ |
| + *LS* | $\underline{0.7165}_{\pm 0.1524}$ | $\underline{0.6809}_{\pm 0.1887}$ | $\underline{0.4440}_{\pm 0.1521}$ | $\underline{0.5260}_{\pm 0.3317}$ | $\underline{0.4231}_{\pm 0.2142}$ | $\underline{1.0522}_{\pm 0.4436}$ | $\underline{0.7805}_{\pm 0.3084}$ |
| + **TRin** | $\mathbf{0.7348}_{\pm 0.1744}$ | $\mathbf{0.6871}_{\pm 0.2336}$ | $\mathbf{0.4248}_{\pm 0.1713}$ | $\mathbf{0.5865}_{\pm 0.3715}$ | $\mathbf{0.6111}_{\pm 0.1573}$ | $\mathbf{0.4643}_{\pm 0.1835}$ | $\mathbf{0.3536}_{\pm 0.1377}$ |
| TSception | $0.6798_{\pm 0.1640}$ | $0.6306_{\pm 0.2102}$ | $0.5121_{\pm 0.1762}$ | $0.4400_{\pm 0.3270}$ | $0.6071_{\pm 0.1804}$ | $0.3679_{\pm 0.2098}$ | $0.3311_{\pm 0.1708}$ |
| + *MSEloss* | $0.6695_{\pm 0.1554}$ | $0.6248_{\pm 0.1973}$ | $0.5111_{\pm 0.1606}$ | $0.4283_{\pm 0.3272}$ | $0.5697_{\pm 0.1842}$ | $0.4509_{\pm 0.2234}$ | $0.3878_{\pm 0.1669}$ |
| + *LS* | $\underline{0.6828}_{\pm 0.1626}$ | $\underline{0.6381}_{\pm 0.2046}$ | $\underline{0.4713}_{\pm 0.1624}$ | $\underline{0.4982}_{\pm 0.3299}$ | $\underline{0.6299}_{\pm 0.1760}$ | $\underline{0.3582}_{\pm 0.1063}$ | $\mathbf{0.2793}_{\pm 0.0920}$ |
| + **TRin** | $\mathbf{0.7390}_{\pm 0.1724}$ | $\mathbf{0.7106}_{\pm 0.2097}$ | $\mathbf{0.4295}_{\pm 0.1729}$ | $\mathbf{0.5598}_{\pm 0.3725}$ | $\mathbf{0.6652}_{\pm 0.1943}$ | $\mathbf{0.3532}_{\pm 0.1484}$ | $\underline{0.2927}_{\pm 0.1130}$ |
| Deformer | $0.7247_{\pm 0.1602}$ | $0.6923_{\pm 0.1993}$ | $0.4680_{\pm 0.1718}$ | $0.5101_{\pm 0.3194}$ | $0.4301_{\pm 0.2216}$ | $1.4305_{\pm 0.5940}$ | $1.0448_{\pm 0.4584}$ |
| + *MSEloss* | $0.6745_{\pm 0.1486}$ | $0.6320_{\pm 0.1923}$ | $0.5115_{\pm 0.1535}$ | $0.4246_{\pm 0.3124}$ | $0.3748_{\pm 0.2044}$ | $1.4320_{\pm 0.7388}$ | $1.0474_{\pm 0.5243}$ |
| + *LS* | $\underline{0.7377}_{\pm 0.1625}$ | $\underline{0.7103}_{\pm 0.1974}$ | $\underline{0.4440}_{\pm 0.1617}$ | $\underline{0.5386}_{\pm 0.3349}$ | $\underline{0.4350}_{\pm 0.2009}$ | $\underline{1.2895}_{\pm 0.4759}$ | $\underline{0.9259}_{\pm 0.3526}$ |
| + **TRin** | $\mathbf{0.7645}_{\pm 0.1539}$ | $\mathbf{0.7326}_{\pm 0.2031}$ | $\mathbf{0.3983}_{\pm 0.1508}$ | $\mathbf{0.6438}_{\pm 0.3090}$ | $\mathbf{0.6529}_{\pm 0.1568}$ | $\mathbf{0.4064}_{\pm 0.1415}$ | $\mathbf{0.3075}_{\pm 0.1065}$ |
| CBraMod | $0.6810_{\pm 0.1708}$ | $0.6327_{\pm 0.2204}$ | $0.5242_{\pm 0.1832}$ | $0.3945_{\pm 0.3548}$ | $0.3393_{\pm 0.2578}$ | $1.9508_{\pm 1.0203}$ | $1.3621_{\pm 0.7000}$ |
| + *LS* | $\underline{0.6859}_{\pm 0.1592}$ | $\underline{0.6442}_{\pm 0.2041}$ | $\underline{0.4707}_{\pm 0.1446}$ | $\underline{0.4651}_{\pm 0.3669}$ | $\underline{0.3561}_{\pm 0.1800}$ | $\underline{1.4142}_{\pm 0.4622}$ | $\underline{0.9997}_{\pm 0.3131}$ |
| + **TRin** | $\mathbf{0.7106}_{\pm 0.1635}$ | $\mathbf{0.6606}_{\pm 0.2181}$ | $\mathbf{0.4425}_{\pm 0.1317}$ | $\mathbf{0.6030}_{\pm 0.3448}$ | $\mathbf{0.5202}_{\pm 0.1451}$ | $\mathbf{0.5628}_{\pm 0.2294}$ | $\mathbf{0.4112}_{\pm 0.1735}$ |

Table 2: Performance comparison on SEED dataset. Results show mean ± standard deviation over cross-validation folds. The best performance is highlighted in bold while the second best performance is highlighted in underline.

| Model | ACC($\uparrow$) | F1-macro($\uparrow$) | RMSE($\downarrow$) | PCC($\uparrow$) | FSE($\uparrow$) | HFSE($\downarrow$) | FDS$_{norm}$($\downarrow$) |
|---|---|---|---|---|---|---|---|
| SVM | $0.7077_{\pm 0.1465}$ | $0.6587_{\pm 0.1951}$ | $0.4654_{\pm 0.1484}$ | $0.5701_{\pm 0.2354}$ | $0.6224_{\pm 0.1402}$ | $0.3458_{\pm 0.1760}$ | $0.1777_{\pm 0.0877}$ |
| EEGNet | $0.7090_{\pm 0.1565}$ | $\underline{0.6829}_{\pm 0.1801}$ | $0.4849_{\pm 0.1648}$ | $0.5099_{\pm 0.3119}$ | $0.6216_{\pm 0.2292}$ | $0.4377_{\pm 0.5531}$ | $0.2228_{\pm 0.2448}$ |
| + *MSEloss* | $0.7132_{\pm 0.1679}$ | $0.6659_{\pm 0.2195}$ | $0.4627_{\pm 0.1656}$ | $0.5271_{\pm 0.3346}$ | $\underline{0.6636}_{\pm 0.1607}$ | $\underline{0.1803}_{\pm 0.0708}$ | $\underline{0.0987}_{\pm 0.0345}$ |
| + *LS* | $\underline{0.7185}_{\pm 0.1742}$ | $0.6748_{\pm 0.2171}$ | $\underline{0.4463}_{\pm 0.1846}$ | $\underline{0.5709}_{\pm 0.3181}$ | $0.6537_{\pm 0.2047}$ | $0.2786_{\pm 0.2566}$ | $0.1476_{\pm 0.1104}$ |
| + **TRin** | $\mathbf{0.7374}_{\pm 0.1708}$ | $\mathbf{0.7048}_{\pm 0.2099}$ | $\mathbf{0.4285}_{\pm 0.1454}$ | $\mathbf{0.6197}_{\pm 0.3259}$ | $\mathbf{0.6757}_{\pm 0.1610}$ | $\mathbf{0.1121}_{\pm 0.1416}$ | $\mathbf{0.0588}_{\pm 0.0594}$ |
| TSception | $0.6774_{\pm 0.1567}$ | $0.6209_{\pm 0.2051}$ | $0.5280_{\pm 0.1711}$ | $0.4324_{\pm 0.2948}$ | $0.6152_{\pm 0.1686}$ | $0.2786_{\pm 0.1291}$ | $0.1713_{\pm 0.0845}$ |
| + *MSEloss* | $\underline{0.6851}_{\pm 0.1454}$ | $\underline{0.6337}_{\pm 0.1913}$ | $0.5228_{\pm 0.1557}$ | $\underline{0.4580}_{\pm 0.2757}$ | $0.6231_{\pm 0.1604}$ | $0.2586_{\pm 0.1135}$ | $0.1675_{\pm 0.0733}$ |
| + *LS* | $0.6816_{\pm 0.1325}$ | $0.6326_{\pm 0.1752}$ | $\underline{0.5150}_{\pm 0.1433}$ | $0.4502_{\pm 0.2315}$ | $\underline{0.6332}_{\pm 0.1319}$ | $\mathbf{0.1811}_{\pm 0.0828}$ | $\mathbf{0.1013}_{\pm 0.0570}$ |
| + **TRin** | $\mathbf{0.6919}_{\pm 0.1534}$ | $\mathbf{0.6430}_{\pm 0.2007}$ | $\mathbf{0.5102}_{\pm 0.1598}$ | $\mathbf{0.4692}_{\pm 0.2963}$ | $\mathbf{0.6432}_{\pm 0.1554}$ | $\underline{0.2082}_{\pm 0.0822}$ | $\underline{0.1265}_{\pm 0.0557}$ |
| Deformer | $0.7242_{\pm 0.1401}$ | $0.6888_{\pm 0.1850}$ | $0.4813_{\pm 0.1535}$ | $0.5113_{\pm 0.2824}$ | $0.4538_{\pm 0.1908}$ | $1.7730_{\pm 1.1919}$ | $0.7936_{\pm 0.5157}$ |
| + *MSEloss* | $\underline{0.7416}_{\pm 0.1536}$ | $\underline{0.7069}_{\pm 0.2013}$ | $\underline{0.4458}_{\pm 0.1749}$ | $\underline{0.5493}_{\pm 0.3117}$ | $\underline{0.5450}_{\pm 0.2469}$ | $\underline{1.1735}_{\pm 1.0654}$ | $\underline{0.5274}_{\pm 0.4618}$ |
| + *LS* | $0.7058_{\pm 0.1455}$ | $0.6770_{\pm 0.1730}$ | $0.4689_{\pm 0.1383}$ | $0.4888_{\pm 0.3020}$ | $0.4970_{\pm 0.2172}$ | $1.5043_{\pm 1.1311}$ | $0.6838_{\pm 0.5045}$ |
| + **TRin** | $\mathbf{0.7445}_{\pm 0.1677}$ | $\mathbf{0.7090}_{\pm 0.1947}$ | $\mathbf{0.4171}_{\pm 0.1478}$ | $\mathbf{0.6076}_{\pm 0.2880}$ | $\mathbf{0.6708}_{\pm 0.1880}$ | $\mathbf{0.2637}_{\pm 0.2754}$ | $\mathbf{0.1203}_{\pm 0.1148}$ |
| CBraMod | $0.6438_{\pm 0.1302}$ | $0.5803_{\pm 0.1785}$ | $0.5780_{\pm 0.1416}$ | $0.3587_{\pm 0.2371}$ | $0.5040_{\pm 0.1293}$ | $1.2050_{\pm 0.7834}$ | $0.5584_{\pm 0.3632}$ |
| + *LS* | $\underline{0.6464}_{\pm 0.1372}$ | $\underline{0.5824}_{\pm 0.1870}$ | $\underline{0.5186}_{\pm 0.1413}$ | $\mathbf{0.4781}_{\pm 0.2345}$ | $\underline{0.5487}_{\pm 0.1449}$ | $\underline{0.5938}_{\pm 0.1885}$ | $\underline{0.2952}_{\pm 0.1003}$ |
| + **TRin** | $\mathbf{0.6869}_{\pm 0.1656}$ | $\mathbf{0.6354}_{\pm 0.2087}$ | $\mathbf{0.5156}_{\pm 0.1786}$ | $\underline{0.4655}_{\pm 0.3093}$ | $\mathbf{0.5846}_{\pm 0.1740}$ | $\mathbf{0.5894}_{\pm 0.8244}$ | $\mathbf{0.2916}_{\pm 0.3837}$ |

and 199% in FDS$_{norm}$ across both datasets. These results underscore the critical need for temporal regularization in DL-based BCIs.

At the same time, evaluation results demonstrate TRin's consistent improvements in temporal stability while enhancing classification performance in all experimental configurations. Across datasets and DL models, TRin achieves an average reduction of **56.27**% in HFSE, **56.18**% in FDS$_{norm}$, and improves classification accuracy by an average of **4.41**% compared to the baselines. Statistical analysis using paired t-tests confirmed significant improvements for HFSE and FDS$_{norm}$ ($p < 0.01$) and FSE ($p < 0.05$), with Cohen's d effect sizes demonstrating large practical significance (1.8-2.9).

**Cognitive Load Tasks:** TRin shows the most significant improvement on cognitive load tasks, represented by the Mental Workload dataset. All model architectures incorporating TRin demonstrate improvements in both temporal stability and classification accuracy compared to baseline variants. Deformer achieves the highest accuracy of 76.45%, while relatively reducing HFSE by 71.60% and FDS$_{norm}$ by 70.58% compared to the baseline. Considering both temporal and classification performance, all models with TRin outperform the baseline variants by an average of 42.21% in FSE.

**Emotion Recognition Tasks:** On the SEED dataset, TRin also benefits classification performance while significantly improving temporal stability. Among the models, EEGNet with TRin achieves the most balanced improvement, yielding a 4.01% increase in accuracy alongside a 74.39% re-

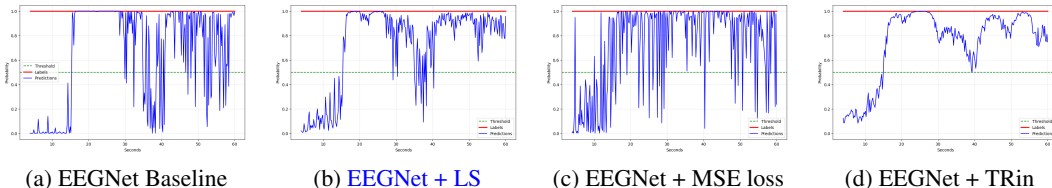

(a) EEGNet Baseline     (b) EEGNet + LS     (c) EEGNet + MSE loss     (d) EEGNet + TRin

Figure 4: Comparison of EEGNet predictions on Mental Workload dataset: (a), (b), (c), (d) represent baseline model, baseline with label smoothing, MSE loss variant, and TRin variant, respectively. Blue, red, and green represent prediction, ground-truth, and threshold, respectively.

duction in HFSE. Deformer demonstrates the strongest temporal improvement, reducing HFSE by 85.12%; CBraMod attains highest relative classification accuracy improvement of 6.69%. Therefore, TRin is effective for both simple and complex architectures.

### 3.2.1 COMPARISON WITH MSE LOSS VARIANTS

TRin consistently outperforms MSE loss variants on both temporal stability and classification performance, achieving on average 29.41% higher FSE scores across all datasets and models. Interestingly, for the SEED dataset, MSE loss variants of EEGNet and Deformer show improvements in temporal stability over their baselines, while such benefits are absent in other configurations. Further analysis (Appendix Table 10) reveals that MSE loss variants behave similarly to baselines when training data are completely randomly shuffled. This result highlights the effectiveness of our proposed temporal reserved clip-based shuffling and dynamic training strategies.

### 3.2.2 COMPARISON WITH LABEL SMOOTHING VARIANTS

The result shows that label smoothing approach is effective for temporal regularization, but still less consistent than TRin except for TSception. In contrast to TRin, label smoothing variants achieve only negligible classification accuracy improvement, even less than the MSE loss variants on SEED dataset. Therefore, despite the effectiveness of label smoothing for temporal regularization, it may not enable models to encode generalizable features that are neurophysiological meaningful.

### 3.3 VISUALIZATION

The visualization in Figure 4 corresponds to the results reported in Table 1 for EEGNet on the Mental Workload dataset. The graphs are taken from the same subject and the same trial, lasting 60 seconds with a feedback rate of 5 Hz. Both baseline and MSE variants exhibit inconsistent and unstable sudden jumps over time. LS variant shows effective temporal regularization, but still incapable to suppress high frequency fluctuations. TRin significantly enhances temporal stability by producing smoother, gradual transitions of mental states while simultaneously improving classification accuracy. A potential reason for TRin's improved classification is that model aligns features between temporally adjacent samples, thereby suppressing overconfident, unstable false predictions. More visualization examples are provided in Appendix N.

## 4 CONCLUSION

We identified prediction fluctuations as a critical challenge in deep learning-based BCIs and quantified them through two novel metrics: **Frequency-Weighted Spectral Entropy** and **First-Order Difference Standard Deviation**. To address this issue, we proposed **TRin**: Temporal Robustness integrated BCI, which mitigates fluctuations via a tailored regularization loss and a curriculum dynamic training strategy. Comprehensive evaluations on three datasets and four representative models demonstrate that TRin substantially enhances temporal stability while also improving classification accuracy. These results highlight the potential of TRin to enable more reliable real-time applications. We hope that future work will extend the principles of temporal robustness to a broader range of BCI paradigms.

## 5 REPRODUCIBILITY STATEMENT

To support reproducibility, we provide an anonymous code repository, `https://anonymous.4open.science/r/TRin`, containing all scripts for data processing, model training, and experiment replication. Table 3 details the hyperparameter configurations for all datasets and models. Preprocessing steps for each dataset is described in Appendix B.

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

# A    RELATED WORK

**Neurofeedback Training**    In neurofeedback training, pretrained machine learning or deep learning models infer users' mental states from real-time EEG signals, and the predictions are presented through graphical indicators or visual effects. EEG headsets record brain signals while ensuring channel positions and sampling rates remain consistent with the training data. The real-time raw EEG signals undergo preprocessing, including, frequency band filtering, artifact removal, and signal segmentation. Belinskaia et al. (2020) suggest that delayed neurofeedback impedes training. To minimize delay in real-time applications (Teo et al., 2021; Jochumsen et al., 2019), 200ms between two consecutive predictions delay is typically set (i.e. 95% overlap rate with 4 seconds window size). This enables users to gain immediate feedback on their mental states and to guide themselves in regulating behavior. For example, an attention-level classifier can detect focus levels in ADHD patients, helping them assess treatment effectiveness and adjust activities (Wender et al., 2001; Lim et al., 2019). Similarly, an emotion classifier can monitor arousal levels in GAD patients (Spielberger, 2013; Huang et al., 2021). Personalized approaches have shown even greater effectiveness (Tan et al., 2024). In such scenarios, the effectiveness of a BCI system primarily depends on the correctness and interpretability of its predictions. Sudden and inconsistent fluctuations in predictions can confuse users, preventing them from making effective interventions.

**Mental State consistency**    Human cognitive and affective states are widely recognized as relatively slow-changing and continuous rather than instantaneous. In affective science, foundational work on emotion dynamics demonstrates that affective states correlate with multiple aspects of the organism's continuously changing internal state (Damasio et al., 2000). Subsequent studies further show that the rate of emotional change can range from seconds to hours (Verduyn et al., 2015). Thus, abrupt changes in emotion within milliseconds contradict the continuous nature of affect. Similar principles apply to cognitive states. Recent research on attention reveals that it is a temporally structured process (Kasten et al., 2024), oscillating at a relatively slow period (about 0.07 Hz), which reinforces the continuous nature of mental states. Studies on time-varying functional connectivity (e.g., default mode network, task positive network, frontoparietal control network) varying across different levels of attentional focus (Seeburger et al., 2024), indicating that cognitive attentional mechanisms exhibit slow-changing dynamics. Together, these findings converge on the view that both cognition and affect should be modeled as temporally structured processes. For computational modeling, this motivates approaches that explicitly guide models to learn temporal consistency.

**Deep Learning based BCI**    Deep learning (DL) models have gained prominence for their powerful representational capacity to capture rich information from input data. Compared to traditional machine learning models, this capability benefits BCI systems to achieve superior classification performance and enable real-time predictions. EEGNet (Lawhern et al., 2018) established the foundation for DL-based BCIs by introducing compact convolutional architectures specifically designed for EEG data. As a lightweight yet effective model, it achieved close to state-of-the-art classification performance with minimal parameters. Subsequent studies expanded on this foundation by enhancing representational capacity with more complex architectures: Song et al. (Song et al., 2018) introduced dynamical graph convolutional networks for emotion recognition, while Sakhavi et al. (Sakhavi et al., 2018) emphasized temporal information learning. TSception (Ding et al., 2023b) further advanced the field by leveraging multi-scale temporal dynamics through temporal and spatial convolutions, capturing both short- and long-term dependencies. More recent work has explored geometric deep learning frameworks (Ju & Guan, 2022) and local-global graph representations (Ding et al., 2023a), while transformer-based models such as Deformer (Ding et al., 2025a) now represent the state of the art, utilizing attention mechanisms to model long-term temporal coarse-to-fine dynamics. Current trend in foundation models like CBraMod (Wang et al., 2025) further advance the field by unsupervised pre-training on large-scale EEG datasets and fine-tuning on downstream tasks.

**Temporal Regularization in DL-based BCIs**    Although state-of-the-art DL-based BCIs improve classification accuracy and incorporate temporal considerations within segments, they fail to address inconsistencies between consecutive segments, as illustrated in Fig. 1 and further visualized in Appendix N. If the features extracted by DL-models are indeed aligned with neuroscientific principles, their predictions should also reflect fundamental characteristics of neural processes, namely that mental states are slow-changing and continuous. However, these models often perform com-

pletely opposite in this regard, especially from models with stronger representational capacity like Deformer. This contradiction highlights that while advanced architectures may capture richer discriminative features, they tend to neglect the inherent temporal continuity of cognitive and affective processes. To the best of our knowledge, we are the first to systematically study the temporal stability of DL-based BCIs and propose a principled framework for both evaluating and mitigating prediction fluctuations in DL-based BCIs. Although previous work (Phan et al., 2023) has proposed a temporal regularization, they aim to force the consecutive epochs' prediction loss (cross entropy) to be as close as possible. Thus, their *temporal* donates training progress and is totally different from what we are discussing here.

Some temporal regularization methods may hint at the idea from other domains, such as computer vision or general tasks with temporal dependencies. Its objective is to encourage consistent predictions over time, which is typically achieved by minimizing differences of predictions (Dileo et al., 2023), graph embeddings (Xu et al., 2021), mutual conditional probabilities (Varghese et al., 2021), cosine similarity between latent states (Zhao et al., 2023), and employing recurrent architectures (García-Durán et al., 2018). However, these approaches are not directly applicable to BCI tasks. Directly minimizing predictions, embeddings, or probabilities may lead to overfitting to specific temporal patterns (Appendix H.2), while recurrent architectures depend on past predictions or segments, introducing feedback latency.

**Fluctuation Metrics and Evaluation**  Classification metrics have been extensively discussed in the literature on DL-based BCI systems, whereas metrics for evaluating temporal stability remain under-explored. Several existing measures appear potentially useful for this purpose. For instance, in emotion regression tasks (Ding et al., 2025b), RMSE, PCC, and CCC are commonly employed. However, RMSE is not a suitable indicator of temporal stability because it evaluates predictions point-wise without considering sequential dependencies. Most EEG datasets (Koelstra et al., 2012; Zheng & Lu, 2015; Chen et al., 2023; Shin et al., 2018; Zyma et al., 2019) for cognitive and affective tasks utilize constant per-trial ground-truth labels. As demonstrated in Theorem 3 and Theorem 4, both PCC and CCC are insufficient for assessing temporal stability in such scenarios. To address this limitation, we propose new metrics specifically designed to evaluate the temporal stability of model predictions when only constant ground-truth labels are available. In other fields, there are inspiring metrices like flow-based or perceptual consistency to rate temporal coherence of video predictions (Zhang et al., 2022). However, they are still not applicable in EEG-based tasks given their fundamental differences in ground-truth and output.

# B  DATASET DETAILS

**Mental Workload Dataset:** The Mental Workload dataset (Zyma et al., 2019) contains EEG recordings from 36 subjects performing cognitive tasks such as serial subtraction. Each trial lasts 60 seconds with one trial per subject, and the last 60 seconds of each subject's baseline EEG during rest are treated as low-workload data (Ding et al., 2025a). The officially preprocessed data include 19 EEG channels downsampled to 500 Hz. For model input, the signals are further segmented into 4-second windows with 70% overlap.

**SEED Dataset:** The SJTU SEED dataset (Zheng & Lu, 2015) contains EEG recordings from 15 subjects who watched film clips across three sessions. Each trial lasts 3–5 minutes, with 15 trials per session, and positive, neutral, and negative emotion labels are evenly distributed. In this study, only the first session is used. From the 15 trials in the first session, we retain five neutral trials per subject and keep the last three minutes of each trial to balance the positive and negative emotion classes. The officially preprocessed data include 62 EEG channels downsampled to 200 Hz with a 0–75 Hz bandpass filter. The signals are then segmented into 4-second windows with 70% overlap.

**Attention Dataset:** The Attention dataset (Shin et al., 2018) includes EEG recordings from 26 subjects performing the Discrimination/Selection Response (DSR) task to measure cognitive attention. Each attention trial lasts 40 seconds, while each rest trial lasts 20 seconds, with a total of 36 trials per subject. Following (Ding et al., 2023a), only the first 20 seconds of each attention trial are used to balance attention and rest classes. The original EEG signals were recorded from 28 channels at a sampling rate of 1000 Hz. Preprocessing included bandpass filtering from 0.5–50 Hz, Electrooculography (EOG) artifact removal using Independent Component Analysis (ICA) implemented in the

MNE toolbox (Gramfort et al., 2014), and subsequent downsampling to $200\,\text{Hz}$ (Delvigne et al., 2022). Finally, the data are segmented into 4-second windows with 90% overlap to ensure sufficient consecutive samples for training.

## C  MODEL DETAILS

**SVM** (Zheng & Lu, 2015) serves as a traditional machine learning baseline. It employs an RBF kernel with frequency-domain features extracted from delta, theta, alpha, beta, and gamma bands. Since SVM does not exhibit fluctuation issues in real-time applications, it is chosen as the baseline model to represent a standard acceptable fluctuation level.

**EEGNet** (Lawhern et al., 2018) is a compact convolutional neural network specifically designed for EEG signals, utilizing deep and separable convolutions. It achieves close to state-of-the-art performance with minimal parameters, making it particularly appealing for real-time applications due to its efficiency and low computational cost.

**TSception** (Ding et al., 2023b) is a multi-scale temporal-spatial convolutional network composed of dynamic temporal layers, asymmetric spatial layers, and adaptive fusion mechanisms. This architecture efficiently extracts discriminative mental-state features, demonstrating strong robustness for emotion recognition tasks.

**Deformer** (Ding et al., 2025a) introduces a Hierarchical Coarse-to-Fine Transformer (HCT) block that integrates a Fine-grained Temporal Learning (FTL) branch into Transformers, together with a Dense Information Purification (DIP) module. This design effectively decodes coarse-to-fine temporal patterns in EEG signals. As a result, Deformer represents the current state-of-the-art transformer-based DL model for BCI applications.

**CBraMod** (Wang et al., 2025) proposes a criss-cross transformer architecture with parallel spatial and temporal attention mechanisms, specifically designed for EEG signal processing. Through self-supervised pre-training on the large-scale Temple University Hospital EEG Corpus (TUEG), CBraMod learns generalizable representations via patch-based masked EEG reconstruction, enabling a robust foundation model for EEG decoding applications.

## D  TRAINING CONFIGURATION

### D.1  ALGORITHM DETAILS OF DYNAMIC TRAINING STRATEGY

The dynamic TRin training strategy (Algorithm 1) is a curriculum learning approach adapted from the static version. In addition to the static hyperparameters clipRate and $\alpha_s$, four more hyperparameters are introduced to control the dynamic schedule: $\alpha_i$, $\alpha_g$, $\text{clip}_{min}$, and $\text{clip}_d$. At the start of training, a smaller initial regularization coefficient $\alpha$ value $\alpha_i$ is applied, allowing the model to overcome the initial regularization barrier. The $\alpha$ value will increase with the growth rate $\alpha_g$ until reaching maximal $\alpha_s$, thus gradually introducing temporal stability considerations. Nevertheless, when $\alpha$ value approaches maximum, the model may still encounter regularization constraint. To counter this, we gradually reduce the clip size at a decay rate $\text{clip}_d$, thereby increasing randomness in the training data to benefit training of cross entropy. Minimal clip size is set to $\text{clipMin} = \text{clip}_{min} \times \text{clipSize}$.

### D.2  MEAN SQUARED ERROR LOSS VARIANT

Denote $y_i$ as the ground truth label, while $\hat{y}_i$ denotes the predicted probability for the positive class. Given that all the datasets involved in experiments are binary classification tasks, focusing solely on the positive class is sufficient for training. The MSE loss is defined as:

$$\mathcal{L}_{MSE} = \frac{1}{N} \sum_{i=1}^{N} (y_i - \hat{y}_i)^2 \tag{12}$$

To illustrate the temporal restriction property of MSE loss, we can rewrite it as:

$$\mathcal{L}_{MSE} = \mathbb{E}[(\hat{y} - y)^2] = \underbrace{(\mathbb{E}[\hat{y}] - y)^2}_{\text{Bias}^2} + \underbrace{\mathbb{E}[(\hat{y} - \mathbb{E}[\hat{y}])^2]}_{\text{Variance}} \tag{13}$$

---

**Algorithm 1** Dynamic TRin Training

---

1: Set hyperparameters: $E_{\text{total}}$, batchSize, $\alpha_i$, $\alpha_g$, $\alpha_s$, clipRate, $\text{clip}_d$, $\text{clip}_{min}$
2: Initialize: $\alpha^{(0)} = \alpha_i$, $\text{clipSize}^{(0)} = \text{clipRate} \times \text{batchSize}$, $\text{clipMin} = \text{clip}_{min} \times \text{clipSize}^{(0)}$
3: Initialize network parameters $\theta$
4: **for** epoch $e = 1$ to $E_{\text{total}}$ **do**
5:     **for** batch $(X, Y)$ in training data **do**
6:         Forward pass: $\hat{Y} = f_\theta(X)$
7:         **for** each clip in batch **do**
8:             $\mathcal{L}_{clip}^{(i)} = \mathcal{L}_{CE} + \alpha^{(e-1)}\mathcal{L}_R$
9:         **end for**
10:        $\mathcal{L} = \text{mean}(\mathcal{L}_{clip}^{(i)})$
11:        Backward pass $\nabla_\theta \mathcal{L}$ and update parameters
12:     **end for**
13:     $\alpha^{(e)} = \min(\alpha_s, \alpha^{(e-1)} \times \alpha_g)$
14:     $\text{clipSize}^{(e)} = \max(\text{clipMin}, \text{roundToFactor}(\text{clipSize}^{(e-1)} \times \text{clip}_d, \text{batchSize}))$
15:     Reshuffle training data according to $\text{clipSize}^{(e)}$
16: **end for**
17: **Note:** $\text{roundToFactor}(x, y)$ returns the factor of $y$ closest to $x$

---

where $\mathbb{E}[\cdot]$ is the expectation operator. Therefore, MSE loss's Variance term naturally punishes the volatility of the output, thus promoting smoothness.

### D.3 TRAINING HYPERPARAMETER SETTINGS

All DL-based models are trained using the Adam optimizer with a learning rate of $1 \times 10^{-3}$, a batch size of 64, and a maximum of 200 training epochs, except 50 epochs for CBraMod following the original paper (Wang et al., 2025). All models process 4-second segments, with the training overlap rate set to 0.7 for the Mental Workload and SEED datasets and 0.9 for the Attention dataset. During inference, a 200 ms sliding window with a 95% overlap rate is applied across all datasets. For the dynamic training strategy, all models use parameters $\alpha_i$ set to 80% of the maximum $\alpha_s$, and the minimal clip length is fixed to half of the clip size (i.e., $\text{clip}_{min} = 0.5$). The specialized training configurations for each model and dataset combination are provided in Table 3. Baseline and label smoothing (LS) varients are identical to each other except for the modification of the training ground truth. All models are trained and tested on an AMD EPYC$^{\text{TM}}$ 75F3 CPU and NVIDIA A100 Tensor Core GPUs.

### D.4 MODEL-SPECIFIC ARCHITECTURE DETAILS

For **SVM**, we use an RBF kernel with frequency-domain differential entropy features extracted from the delta, theta, alpha, beta, and gamma bands. The SVM hyperparameters are set to $C = 1.0$ and $\gamma = scale$. All available channels are used: 19 channels for the Mental Workload dataset, 62 channels for SEED, and 28 channels for Attention.

For **EEGNet** (Lawhern et al., 2018), hyperparameters are configured as $C1 = 64$, $F1 = 8$, and $D = 2$. We employ all available channels for each dataset (19 for Mental Workload, 62 for SEED, and 28 for Attention).

For **TSception** (Ding et al., 2023b), hyperparameters are set to $T = 64$ and $S = 64$. Due to the model's spatial-aware design, only electrodes from a single hemisphere (left or right) are used; consequently, we use 16, 54, and 24 channels for the Mental Workload, SEED, and Attention datasets, respectively.

For **Deformer** (Ding et al., 2025a), all available channels are included for each dataset. The model hyperparameters are $AT = 16$ and $\text{num\_layers} = 6$ across datasets. Kernel lengths are set per dataset as follows: 51 for Mental Workload, 21 for SEED, and 21 for Attention.

Table 3: The hyperparameter settings for different models across datasets are summarized in Table 3. MWL, SEED, and ATTEN refer to the Mental Workload, SJTU SEED, and Attention datasets, respectively. The term Dynamic indicates whether a dynamic training strategy is used during training. ClipRate refers to the ratio of clip size to batch size. $\alpha_s$ is the maximum temporal regularization alpha value, $\alpha_g$ is the alpha growth rate, and $\text{clip}_d$ represents the clip descent rate.

| Dataset | Model | Variant | Overlap | Dropout | Dynamic | clipRate | $\alpha_{\mathbf{s}}$ | $\text{clip}_d$ | $\alpha_{\mathbf{g}}$ |
|---|---|---|---|---|---|---|---|---|---|
| MWL | SVM | N/A | 0.7 | N/A | N/A | N/A | N/A | N/A | N/A |
| | EEGNet | + Baseline/LS | 0.7 | 0.25 | False | N/A | N/A | N/A | N/A |
| | | + MSE Loss | 0.7 | 0.25 | True | 0.125 | N/A | 0.75 | N/A |
| | | + TRin | 0.7 | 0.25 | True | 0.125 | 20 | 0.75 | 1.25 |
| | TSception | + Baseline/LS | 0.7 | 0.25 | False | N/A | N/A | N/A | N/A |
| | | + MSE Loss | 0.7 | 0.25 | True | 0.25 | N/A | 0.95 | N/A |
| | | + TRin | 0.7 | 0.25 | True | 0.25 | 80 | 0.95 | 1.05 |
| | Deformer | + Baseline/LS | 0.7 | 0.25 | False | N/A | N/A | N/A | N/A |
| | | + MSE Loss | 0.7 | 0.25 | True | 0.25 | N/A | 0.95 | N/A |
| | | + TRin | 0.7 | 0.25 | True | 0.25 | 100 | 0.95 | 1.05 |
| | CBraMod | + Baseline/LS | 0.7 | 0.25 | False | N/A | N/A | N/A | N/A |
| | | + TRin | 0.7 | 0.25 | True | 0.25 | 50 | 0.95 | 1.05 |
| SEED | SVM | N/A | 0.7 | N/A | N/A | N/A | N/A | N/A | N/A |
| | EEGNet | + Baseline/LS | 0.7 | 0.5 | False | N/A | N/A | N/A | N/A |
| | | + MSE Loss | 0.7 | 0.5 | True | 0.25 | N/A | 0.95 | N/A |
| | | + TRin | 0.7 | 0.5 | True | 0.25 | 32 | 0.95 | 1.05 |
| | TSception | + Baseline/LS | 0.7 | 0.5 | False | N/A | N/A | N/A | N/A |
| | | + MSE Loss | 0.7 | 0.5 | True | 0.125 | N/A | 0.95 | N/A |
| | | + TRin | 0.7 | 0.5 | True | 0.125 | 16 | 0.95 | 1.05 |
| | Deformer | + Baseline/LS | 0.7 | 0.5 | False | N/A | N/A | N/A | N/A |
| | | + MSE Loss | 0.7 | 0.5 | True | 0.25 | N/A | 0.95 | N/A |
| | | + TRin | 0.7 | 0.5 | True | 0.25 | 64 | 0.95 | 1.05 |
| | CBraMod | + Baseline/LS | 0.7 | 0.5 | False | N/A | N/A | N/A | N/A |
| | | + TRin | 0.7 | 0.5 | True | 0.25 | 64 | 0.95 | 1.05 |
| ATTEN | SVM | N/A | 0.9 | N/A | N/A | N/A | N/A | N/A | N/A |
| | EEGNet | + Baseline/LS | 0.9 | 0.5 | False | N/A | N/A | N/A | N/A |
| | | + MSE Loss | 0.9 | 0.5 | True | 0.0625 | N/A | 0.75 | N/A |
| | | + TRin | 0.9 | 0.5 | True | 0.0625 | 1 | 0.75 | 1.25 |
| | TSception | + Baseline/LS | 0.9 | 0.5 | False | N/A | N/A | N/A | N/A |
| | | + MSE Loss | 0.9 | 0.5 | True | 0.0625 | N/A | 0.9 | N/A |
| | | + TRin | 0.9 | 0.5 | True | 0.0625 | 32 | 0.9 | 1.1 |
| | Deformer | + Baseline/LS | 0.9 | 0.5 | False | N/A | N/A | N/A | N/A |
| | | + MSE Loss | 0.9 | 0.5 | True | 0.0625 | N/A | 0.9 | N/A |
| | | + TRin | 0.9 | 0.5 | True | 0.0625 | 16 | 0.9 | 1.1 |
| | CBraMod | + Baseline/LS | 0.9 | 0.5 | False | N/A | N/A | N/A | N/A |
| | | + TRin | 0.9 | 0.5 | True | 0.0625 | 8 | 0.9 | 1.1 |

For **CBraMod** (Wang et al., 2025), all available channels are included for each dataset. All model architecture hyperparameters are identical to the original paper. The pretrained model from the original publication is loaded and finetuned for 50 epochs for each dataset.

## D.5 FSE METRICS PARAMETERS

As described in Section 2.1, a Hanning window is applied to the error signal to reduce spectral leakage. Only the first half of the frequency range is considered because of the symmetry of the DFT applied to real-valued signals. After the DFT, the frequency range is rescaled to $(0, 1 - \epsilon]$ with $\epsilon = 10^{-5}$. The softmax temperature for the final FSE calculation is $\alpha = 2.0$.

Table 4: Performance comparison on Attention dataset. Results show mean ± standard deviation over cross-validation folds. The best performance is highlighted in bold while the second best performance is highlighted in underline.

| Model | ACC($\uparrow$) | F1-macro($\uparrow$) | RMSE($\downarrow$) | PCC($\uparrow$) | FSE($\uparrow$) | HFSE($\downarrow$) | FDS$_{norm}$($\downarrow$) |
|---|---|---|---|---|---|---|---|
| SVM | $0.6108_{\pm0.0788}$ | $0.5628_{\pm0.1283}$ | $0.5367_{\pm0.0749}$ | $0.3233_{\pm0.1603}$ | $0.5313_{\pm0.0647}$ | $0.3200_{\pm0.0927}$ | $0.3551_{\pm0.1029}$ |
| EEGNet | $0.6820_{\pm0.0935}$ | $\underline{0.6505}_{\pm0.1333}$ | $\underline{0.4736}_{\pm0.0898}$ | $0.5068_{\pm0.1649}$ | $0.6277_{\pm0.0903}$ | $0.2035_{\pm0.0690}$ | $0.2381_{\pm0.0776}$ |
| + *MSEloss* | $\underline{0.6865}_{\pm0.1143}$ | $0.6444_{\pm0.1712}$ | $0.4752_{\pm0.1126}$ | $\underline{0.5133}_{\pm0.1833}$ | $\underline{0.6392}_{\pm0.1012}$ | $\underline{0.1898}_{\pm0.0741}$ | $\underline{0.2184}_{\pm0.0878}$ |
| + *LS* | $0.6889_{\pm0.0985}$ | $0.6552_{\pm0.1450}$ | $0.4675_{\pm0.0948}$ | $0.5163_{\pm0.1690}$ | $0.6318_{\pm0.0854}$ | $0.1863_{\pm0.0619}$ | $0.2183_{\pm0.0733}$ |
| + TRin | $\mathbf{0.6972}_{\pm0.0928}$ | $\mathbf{0.6722}_{\pm0.1294}$ | $\mathbf{0.4528}_{\pm0.0779}$ | $\mathbf{0.5187}_{\pm0.1552}$ | $\mathbf{0.6417}_{\pm0.0838}$ | $\mathbf{0.1529}_{\pm0.0387}$ | $\mathbf{0.1843}_{\pm0.0476}$ |
| TSception | $\underline{0.7015}_{\pm0.0808}$ | $0.6823_{\pm0.1120}$ | $0.5199_{\pm0.0766}$ | $0.4532_{\pm0.1545}$ | $0.4661_{\pm0.0809}$ | $1.1593_{\pm0.3673}$ | $0.9575_{\pm0.2692}$ |
| + *MSEloss* | $0.6974_{\pm0.0816}$ | $0.6762_{\pm0.1167}$ | $0.5249_{\pm0.0786}$ | $0.4421_{\pm0.1545}$ | $0.4631_{\pm0.0876}$ | $1.0911_{\pm0.3734}$ | $0.9089_{\pm0.2803}$ |
| + *LS* | $0.7003_{\pm0.0811}$ | $\underline{0.6833}_{\pm0.1054}$ | $\underline{0.5082}_{\pm0.0704}$ | $\mathbf{0.4679}_{\pm0.1548}$ | $\underline{0.4765}_{\pm0.1027}$ | $\underline{0.8947}_{\pm0.2076}$ | $\mathbf{0.7412}_{\pm0.1388}$ |
| + TRin | $\mathbf{0.7022}_{\pm0.0821}$ | $\mathbf{0.6851}_{\pm0.1068}$ | $\mathbf{0.5063}_{\pm0.0780}$ | $\underline{0.4654}_{\pm0.1482}$ | $\mathbf{0.5082}_{\pm0.1007}$ | $\mathbf{0.8874}_{\pm0.2574}$ | $\underline{0.7527}_{\pm0.1891}$ |
| Deformer | $\underline{0.8366}_{\pm0.0828}$ | $0.8302_{\pm0.1020}$ | $0.3638_{\pm0.0965}$ | $0.7193_{\pm0.1412}$ | $0.6452_{\pm0.1308}$ | $0.9260_{\pm0.2808}$ | $0.8753_{\pm0.2294}$ |
| + *MSEloss* | $0.8317_{\pm0.0867}$ | $0.8249_{\pm0.1027}$ | $0.3642_{\pm0.0961}$ | $0.7168_{\pm0.1340}$ | $0.6360_{\pm0.1175}$ | $0.9211_{\pm0.2607}$ | $0.8837_{\pm0.2230}$ |
| + *LS* | $0.8346_{\pm0.0737}$ | $\underline{0.8308}_{\pm0.0840}$ | $\mathbf{0.3292}_{\pm0.0808}$ | $\underline{0.7335}_{\pm0.1126}$ | $\underline{0.6869}_{\pm0.1143}$ | $\underline{0.7009}_{\pm0.1795}$ | $\underline{0.6638}_{\pm0.1562}$ |
| + TRin | $\mathbf{0.8380}_{\pm0.0874}$ | $\mathbf{0.8325}_{\pm0.1021}$ | $\underline{0.3389}_{\pm0.1014}$ | $\mathbf{0.7393}_{\pm0.1424}$ | $\mathbf{0.7153}_{\pm0.1188}$ | $\mathbf{0.6055}_{\pm0.1620}$ | $\mathbf{0.5732}_{\pm0.1465}$ |
| CBraMod | $0.6432_{\pm0.0735}$ | $0.6155_{\pm0.0969}$ | $0.5725_{\pm0.0632}$ | $0.3351_{\pm0.1496}$ | $0.3382_{\pm0.1008}$ | $1.3764_{\pm0.3332}$ | $1.4612_{\pm0.3520}$ |
| + *LS* | $0.6457_{\pm0.0754}$ | $\mathbf{0.6247}_{\pm0.0954}$ | $\underline{0.5316}_{\pm0.0645}$ | $\underline{0.3692}_{\pm0.1540}$ | $\underline{0.4136}_{\pm0.0962}$ | $\underline{0.8914}_{\pm0.1564}$ | $\underline{1.0224}_{\pm0.1765}$ |
| + TRin | $\mathbf{0.6512}_{\pm0.0912}$ | $\underline{0.6198}_{\pm0.1247}$ | $\mathbf{0.5223}_{\pm0.0755}$ | $\mathbf{0.3956}_{\pm0.1671}$ | $\mathbf{0.5122}_{\pm0.0963}$ | $\mathbf{0.5793}_{\pm0.1613}$ | $\mathbf{0.6754}_{\pm0.1837}$ |

# E    ATTENTION DATASET RESULTS

The quantitative results on the Attention dataset are reported in Table 4. For attention detection tasks, TRin consistently improves temporal stability across all models while preserving high classification accuracy. Deformer with TRin achieves significant (p<0.05) stability gains (34.62% reduction in HFSE and 34.51% reduction in FDS$_{norm}$). On average across models, TRin reduces the fluctuations by an average of 27.64% in HFSE and 26.16% in FDS$_{norm}$. Owing to the relatively short trial length, TRin with clip-shuffling inherits a disadvantage in classification accuracy. Nevertheless, all DL-based models with TRin maintain high classification accuracy. All models improve FSE by an average of 7.38%, further demonstrating the effectiveness of TRin in balancing temporal stability and classification performance for attention detection tasks.

# F    THEORETICAL ANALYSIS

## F.1    THEORETICAL ANALYSIS OF TEMPORAL REGULARIZATION LOSS

### F.1.1    CONVEXITY ANALYSIS OF TEMPORAL REGULARIZATION LOSS

**Theorem 1.** *Let $y_{t,c}$ be one-hot encoded, and define the loss function $\mathcal{L} = \mathcal{L}_{CE} + \alpha\mathcal{L}_R$ where*

$$\mathcal{L}_{CE} = -\frac{1}{T}\sum_{t=1}^{T}\sum_{c=1}^{C} y_{t,c}\log(\hat{y}_{t,c}), \quad \mathcal{L}_R = \frac{1}{(T-1)(C-1)}\sum_{t=2}^{T}\sum_{c=2}^{C}(\hat{y}_{t,c} - \hat{y}_{t-1,c})^2,$$

*and $\alpha > 0$. Then $\mathcal{L}$ is convex with respect to $\hat{\mathbf{y}}$ on*

$$\mathcal{D} = \prod_{t=1}^{T}\Delta^{C-1}, \quad \Delta^{C-1} = \left\{\mathbf{p} \in \mathbb{R}^C \mid p_c > 0, \sum_{c=1}^{C} p_c = 1\right\}.$$

*Proof.* The domain $\mathcal{D}$ is convex as a Cartesian product of simplices $\Delta^{C-1}$.

Given the one-hot $y_{t,c}$, $\mathcal{L}_{CE} = -\frac{1}{T}\sum_{t=1}^{T}\log(\hat{y}_{t,c_t^*})$ for the true class $c_t^*$ at $t$. Each function $-\log(\hat{y}_{t,c_t^*})$ is convex on $\Delta^{C-1}$ because of the convexity of $-\log x$ on $(0, \infty)$. Thus $\mathcal{L}_{CE}$ is convex on $\mathcal{D}$.

Each term $(\hat{y}_{t,c} - \hat{y}_{t-1,c})^2$ in $\mathcal{L}_R$ is convex because it is the composition of the affine function $\hat{\mathbf{y}} \mapsto \hat{y}_{t,c} - \hat{y}_{t-1,c}$ and convex function $g(z) = z^2$. Nonnegative scaling and summation preserve convexity, so $\mathcal{L}_R$ is convex on $\mathbb{R}^{T \times C}$, and hence on $\mathcal{D}$.

Since $\mathcal{L}_{CE}$ and $\mathcal{L}_R$ are convex on $\mathcal{D}$ and $\alpha > 0$, their sum $\mathcal{L}$ is convex on $\mathcal{D}$. $\qquad\square$

F.1.2  STABILITY GUARANTEES OF TEMPORAL REGULARIZATION LOSS

According to Neural Tangent Kernel (NTK) theory (Jacot et al., 2018), in the NTK regime (the network width is infinite and the NTK $\Theta$ remains constant and positive definite during training), the loss function $\mathcal{L}$ is convex and $\mathcal{L}$ is bounded below. Consequently, as $t \to \infty$, the loss $\mathcal{L}$ converges to its global minimum, where $t$ is the training steps. By Theorem 1, $\mathcal{L}$ is convex. Note that $\mathcal{L}$ bounded below by 0, so we have the following corollary:

**Corollary 1.** *Consider a neural network trained via gradient descent under the Neural Tangent Kernel (NTK) regime. Let $\mathcal{L} = \mathcal{L}_{CE} + \alpha\mathcal{L}_R$ with $\alpha > 0$, where: $y_{t,c}$ is one-hot encoded for all $t$, and $\hat{y}_{t,c}$ are probabilities ($\hat{y}_{t,c} > 0$, $\sum_c \hat{y}_{t,c} = 1$). Then the loss $\mathcal{L}$ converges to its global minimum when $t \to \infty$.*

The loss function $\mathcal{L}$ has global infimum is 0, which is approached when the predicted outputs are nearly perfect at all time steps: $\hat{y}_{t,c} \to \delta_{c,c^*}$ for all $t, c$ (i.e., $\hat{y}_{t,c^*} \to 1$, and for all $c \neq c^*$, $\hat{y}_{t,c} \to 0$). This provides a theoretical guarantee for the stability and accuracy of training process.

Please note that though this corollary is still true for $\mathcal{L}_{CE}$ when softmax function is applied, it may not strictly hold when softmax function is used before $\mathcal{L}_R$. However, unlike $\mathcal{L}_{CE}$, $\mathcal{L}_R$ is convex on $\mathbb{R}^{T \times C}$ and does not require the assumption that $\hat{y}_t$ is probabilities.

**Theorem 2.** *Consider a time-series model where predictions $\hat{y}_{t,c}$ for classes $c = 2, \ldots, C$ and times $t = 1, \ldots, T$ are random variables. Assume increments $\Delta\hat{y}_{t,c} \equiv \hat{y}_{t,c} - \hat{y}_{t-1,c}$ satisfy:*

$$\Delta\hat{y}_{t,c} \mid \sigma^2 \overset{i.i.d.}{\sim} \mathcal{N}(0, \sigma^2), \quad \forall t \in \{2, \ldots, T\}, \forall c \in \{2, \ldots, C\} \tag{14}$$

*with $\hat{y}_{1,c}$ having improper uniform priors. Then:*

$$-\log p(\hat{\mathbf{y}} \mid \sigma^2) = \frac{1}{2\sigma^2} \sum_{c=2}^{C} \sum_{t=2}^{T} (\hat{y}_{t,c} - \hat{y}_{t-1,c})^2 + K(\sigma^2), \tag{15}$$

*where $K(\sigma^2)$ is constant in $\hat{\mathbf{y}} = \{\hat{y}_{t,c}\}$. Hence, the regularization*

$$\mathcal{L}_R \equiv \frac{1}{(T-1)(C-1)} \sum_{c=2}^{C} \sum_{t=2}^{T} (\hat{y}_{t,c} - \hat{y}_{t-1,c})^2$$

*is proportional to the negative log-prior in MAP estimation.*

*Proof.* The joint density of increments is:

$$p\left(\{\Delta\hat{y}_{t,c}\} \mid \sigma^2\right) = \prod_{c=2}^{C} \prod_{t=2}^{T} (2\pi\sigma^2)^{-1/2} \exp\left(-\frac{(\Delta\hat{y}_{t,c})^2}{2\sigma^2}\right) \tag{16}$$

Taking negative logarithm:

$$-\log p = \frac{(T-1)(C-1)}{2} \log(2\pi\sigma^2) + \frac{1}{2\sigma^2} \sum_{c,t} (\Delta\hat{y}_{t,c})^2 \tag{17}$$

The transformation from $(\{\hat{y}_{1,c}\}, \{\Delta\hat{y}_{t,c}\})$ to $\hat{\mathbf{y}}$ is linear with unit determinant Jacobian. Thus:

$$p(\hat{\mathbf{y}} \mid \sigma^2) = p(\{\Delta\hat{y}_{t,c}\} \mid \sigma^2) \cdot p(\{\hat{y}_{1,c}\})$$

Since $p(\{\hat{y}_{1,c}\})$ is constant, we have:

$$-\log p(\hat{\mathbf{y}} \mid \sigma^2) = \frac{1}{2\sigma^2} \sum_{c,t} (\hat{y}_{t,c} - \hat{y}_{t-1,c})^2 + K(\sigma^2)$$

where $K(\sigma^2) = \frac{(T-1)(C-1)}{2} \log(2\pi\sigma^2) + \text{const}$. Substitution yields:

$$\sum_{c,t} (\hat{y}_{t,c} - \hat{y}_{t-1,c})^2 = (T-1)(C-1)\mathcal{L}_R$$

proving proportionality to $-\log p(\hat{\mathbf{y}} \mid \sigma^2)$. $\qquad\square$

## F.2 TRADITIONAL METRICS

### F.2.1 DEFINITION OF TRADITIONAL METRICS

Traditional metrics are defined as:

$$\text{ACC} = \frac{TP + TN}{TP + FP + TN + FN} \tag{18}$$

$$\text{F1-macro} = \frac{1}{N} \sum_{i=1}^{N} \text{F1-score}_k \tag{19}$$

$$\text{F1-score}_k = \frac{2 \cdot TPk}{2 \cdot TPk + FPk + FNk} \tag{20}$$

$$\text{RMSE} = \sqrt{\frac{1}{T} \sum_{i=1}^{T} (y_i - \hat{y}_i)^2} \tag{21}$$

$$\text{PCC} = \frac{\text{Cov}(\hat{y}, y)}{\sqrt{\text{Var}(\hat{y}) \cdot \text{Var}(y)}} \tag{22}$$

where TP is the true positive, TN is the true negative, FP is the false positive, FN is the false negative, the subscript indicates the $k$-th class, $N$ is the number of classes, $T$ is the number of samples, Cov is the covariance, and Var is the variance.

### F.2.2 DEGRADATION OF CORRELATION

**Theorem 3.** *Let $\hat{y}$ be a constant label. For any prediction $y$, the covariance $\text{Cov}(y, \hat{y})$ is zero.*

*Proof.* $|\text{Cov}(\hat{y}, y)| \leq \sqrt{\text{Var}(\hat{y}) \cdot \text{Var}(y)} = 0$ □

**Theorem 4.** *Let $\hat{y} \in \{0, 1\}$ be a binary label with constant class frequencies $P(\hat{y} = 0) = a/(a+b)$ and $P(\hat{y} = 1) = b/(a + b)$. For any predicted probability $y \in [0, 1]$, the covariance $\text{Cov}(y, \hat{y})$ is expressed as:*

$$\text{Cov}(y, \hat{y}) = \frac{ab(\mu_1 - \mu_0)}{(a + b)^2} \tag{23}$$

*where $\mu_0 = \mathbb{E}[y \mid \hat{y} = 0]$ and $\mu_1 = \mathbb{E}[y \mid \hat{y} = 1]$.*

*Proof.*

$$\text{Cov}(y, \hat{y}) = \mathbb{E}[y\hat{y}] - \mathbb{E}[y]\mathbb{E}[\hat{y}]$$
$$= \frac{\mu_1 b(a+b) - (\mu_0 ab + \mu_1 b^2)}{(a+b)^2} = \frac{ab(\mu_1 - \mu_0)}{(a+b)^2}$$ □

Therefore, no matter how temporal unstable a prediction $y$ is, covariance stays the same if $E[y|\hat{y} = 1] - E[y|\hat{y} = 0]$ stays the same. In experiments, since we concatenate trials together to avoid zero nominator, $\sigma_y$ and $\sigma_{\hat{y}}$ have no physical meaning but just to normalize the covariance. In this case, PCC is not able to measure temporal stability.

## G SENSITIVITY ANALYSIS

### G.1 SENSITIVITY ANALYSIS ON CLIP SIZE

The clip-size sensitivity analysis on the Mental Workload dataset is reported in Tables 5 and 6. Validation results (Table 5) indicate that reducing the clip size tends to improve classification accuracy while progressively diminishing the benefits of temporal regularization. Conversely, larger clip sizes produce stronger temporal regularization but can mildly degrade classification performance when taken to an extreme. Accordingly, FSE is used as the primary criterion to balance classification accuracy and temporal-regularization effects. The selected optimal clip sizes by FSE are 0.125 for EEGNet and 0.25 for TSception and Deformer.

Table 5: Validation results for sensitivity analysis on Mental Workload dataset with different clip sizes. Results show mean ± standard deviation over cross-validation folds. The most optimal value is highlighted in bold and secondary optimal value is highlighted in underline. Optimal parameter value is selected according to the best performance on FSE and highlighted in bold.

| Model | clipRate | ACC($\uparrow$) | F1-macro($\uparrow$) | RMSE($\downarrow$) | PCC($\uparrow$) | FSE($\uparrow$) | FDS$_{norm}$($\downarrow$) |
|---|---|---|---|---|---|---|---|
| EEGNet | 0.0625 | **0.9325**$_{\pm0.0122}$ | **0.9325**$_{\pm0.0122}$ | **0.2417**$_{\pm0.0174}$ | **0.8788**$_{\pm0.0190}$ | 0.7121$_{\pm0.0382}$ | 0.0076$_{\pm0.0005}$ |
| | **0.125** | _0.9157_$_{\pm0.0155}$ | _0.9157_$_{\pm0.0155}$ | _0.2683_$_{\pm0.0181}$ | _0.8521_$_{\pm0.0235}$ | **0.7857**$_{\pm0.0259}$ | 0.0040$_{\pm0.0003}$ |
| | 0.25 | 0.9037$_{\pm0.0182}$ | 0.9036$_{\pm0.0182}$ | 0.3119$_{\pm0.0194}$ | 0.8088$_{\pm0.0314}$ | _0.7231_$_{\pm0.0294}$ | _0.0038_$_{\pm0.0002}$ |
| | 0.5 | 0.8524$_{\pm0.0352}$ | 0.8519$_{\pm0.0358}$ | 0.3882$_{\pm0.0202}$ | 0.6849$_{\pm0.0599}$ | 0.6054$_{\pm0.0445}$ | **0.0030**$_{\pm0.0004}$ |
| TSception | 0.0625 | **0.9883**$_{\pm0.0119}$ | **0.9883**$_{\pm0.0119}$ | **0.1471**$_{\pm0.0201}$ | **0.9566**$_{\pm0.0132}$ | 0.8366$_{\pm0.0307}$ | 0.0037$_{\pm0.0007}$ |
| | 0.125 | 0.9645$_{\pm0.0105}$ | 0.9645$_{\pm0.0105}$ | 0.1738$_{\pm0.0212}$ | 0.9374$_{\pm0.0153}$ | _0.8934_$_{\pm0.0235}$ | _0.0035_$_{\pm0.0004}$ |
| | **0.25** | _0.9778_$_{\pm0.0109}$ | _0.9778_$_{\pm0.0109}$ | _0.1525_$_{\pm0.0164}$ | _0.9528_$_{\pm0.0105}$ | **0.9281**$_{\pm0.0152}$ | **0.0023**$_{\pm0.0004}$ |
| | 0.5 | 0.8965$_{\pm0.0153}$ | 0.8961$_{\pm0.0149}$ | 0.3097$_{\pm0.0392}$ | 0.8016$_{\pm0.0285}$ | 0.7530$_{\pm0.0703}$ | 0.0047$_{\pm0.0010}$ |
| Deformer | 0.0625 | **0.9947**$_{\pm0.0024}$ | **0.9947**$_{\pm0.0024}$ | **0.1145**$_{\pm0.0161}$ | _0.9628_$_{\pm0.0056}$ | _0.9320_$_{\pm0.0200}$ | 0.0030$_{\pm0.0006}$ |
| | 0.125 | 0.9733$_{\pm0.0066}$ | 0.9733$_{\pm0.0066}$ | 0.1771$_{\pm0.0151}$ | 0.9421$_{\pm0.0100}$ | 0.8915$_{\pm0.0180}$ | 0.0035$_{\pm0.0003}$ |
| | **0.25** | _0.9853_$_{\pm0.0061}$ | _0.9853_$_{\pm0.0061}$ | _0.1255_$_{\pm0.0180}$ | **0.9688**$_{\pm0.0089}$ | **0.9324**$_{\pm0.0169}$ | _0.0027_$_{\pm0.0004}$ |
| | 0.5 | 0.8705$_{\pm0.0157}$ | 0.8701$_{\pm0.0153}$ | 0.3760$_{\pm0.0498}$ | 0.7215$_{\pm0.0276}$ | 0.6731$_{\pm0.0305}$ | **0.0024**$_{\pm0.0002}$ |

Table 6: Test results for sensitivity analysis on Mental Workload dataset with different clip sizes. Results show mean ± standard deviation over cross-validation folds. The most optimal value is highlighted in bold and secondary optimal value is highlighted in underline. Optimal parameter values selected based on the validation set are highlighted in bold.

| Model | clipRate | ACC($\uparrow$) | F1-macro($\uparrow$) | RMSE($\downarrow$) | PCC($\uparrow$) | FSE($\uparrow$) | FDS$_{norm}$($\downarrow$) |
|---|---|---|---|---|---|---|---|
| EEGNet | *BL* | 0.7111$_{\pm0.1612}$ | 0.6788$_{\pm0.1984}$ | 0.4774$_{\pm0.1679}$ | 0.4862$_{\pm0.3270}$ | 0.3964$_{\pm0.2278}$ | 1.0217$_{\pm0.4664}$ |
| | 0.0625 | _0.7272_$_{\pm0.1694}$ | _0.6813_$_{\pm0.2224}$ | _0.4281_$_{\pm0.1720}$ | **0.5911**$_{\pm0.3225}$ | _0.6086_$_{\pm0.1706}$ | 0.3867$_{\pm0.1693}$ |
| | **0.125** | **0.7348**$_{\pm0.1744}$ | **0.6871**$_{\pm0.2336}$ | **0.4248**$_{\pm0.1713}$ | _0.5865_$_{\pm0.3715}$ | **0.6111**$_{\pm0.1573}$ | 0.3536$_{\pm0.1377}$ |
| | 0.25 | 0.7104$_{\pm0.1854}$ | 0.6716$_{\pm0.2252}$ | 0.4319$_{\pm0.1306}$ | 0.5815$_{\pm0.3666}$ | 0.5568$_{\pm0.1652}$ | _0.3231_$_{\pm0.1120}$ |
| | 0.5 | 0.7137$_{\pm0.1757}$ | 0.6708$_{\pm0.2231}$ | 0.4496$_{\pm0.0912}$ | 0.5811$_{\pm0.3073}$ | 0.5188$_{\pm0.1099}$ | **0.2730**$_{\pm0.0791}$ |
| TSception | *BL* | 0.6798$_{\pm0.1640}$ | 0.6306$_{\pm0.2102}$ | 0.5121$_{\pm0.1762}$ | 0.4400$_{\pm0.3270}$ | 0.6071$_{\pm0.1804}$ | 0.3311$_{\pm0.1708}$ |
| | 0.0625 | **0.7578**$_{\pm0.1711}$ | **0.7268**$_{\pm0.2133}$ | **0.4052**$_{\pm0.1928}$ | **0.6110**$_{\pm0.3067}$ | **0.6911**$_{\pm0.1924}$ | 0.3127$_{\pm0.1547}$ |
| | 0.125 | 0.7358$_{\pm0.1644}$ | 0.7013$_{\pm0.2064}$ | 0.4326$_{\pm0.1821}$ | _0.5699_$_{\pm0.3214}$ | 0.6537$_{\pm0.1859}$ | 0.3216$_{\pm0.1478}$ |
| | **0.25** | _0.7390_$_{\pm0.1724}$ | _0.7106_$_{\pm0.2097}$ | _0.4295_$_{\pm0.1729}$ | 0.5598$_{\pm0.3725}$ | _0.6652_$_{\pm0.1943}$ | _0.2927_$_{\pm0.1130}$ |
| | 0.5 | 0.6846$_{\pm0.1735}$ | 0.6371$_{\pm0.2204}$ | 0.4626$_{\pm0.1575}$ | 0.5103$_{\pm0.3570}$ | 0.6326$_{\pm0.1783}$ | **0.2103**$_{\pm0.0832}$ |
| Deformer | *BL* | 0.7247$_{\pm0.1642}$ | 0.6923$_{\pm0.1993}$ | 0.4680$_{\pm0.1718}$ | 0.5101$_{\pm0.3194}$ | 0.4301$_{\pm0.2216}$ | 1.0448$_{\pm0.4584}$ |
| | 0.0625 | _0.7417_$_{\pm0.1573}$ | _0.7047_$_{\pm0.2025}$ | _0.4192_$_{\pm0.1653}$ | _0.6194_$_{\pm0.2623}$ | _0.6317_$_{\pm0.1839}$ | 0.3600$_{\pm0.1352}$ |
| | 0.125 | 0.7247$_{\pm0.1642}$ | 0.6850$_{\pm0.2137}$ | 0.4341$_{\pm0.1697}$ | 0.5587$_{\pm0.3364}$ | 0.6154$_{\pm0.1789}$ | 0.3632$_{\pm0.1473}$ |
| | **0.25** | **0.7645**$_{\pm0.1539}$ | **0.7326**$_{\pm0.2031}$ | **0.3983**$_{\pm0.1508}$ | **0.6438**$_{\pm0.3090}$ | **0.6529**$_{\pm0.1568}$ | _0.3075_$_{\pm0.1065}$ |
| | 0.5 | 0.7205$_{\pm0.1703}$ | 0.6918$_{\pm0.2010}$ | _0.4192_$_{\pm0.1182}$ | 0.6054$_{\pm0.3500}$ | 0.5897$_{\pm0.1614}$ | **0.2714**$_{\pm0.0669}$ |

Test results (Table 6) confirm these trends and demonstrate the robustness of the TRin framework with respect to temporal regularization. Notably, classification performance at the selected optimal clip sizes exceeds that at smaller clip sizes, which is plausible because temporally robust features help prevent overfitting and thus improve generalization. By contrast, very high validation accuracies observed without temporal regularization are likely indicative of overfitting.

### G.2 SENSITIVITY ANALYSIS ON TEMPORAL REGULARIZATION ALPHA

The sensitivity analysis of the maximum temporal regularization parameter $\alpha_s$ on the Mental Workload dataset is reported in Tables 7 and 8. Similar to clip-size selection, the optimal $\alpha_s$ is determined by FSE on validation set to balance classification accuracy and temporal stability. Based on validation results (Table 7), the optimal values are $\alpha_s = 20$ for EEGNet, $\alpha_s = 80$ for TSception, and $\alpha_s = 100$ for Deformer. The effect of $\alpha_s$ is analogous to that of clip size: increasing $\alpha_s$ strengthens temporal regularization but can eventually impair classification performance. For example, as reported in Table 8, raising $\alpha_s$ beyond 100 produces stronger temporal regularization, evidenced by decreasing FDS$_{norm}$, while degrading Deformer's classification accuracy. These findings suggest that excessively strong temporal regularization may prevent the model from making confident predictions, whereas a suitably chosen $\alpha_s$ promotes both classification accuracy and temporal stability.

Table 7: Validation results for sensitivity analysis on Mental Workload dataset with different temporal regularization $\alpha_s$ values. Results show mean ± standard deviation over cross-validation folds. The most optimal value is highlighted in bold and secondary optimal value is highlighted in underline. Optimal parameter value is selected according to the best performance on FSE and highlighted in bold.

| Model | $\alpha_s$ | ACC(↑) | F1-macro(↑) | RMSE(↓) | PCC(↑) | FSE(↑) | $FDS_{norm}$(↓) |
|---|---|---|---|---|---|---|---|
| EEGNet | *MSE* | $0.9050_{\pm0.0097}$ | $0.9050_{\pm0.0097}$ | $0.2804_{\pm0.0145}$ | $0.8329_{\pm0.0175}$ | $\underline{0.7916}_{\pm0.0206}$ | $0.0073_{\pm0.0006}$ |
| | 15 | $\mathbf{0.9299}_{\pm0.0151}$ | $\mathbf{0.9299}_{\pm0.0151}$ | $\underline{0.2683}_{\pm0.0181}$ | $\underline{0.8521}_{\pm0.0235}$ | $0.7857_{\pm0.0259}$ | $0.0044_{\pm0.0004}$ |
| | **20** | $\underline{0.9157}_{\pm0.0155}$ | $\underline{0.9157}_{\pm0.0155}$ | $\mathbf{0.2423}_{\pm0.0203}$ | $\mathbf{0.8778}_{\pm0.0222}$ | $\mathbf{0.8110}_{\pm0.0283}$ | $0.0040_{\pm0.0003}$ |
| | 25 | $0.9008_{\pm0.0173}$ | $0.9007_{\pm0.0174}$ | $0.2966_{\pm0.0193}$ | $0.8197_{\pm0.0292}$ | $0.7614_{\pm0.0271}$ | $0.0036_{\pm0.0003}$ |
| | 30 | $0.8858_{\pm0.0131}$ | $0.8857_{\pm0.0131}$ | $0.3261_{\pm0.0167}$ | $0.7799_{\pm0.0257}$ | $0.7280_{\pm0.0286}$ | $\underline{0.0031}_{\pm0.0002}$ |
| | 40 | $0.8701_{\pm0.0123}$ | $0.8698_{\pm0.0123}$ | $0.3654_{\pm0.0148}$ | $0.7215_{\pm0.0276}$ | $0.6731_{\pm0.0305}$ | $\mathbf{0.0024}_{\pm0.0002}$ |
| TSception | *MSE* | $0.9584_{\pm0.0091}$ | $0.9584_{\pm0.0091}$ | $0.1839_{\pm0.0196}$ | $0.9298_{\pm0.0154}$ | $\underline{0.8952}_{\pm0.0228}$ | $0.0043_{\pm0.0007}$ |
| | 40 | $\mathbf{0.9688}_{\pm0.0095}$ | $\mathbf{0.9687}_{\pm0.0095}$ | $\underline{0.1746}_{\pm0.0201}$ | $\underline{0.9338}_{\pm0.0137}$ | $0.8933_{\pm0.0213}$ | $0.0037_{\pm0.0005}$ |
| | **80** | $0.9645_{\pm0.0105}$ | $0.9645_{\pm0.0105}$ | $\mathbf{0.1738}_{\pm0.0212}$ | $\mathbf{0.9374}_{\pm0.0153}$ | $\mathbf{0.8964}_{\pm0.0235}$ | $0.0035_{\pm0.0004}$ |
| | 120 | $0.9637_{\pm0.0127}$ | $0.9637_{\pm0.0127}$ | $0.2553_{\pm0.0651}$ | $0.9239_{\pm0.0287}$ | $0.8186_{\pm0.0850}$ | $0.0028_{\pm0.0006}$ |
| | 160 | $0.9665_{\pm0.0106}$ | $0.9665_{\pm0.0106}$ | $0.4141_{\pm0.0073}$ | $0.9066_{\pm0.0129}$ | $0.5736_{\pm0.0185}$ | $\underline{0.0011}_{\pm0.0001}$ |
| | 200 | $\underline{0.9678}_{\pm0.0107}$ | $\underline{0.9677}_{\pm0.0108}$ | $0.4328_{\pm0.0053}$ | $0.9099_{\pm0.0137}$ | $0.5329_{\pm0.0145}$ | $\mathbf{0.0009}_{\pm0.0001}$ |
| Deformer | *MSE* | $0.8481_{\pm0.1162}$ | $0.8287_{\pm0.1692}$ | $0.3399_{\pm0.1302}$ | $0.7225_{\pm0.2414}$ | $0.7437_{\pm0.1054}$ | $0.0083_{\pm0.0039}$ |
| | 50 | $\underline{0.9718}_{\pm0.0073}$ | $\underline{0.9718}_{\pm0.0073}$ | $\mathbf{0.1768}_{\pm0.0154}$ | $\underline{0.9401}_{\pm0.0109}$ | $\underline{0.8911}_{\pm0.0160}$ | $0.0036_{\pm0.0004}$ |
| | **100** | $\mathbf{0.9733}_{\pm0.0066}$ | $\mathbf{0.9733}_{\pm0.0066}$ | $\underline{0.1771}_{\pm0.0151}$ | $\mathbf{0.9421}_{\pm0.0100}$ | $\mathbf{0.8915}_{\pm0.0180}$ | $0.0035_{\pm0.0003}$ |
| | 150 | $0.9669_{\pm0.0095}$ | $0.9669_{\pm0.0095}$ | $0.2025_{\pm0.0184}$ | $0.9266_{\pm0.0144}$ | $0.8682_{\pm0.0246}$ | $0.0036_{\pm0.0003}$ |
| | 200 | $0.9566_{\pm0.0122}$ | $0.9566_{\pm0.0122}$ | $0.2285_{\pm0.0173}$ | $0.9085_{\pm0.0165}$ | $0.8443_{\pm0.0249}$ | $\underline{0.0034}_{\pm0.0003}$ |
| | 300 | $0.9405_{\pm0.0152}$ | $0.9404_{\pm0.0153}$ | $0.2924_{\pm0.0171}$ | $0.8662_{\pm0.0203}$ | $0.7768_{\pm0.0268}$ | $\mathbf{0.0028}_{\pm0.0002}$ |

Table 8: Test results for sensitivity analysis on Mental Workload dataset with different temporal regularization $\alpha_s$ values. Results show mean ± standard deviation over cross-validation folds. The most optimal value is highlighted in bold and secondary optimal value is highlighted in underline. Optimal parameter values selected based on the validation set are highlighted in bold.

| Model | $\alpha_s$ | ACC(↑) | F1-macro(↑) | RMSE(↓) | PCC(↑) | FSE(↑) | $FDS_{norm}$(↓) |
|---|---|---|---|---|---|---|---|
| EEGNet | *BL* | $0.7111_{\pm0.1612}$ | $0.6788_{\pm0.1984}$ | $0.4774_{\pm0.1679}$ | $0.4862_{\pm0.3270}$ | $0.3964_{\pm0.2278}$ | $1.0126_{\pm0.4664}$ |
| | *MSE* | $0.6869_{\pm0.1255}$ | $0.6523_{\pm0.1613}$ | $0.5059_{\pm0.1219}$ | $0.4673_{\pm0.2822}$ | $0.3782_{\pm0.1578}$ | $1.0856_{\pm0.3412}$ |
| | 15 | $0.7163_{\pm0.1732}$ | $0.6749_{\pm0.2212}$ | $0.4386_{\pm0.1654}$ | $0.5371_{\pm0.3944}$ | $0.5671_{\pm0.1970}$ | $0.4537_{\pm0.1714}$ |
| | **20** | $\underline{0.7348}_{\pm0.1744}$ | $\underline{0.6871}_{\pm0.2336}$ | $\underline{0.4248}_{\pm0.1713}$ | $\underline{0.5865}_{\pm0.3715}$ | $0.6111_{\pm0.1573}$ | $0.3536_{\pm0.1377}$ |
| | 25 | $0.7315_{\pm0.1951}$ | $0.6838_{\pm0.2492}$ | $\mathbf{0.4228}_{\pm0.1777}$ | $0.5470_{\pm0.4418}$ | $\mathbf{0.6201}_{\pm0.1836}$ | $0.2664_{\pm0.1033}$ |
| | 30 | $\mathbf{0.7384}_{\pm0.1899}$ | $\mathbf{0.6878}_{\pm0.2499}$ | $0.4261_{\pm0.1595}$ | $0.5622_{\pm0.4337}$ | $\underline{0.6141}_{\pm0.1586}$ | $\underline{0.2217}_{\pm0.0796}$ |
| | 40 | $0.7245_{\pm0.1880}$ | $0.6652_{\pm0.2542}$ | $0.4389_{\pm0.1303}$ | $\mathbf{0.5951}_{\pm0.4274}$ | $0.6037_{\pm0.1375}$ | $\mathbf{0.1564}_{\pm0.0467}$ |
| TSception | *BL* | $0.6798_{\pm0.1640}$ | $0.6306_{\pm0.2102}$ | $0.5121_{\pm0.1762}$ | $0.4400_{\pm0.3270}$ | $0.6071_{\pm0.1804}$ | $0.3311_{\pm0.1708}$ |
| | *MSE* | $0.6695_{\pm0.1554}$ | $0.6248_{\pm0.1973}$ | $0.5111_{\pm0.1606}$ | $0.4283_{\pm0.3272}$ | $0.5697_{\pm0.1842}$ | $0.3878_{\pm0.1669}$ |
| | 40 | $0.7206_{\pm0.1684}$ | $0.6796_{\pm0.2190}$ | $0.4498_{\pm0.1824}$ | $0.5375_{\pm0.3427}$ | $\underline{0.6574}_{\pm0.1812}$ | $0.3003_{\pm0.1355}$ |
| | **80** | $\mathbf{0.7390}_{\pm0.1724}$ | $\mathbf{0.7106}_{\pm0.2097}$ | $\mathbf{0.4295}_{\pm0.1729}$ | $0.5598_{\pm0.3725}$ | $\mathbf{0.6652}_{\pm0.1943}$ | $0.2927_{\pm0.1130}$ |
| | 120 | $\underline{0.7271}_{\pm0.1758}$ | $\underline{0.6892}_{\pm0.2209}$ | $\underline{0.4459}_{\pm0.1369}$ | $0.5662_{\pm0.3446}$ | $0.6322_{\pm0.1538}$ | $0.1759_{\pm0.0903}$ |
| | 160 | $0.7131_{\pm0.1681}$ | $0.6695_{\pm0.2162}$ | $0.4639_{\pm0.0355}$ | $\mathbf{0.6103}_{\pm0.3342}$ | $0.5540_{\pm0.0623}$ | $\underline{0.0459}_{\pm0.0087}$ |
| | 200 | $0.7157_{\pm0.1723}$ | $0.6715_{\pm0.2217}$ | $0.4712_{\pm0.0286}$ | $\underline{0.5832}_{\pm0.3758}$ | $0.5415_{\pm0.0522}$ | $\mathbf{0.0357}_{\pm0.0059}$ |
| Deformer | *BL* | $0.7247_{\pm0.1602}$ | $0.6923_{\pm0.1993}$ | $0.4680_{\pm0.1718}$ | $0.5101_{\pm0.3194}$ | $0.4301_{\pm0.2216}$ | $1.0448_{\pm0.4584}$ |
| | *MSE* | $0.6745_{\pm0.1486}$ | $0.6320_{\pm0.1923}$ | $0.5115_{\pm0.1535}$ | $0.4246_{\pm0.3124}$ | $0.3748_{\pm0.2044}$ | $1.0474_{\pm0.5243}$ |
| | 50 | $0.7287_{\pm0.1631}$ | $0.6894_{\pm0.2128}$ | $0.4288_{\pm0.1627}$ | $0.5869_{\pm0.3279}$ | $0.6069_{\pm0.1787}$ | $0.3700_{\pm0.1355}$ |
| | **100** | $\mathbf{0.7645}_{\pm0.1539}$ | $\mathbf{0.7326}_{\pm0.2031}$ | $\mathbf{0.3983}_{\pm0.1508}$ | $\underline{0.6438}_{\pm0.3090}$ | $\mathbf{0.6529}_{\pm0.1568}$ | $0.3075_{\pm0.1065}$ |
| | 150 | $0.7377_{\pm0.1590}$ | $0.6985_{\pm0.2083}$ | $0.4147_{\pm0.1354}$ | $0.6394_{\pm0.2951}$ | $0.6381_{\pm0.1441}$ | $0.2730_{\pm0.0702}$ |
| | 200 | $\underline{0.7535}_{\pm0.1676}$ | $\underline{0.7169}_{\pm0.2148}$ | $\underline{0.4034}_{\pm0.1328}$ | $\mathbf{0.6547}_{\pm0.3111}$ | $\underline{0.6395}_{\pm0.1494}$ | $\underline{0.2493}_{\pm0.0671}$ |
| | 300 | $0.7471_{\pm0.1659}$ | $0.7080_{\pm0.2146}$ | $0.4186_{\pm0.1033}$ | $0.6427_{\pm0.3373}$ | $0.6173_{\pm0.1182}$ | $\mathbf{0.1766}_{\pm0.0362}$ |

# H ABLATION STUDY

## H.1 ABLATION STUDY ON DYNAMIC TRAINING STRATEGY

The ablation study of the dynamic training strategy across all models on the Mental Workload dataset (Table 9) demonstrates the consistent effectiveness of curriculum learning as a component of the TRin framework. In the table, w/o dynamic training and full TRin denote the static and dynamic training strategies, respectively. Static training imposes a strong regularization barrier: it yields the lowest $FDS_{norm}$ but degrades classification performance, in Deformer's case performing worse

Table 9: Ablation study of TRin dynamic training strategy on Mental Workload dataset across models. Results show mean ± standard deviation over cross-validation folds. The most optimal value is highlighted in bold and secondary optimal value is highlighted in underline.

| Model | Configuration | ACC(↑) | F1-macro(↑) | RMSE(↓) | PCC(↑) | FSE(↑) | FDS$_{norm}$(↓) |
|---|---|---|---|---|---|---|---|
| EEGNet | Baseline | $0.7111_{\pm0.1612}$ | $0.6788_{\pm0.1984}$ | $0.4774_{\pm0.1679}$ | $0.4862_{\pm0.3270}$ | $0.3964_{\pm0.2278}$ | $1.0126_{\pm0.4664}$ |
| | w/o Dynamic | $0.7010_{\pm0.1724}$ | $0.6476_{\pm0.2286}$ | $0.4473_{\pm0.1389}$ | $0.5623_{\pm0.3505}$ | $0.5580_{\pm0.1419}$ | $\mathbf{0.3052}_{\pm0.1139}$ |
| | Full TRin | $\mathbf{0.7348}_{\pm0.1744}$ | $\mathbf{0.6871}_{\pm0.2336}$ | $\mathbf{0.4248}_{\pm0.1713}$ | $\mathbf{0.5865}_{\pm0.3715}$ | $\mathbf{0.6111}_{\pm0.1573}$ | $\underline{0.3536}_{\pm0.1377}$ |
| TSception | Baseline | $0.6798_{\pm0.1640}$ | $0.6306_{\pm0.2102}$ | $0.5121_{\pm0.1762}$ | $0.4400_{\pm0.3270}$ | $0.6071_{\pm0.1804}$ | $0.3311_{\pm0.1708}$ |
| | w/o Dynamic | $\underline{0.7046}_{\pm0.1632}$ | $\underline{0.6595}_{\pm0.2135}$ | $\underline{0.4514}_{\pm0.1491}$ | $\underline{0.5530}_{\pm0.3118}$ | $\underline{0.6466}_{\pm0.1572}$ | $\mathbf{0.2077}_{\pm0.0741}$ |
| | Full TRin | $\mathbf{0.7390}_{\pm0.1724}$ | $\mathbf{0.7106}_{\pm0.2097}$ | $\mathbf{0.4295}_{\pm0.1729}$ | $\mathbf{0.5598}_{\pm0.3725}$ | $\mathbf{0.6652}_{\pm0.1943}$ | $\underline{0.2927}_{\pm0.1130}$ |
| Deformer | Baseline | $\underline{0.7247}_{\pm0.1602}$ | $\underline{0.6923}_{\pm0.1993}$ | $0.4680_{\pm0.1718}$ | $0.5101_{\pm0.3194}$ | $0.4301_{\pm0.2216}$ | $1.0448_{\pm0.4584}$ |
| | w/o Dynamic | $0.6893_{\pm0.1598}$ | $0.6419_{\pm0.2040}$ | $\underline{0.4485}_{\pm0.1113}$ | $\underline{0.5951}_{\pm0.2882}$ | $\underline{0.5776}_{\pm0.1375}$ | $\mathbf{0.2516}_{\pm0.0680}$ |
| | Full TRin | $\mathbf{0.7645}_{\pm0.1539}$ | $\mathbf{0.7326}_{\pm0.2031}$ | $\mathbf{0.3983}_{\pm0.1508}$ | $\mathbf{0.6438}_{\pm0.3090}$ | $\mathbf{0.6529}_{\pm0.1568}$ | $\underline{0.3075}_{\pm0.1065}$ |

Table 10: Ablation study of dynamic training strategy on MSE loss on SEED dataset across models. Results show mean ± standard deviation over cross-validation folds. The most optimal value is highlighted in bold and secondary optimal value is highlighted in underline.

| Model | Configuration | ACC(↑) | F1-macro(↑) | RMSE(↓) | PCC(↑) | FSE(↑) | FDS$_{norm}$(↓) |
|---|---|---|---|---|---|---|---|
| EEGNet | BL-CE | $0.7090_{\pm0.1565}$ | $\mathbf{0.6829}_{\pm0.1801}$ | $0.4849_{\pm0.1648}$ | $0.5099_{\pm0.3119}$ | $0.6216_{\pm0.2292}$ | $0.2228_{\pm0.2448}$ |
| | BL-MSE | $0.6975_{\pm0.1609}$ | $0.6629_{\pm0.1925}$ | $0.4936_{\pm0.1783}$ | $0.4813_{\pm0.3286}$ | $0.6148_{\pm0.2244}$ | $0.2119_{\pm0.2717}$ |
| | w/o Dynamic | $\underline{0.7120}_{\pm0.1454}$ | $\underline{0.6748}_{\pm0.1882}$ | $\underline{0.4634}_{\pm0.1528}$ | $\mathbf{0.5314}_{\pm0.2968}$ | $\underline{0.6521}_{\pm0.1524}$ | $\underline{0.1157}_{\pm0.0511}$ |
| | Dynamic | $\mathbf{0.7132}_{\pm0.1679}$ | $0.6659_{\pm0.2195}$ | $\mathbf{0.4627}_{\pm0.1656}$ | $\underline{0.5271}_{\pm0.3346}$ | $\mathbf{0.6636}_{\pm0.1607}$ | $\mathbf{0.0987}_{\pm0.0345}$ |
| TSception | BL-CE | $0.6774_{\pm0.1567}$ | $0.6209_{\pm0.2051}$ | $\underline{0.5280}_{\pm0.1711}$ | $0.4324_{\pm0.2948}$ | $\underline{0.6152}_{\pm0.1686}$ | $0.1713_{\pm0.0845}$ |
| | BL-MSE | $\underline{0.6790}_{\pm0.1491}$ | $\mathbf{0.6338}_{\pm0.1850}$ | $0.5357_{\pm0.1514}$ | $\underline{0.4344}_{\pm0.2943}$ | $0.6118_{\pm0.1584}$ | $0.1972_{\pm0.0778}$ |
| | w/o Dynamic | $0.6682_{\pm0.1476}$ | $0.6131_{\pm0.1889}$ | $0.5402_{\pm0.1579}$ | $0.4244_{\pm0.2813}$ | $0.6019_{\pm0.1684}$ | $\mathbf{0.1646}_{\pm0.0712}$ |
| | Dynamic | $\mathbf{0.6851}_{\pm0.1454}$ | $\underline{0.6337}_{\pm0.1913}$ | $\mathbf{0.5228}_{\pm0.1557}$ | $\mathbf{0.4580}_{\pm0.2757}$ | $\mathbf{0.6231}_{\pm0.1604}$ | $\underline{0.1675}_{\pm0.0733}$ |
| Deformer | BL-CE | $0.7242_{\pm0.1401}$ | $0.6888_{\pm0.1850}$ | $0.4813_{\pm0.1535}$ | $0.5113_{\pm0.2824}$ | $0.4538_{\pm0.1908}$ | $0.7936_{\pm0.5157}$ |
| | BL-MSE | $0.7039_{\pm0.1563}$ | $0.6632_{\pm0.1993}$ | $0.4941_{\pm0.1626}$ | $0.4788_{\pm0.3064}$ | $0.4676_{\pm0.2237}$ | $0.7207_{\pm0.4617}$ |
| | w/o Dynamic | $\underline{0.7324}_{\pm0.1669}$ | $\underline{0.6924}_{\pm0.2152}$ | $\underline{0.4543}_{\pm0.1786}$ | $\underline{0.5327}_{\pm0.3472}$ | $\mathbf{0.5478}_{\pm0.1948}$ | $\mathbf{0.5012}_{\pm0.3362}$ |
| | Dynamic | $\mathbf{0.7416}_{\pm0.1536}$ | $\mathbf{0.7069}_{\pm0.2013}$ | $\mathbf{0.4458}_{\pm0.1749}$ | $\mathbf{0.5493}_{\pm0.3117}$ | $\underline{0.5450}_{\pm0.2469}$ | $\underline{0.5274}_{\pm0.4618}$ |

than the baseline. By contrast, dynamic training attains higher FDS$_{norm}$ than static training but consistently produces incremental improvements in classification metrics and FSE across all models. These results indicate that progressive temporal regularization helps models overcome the static regularization barrier and better balance temporal stability with classification accuracy.

The clip based shuffling strategy likewise reduces temporal instability for conventional regularizers such as MSE loss (Table 10) on the SEED dataset. In the table, the baseline cross entropy and MSE implementations use complete random shuffling of segments and are denoted by BL-CE and BL-MSE, respectively. The entries labeled w/o Dynamic and Dynamic report MSE combined with the TRin data treatment under static and dynamic training regimes, respectively. Combining MSE loss with either the static or dynamic TRin data treatment improves temporal stability. These results indicate that our sequence-preserving data treatment strategies are also effective stabilization for conventional objectives such as MSE in emotion recognition tasks.

### H.2 ABLATION STUDY WITH TEMPORAL LOSS ONLY

In this section, we evaluate the effectiveness of the temporal regularization loss function alone, without the data treatment strategy in TRin framework. We compare the performance of the temporal regularization loss function with and without TRin data treatment strategy.

Since the temporal regularization loss function can only be applied to a series of predictions with temporal order. To compare the effectiveness of TRin data treatment strategy, we adapt the training strategy from (Dileo et al., 2023) to apply the temporal regularization loss function to the predictions of the model directly. This training strategy is to ensure each batch only contains data from adjacent time stamps. We denote the temporal loss function alone as BL-TL. Results are shown in Table 11.

The results show that the temporal regularization loss function alone is effective in improving temporal stability (still less effective than the combined TRin framework). However, its classification performance is significantly worse than the baseline, nearly as bad as the random guessing. The

Table 11: Ablation study of temporal regularization loss function alone on Mental Workload dataset across models. Results show mean ± standard deviation over cross-validation folds. The most optimal value is highlighted in bold and secondary optimal value is highlighted in underline.

| Model | Config | ACC($\uparrow$) | F1-macro($\uparrow$) | RMSE($\downarrow$) | PCC($\uparrow$) | FSE($\uparrow$) | FDS$_{norm}$($\downarrow$) |
|---|---|---|---|---|---|---|---|
| | BL-CE | $\underline{0.7111}_{\pm 0.1612}$ | $\underline{0.6788}_{\pm 0.1984}$ | $\underline{0.4774}_{\pm 0.1679}$ | $\underline{0.4862}_{\pm 0.3270}$ | $0.3964_{\pm 0.2278}$ | $1.0126_{\pm 0.4664}$ |
| EEGNet | BL-TL | $0.4988_{\pm 0.0278}$ | $0.4894_{\pm 0.0324}$ | $0.5297_{\pm 0.0415}$ | $0.0027_{\pm 0.0800}$ | $0.1576_{\pm 0.1044}$ | $\underline{0.8915}_{\pm 0.6609}$ |
| | TRin | $\mathbf{0.7348}_{\pm 0.1744}$ | $\mathbf{0.6871}_{\pm 0.2336}$ | $\mathbf{0.4248}_{\pm 0.1713}$ | $\mathbf{0.5865}_{\pm 0.3715}$ | $\mathbf{0.6111}_{\pm 0.1573}$ | $\mathbf{0.3536}_{\pm 0.1377}$ |
| | BL-CE | $\underline{0.6798}_{\pm 0.1640}$ | $\underline{0.6306}_{\pm 0.2102}$ | $\underline{0.5121}_{\pm 0.1762}$ | $\underline{0.4400}_{\pm 0.3270}$ | $0.6071_{\pm 0.1804}$ | $0.3311_{\pm 0.1708}$ |
| TSception | BL-TL | $0.5356_{\pm 0.1133}$ | $0.4897_{\pm 0.1327}$ | $0.5596_{\pm 0.1131}$ | $0.1580_{\pm 0.3268}$ | $0.4128_{\pm 0.1621}$ | $\underline{0.3293}_{\pm 0.2863}$ |
| | TRin | $\mathbf{0.7390}_{\pm 0.1724}$ | $\mathbf{0.7106}_{\pm 0.2097}$ | $\mathbf{0.4295}_{\pm 0.1729}$ | $\mathbf{0.5598}_{\pm 0.3725}$ | $\mathbf{0.6652}_{\pm 0.1943}$ | $\mathbf{0.2927}_{\pm 0.1130}$ |
| | BL-CE | $\underline{0.7247}_{\pm 0.1602}$ | $\underline{0.6923}_{\pm 0.1993}$ | $\underline{0.4680}_{\pm 0.1718}$ | $\underline{0.5101}_{\pm 0.3194}$ | $0.4301_{\pm 0.2216}$ | $1.0448_{\pm 0.4584}$ |
| Deformer | BL-TL | $0.5024_{\pm 0.1472}$ | $0.4482_{\pm 0.1577}$ | $0.5154_{\pm 0.0496}$ | $0.0237_{\pm 0.4094}$ | $0.3898_{\pm 0.1066}$ | $\underline{0.3256}_{\pm 0.1161}$ |
| | TRin | $\mathbf{0.7645}_{\pm 0.1539}$ | $\mathbf{0.7326}_{\pm 0.2031}$ | $\mathbf{0.3983}_{\pm 0.1508}$ | $\mathbf{0.6438}_{\pm 0.3090}$ | $\mathbf{0.6529}_{\pm 0.1568}$ | $\mathbf{0.3075}_{\pm 0.1065}$ |

reason direct batch-wise training strategy performs poorly is because EEG data is significantly different from other modalities. Specifically, the amount and variety of EEG data is much less than other modalities, sometimes only one batch is sampled from a trial. Therefore, if a batch of training data is not shuffled, models tend to fit with the whole sequence of data instead of learning meaningful temporal patterns. This result highlights the importance of the TRin data treatment strategy using clip-wise shuffling.

# I LATENCY ANALYSIS

In this section, we simulate the latency of the model by testing model behavior when the ground truth changes. Current BCI datasets mainly rely on self-reported ground truth labels, while lack reliable methods of generating fine-grained ground truth labels. Moreover, abrupt mental state changes is rare under well established mental stimuli such as watching videos. Therefore, we have to simulate the change in ground truth by concatenating the trials from the same subject but with different ground truth labels. Donate the change point of ground truth label as the reference $t_c$, ground truth label at time $t$ as $y(t)$, and model prediction at time $t$ as $\hat{y}(t)$. We quantify the latency as the shortest time difference between a correct prediction after $t_c$ and $t_c$. We set a constant positive $T$ serving as a threshold. If there is no such correct prediction after $T$ seconds, it may indicate that the other prediction changes are not related to change in ground truth. In this case, we mark the transition as unsuccessful transition (UT). The definition of the latency $Lt$ is shown in Equation 24.

$$Lt = \min_t(t - t_c) \text{ s.t. } \hat{y}(t) = y(t) \quad \text{where } 0 \le t - t_c \le T \tag{24}$$

In our implementation, we set $T = 4$ seconds equals to the segment length. The results using Mental Workload dataset are shown in Table 12. Lt refers to the average latency and UT refers to the number of unsuccessful transitions. SW refers to moving average of posteriors with window size 0.6s in order to achieve similar temporal stability performance as original TRin. Since post moving average is applicable to both baseline and TRin, we also implement a post sliding window version for TRin, named as TRinS.

The result shows that TRin framework is the only one that can consistently improve the classification performance across all models. Post hoc sliding window size is a chosen to achieve comparable performance to TRin framework in terms of temporal stability, but the result still attains ignorable improvement in terms of classification accuracy.

In terms of latency, TRin framework achieves comparable performance to baseline especially for EEGNet and Deformer. In case of TSception and CBraMod, TRin framework has higher latency but less unsuccessfull transitions. In this case, note that both baselines of TSception and CBraMod have inferior performance on accuracy, it is plausible that low baseline latency might be due to false prediction fluctuation rather than distiguishing a real ground truth change.

Besides, the result shows that post hoc sliding window approach introduces additional latency than TRin framework. This is because the nature of post hoc sliding window approach is totally different from TRin framework. Post hoc sliding window approach relies on previous prediction to make current prediction during inference, while TRin framework is only applied during the model training

Table 12: Latency analysis with change in ground truth on Mental Workload dataset. Results show mean ± standard deviation over cross-validation folds. The best and performance is highlighted in bold while the second best performance is highlighted in underline.

| Model | ACC(↑) | F1-macro(↑) | RMSE(↓) | FSE(↑) | HFSE(↓) | FDS$_{norm}$(↓) | Lt(↓) | UT(↓) |
|---|---|---|---|---|---|---|---|---|
| EEGNet | $0.7091_{\pm0.1584}$ | $0.6773_{\pm0.1951}$ | $0.4811_{\pm0.1625}$ | $0.1837_{\pm0.2179}$ | $2.4239_{\pm0.8844}$ | $2.3286_{\pm0.8305}$ | $\underline{0.8457}_{\pm0.6848}$ | $\mathbf{0.1111}_{\pm0.3143}$ |
| + *SW* | $0.7167_{\pm0.1638}$ | $0.6834_{\pm0.2026}$ | $0.4555_{\pm0.1685}$ | $0.6147_{\pm0.2553}$ | $1.0076_{\pm0.2252}$ | $0.9627_{\pm0.2695}$ | $1.0055_{\pm0.8142}$ | $0.1944_{\pm0.3958}$ |
| + **TRin** | $\underline{0.7327}_{\pm0.1701}$ | $\underline{0.6866}_{\pm0.2270}$ | $\underline{0.4270}_{\pm0.1663}$ | $\underline{0.6257}_{\pm0.2596}$ | $\underline{0.9739}_{\pm0.1439}$ | $\underline{0.8757}_{\pm0.2165}$ | $\mathbf{0.7736}_{\pm0.6787}$ | $\mathbf{0.1111}_{\pm0.3143}$ |
| + **TRin***S* | $\mathbf{0.7369}_{\pm0.1747}$ | $\mathbf{0.6897}_{\pm0.2329}$ | $\mathbf{0.4232}_{\pm0.1683}$ | $\mathbf{0.7158}_{\pm0.2482}$ | $\mathbf{0.6831}_{\pm0.0617}$ | $\mathbf{0.5157}_{\pm0.0775}$ | $0.9204_{\pm0.8783}$ | $\underline{0.1389}_{\pm0.3458}$ |
| TSception | $0.6771_{\pm0.1607}$ | $0.6288_{\pm0.2058}$ | $0.5166_{\pm0.1680}$ | $0.5545_{\pm0.3682}$ | $0.9376_{\pm0.1940}$ | $0.8842_{\pm0.2638}$ | $\mathbf{0.6455}_{\pm0.8027}$ | $\underline{0.1667}_{\pm0.3727}$ |
| + *SW* | $0.6777_{\pm0.1612}$ | $0.6293_{\pm0.2062}$ | $0.5117_{\pm0.1708}$ | $0.6116_{\pm0.2623}$ | $\underline{0.7391}_{\pm0.0796}$ | $\underline{0.6538}_{\pm0.1478}$ | $0.8084_{\pm0.8415}$ | $0.1667_{\pm0.3727}$ |
| + **TRin** | $\underline{0.7367}_{\pm0.1698}$ | $\mathbf{0.7084}_{\pm0.2071}$ | $\underline{0.4327}_{\pm0.1700}$ | $\underline{0.6897}_{\pm0.2177}$ | $0.8531_{\pm0.1125}$ | $0.7815_{\pm0.1643}$ | $\underline{0.7596}_{\pm0.7819}$ | $\mathbf{0.1389}_{\pm0.3458}$ |
| + **TRin***S* | $\underline{0.7369}_{\pm0.1747}$ | $\underline{0.6897}_{\pm0.2329}$ | $\mathbf{0.4232}_{\pm0.1683}$ | $\mathbf{0.7158}_{\pm0.2482}$ | $\mathbf{0.6831}_{\pm0.0617}$ | $\mathbf{0.5157}_{\pm0.0775}$ | $0.9187_{\pm0.9565}$ | $0.1389_{\pm0.3458}$ |
| Deformer | $0.7228_{\pm0.1572}$ | $0.6910_{\pm0.1956}$ | $0.4721_{\pm0.1663}$ | $0.1994_{\pm0.2123}$ | $2.4212_{\pm0.8142}$ | $2.3758_{\pm0.8505}$ | $\underline{0.7736}_{\pm0.6193}$ | $\mathbf{0.1111}_{\pm0.3143}$ |
| + *SW* | $0.7299_{\pm0.1633}$ | $0.6967_{\pm0.2037}$ | $0.4438_{\pm0.1728}$ | $0.6263_{\pm0.2682}$ | $0.9803_{\pm0.2115}$ | $0.9633_{\pm0.2702}$ | $0.9441_{\pm0.7902}$ | $0.1111_{\pm0.3143}$ |
| + **TRin** | $\underline{0.7620}_{\pm0.1504}$ | $\underline{0.7311}_{\pm0.1978}$ | $\underline{0.4018}_{\pm0.1460}$ | $\underline{0.6948}_{\pm0.2231}$ | $\underline{0.9164}_{\pm0.1500}$ | $\underline{0.7961}_{\pm0.1737}$ | $\mathbf{0.7374}_{\pm0.8119}$ | $\mathbf{0.0833}_{\pm0.2764}$ |
| + **TRin***S* | $\mathbf{0.7626}_{\pm0.1519}$ | $\mathbf{0.7315}_{\pm0.1993}$ | $\mathbf{0.3984}_{\pm0.1474}$ | $\mathbf{0.7652}_{\pm0.2191}$ | $\mathbf{0.6683}_{\pm0.0840}$ | $\mathbf{0.5065}_{\pm0.0749}$ | $0.9047_{\pm0.9916}$ | $\underline{0.1111}_{\pm0.3143}$ |
| CBraMod | $0.6730_{\pm0.1592}$ | $0.6214_{\pm0.2087}$ | $0.5329_{\pm0.1748}$ | $0.0609_{\pm0.1803}$ | $3.5570_{\pm1.2185}$ | $3.0957_{\pm1.2356}$ | $\mathbf{0.6158}_{\pm0.6967}$ | $\underline{0.1389}_{\pm0.3458}$ |
| + *SW* | $0.6795_{\pm0.1668}$ | $0.6232_{\pm0.2200}$ | $0.4948_{\pm0.1870}$ | $0.3837_{\pm0.3400}$ | $1.1396_{\pm0.3929}$ | $1.1396_{\pm0.3929}$ | $0.8182_{\pm0.8824}$ | $0.1667_{\pm0.3727}$ |
| + **TRin** | $\underline{0.7096}_{\pm0.1667}$ | $\underline{0.6602}_{\pm0.2148}$ | $\underline{0.4441}_{\pm0.1292}$ | $\underline{0.5256}_{\pm0.2589}$ | $\underline{1.2520}_{\pm0.3902}$ | $\underline{1.0022}_{\pm0.3408}$ | $\underline{0.7605}_{\pm0.7012}$ | $\mathbf{0.1111}_{\pm0.3143}$ |
| + **TRin***S* | $\mathbf{0.7149}_{\pm0.1681}$ | $\mathbf{0.6646}_{\pm0.2239}$ | $\mathbf{0.4403}_{\pm0.1305}$ | $\mathbf{0.7098}_{\pm0.2152}$ | $\mathbf{0.9745}_{\pm0.9905}$ | $\mathbf{0.5317}_{\pm0.0900}$ | $0.9745_{\pm0.9905}$ | $0.1389_{\pm0.3458}$ |

phase and is independent of previous prediction during inference. The visualization in Figure 5 shows the actual example of latency comparison on the Mental Workload dataset using EEGNet.

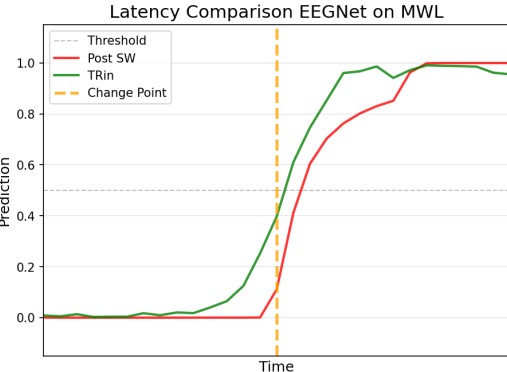

Figure 5: Example visualization of latency comparison on the Mental Workload dataset using EEG-Net zoom into 6 seconds. Baseline plus post hoc sliding window approach is denoted as Post SW, TRin framework is denoted as TRin. Change in ground truth is denoted as Change Point.

## J INFERENCE OVERLAP ANALYSIS

During real-time applications such as neurofeedback, the feedback rate is crucial for the effectiveness of the system, typically less than 200ms delay is considered as plausible (Teo et al., 2021; Jochumsen et al., 2019). To make sure our quantitative evaluation on TRin framework aligns with real-time scenarios, we keep the overlap rate of 95% (i.e. 200ms delay between two consecutive predictions). In this case, it is plausible to question the effectiveness of TRin on lower feedback rate. Therefore, this section analyzes performance with different overlap rate. The results are shown in Table 13. Since this analysis include different feedback rate which FDS is sensitive to, we exclude the FDS results between different overlap rate to avoid bias.

It is important to note that both the baseline models and our TRin framework rely solely on the current input segment to generate predictions. Consequently, using a lower overlap rate is equivalent to post-process the predictions with downsampling. This, in turn, introduces a strict temporal delay. For example, in our experiments, overlap rates of 95%, 75%, and 50% correspond to delays 200ms, 1s, and 2s between consecutive predictions, respectively. Therefore, any observed performance improvement at lower overlap rates should not be interpreted as enhanced real-time effectiveness. Instead, these results serve as a sanity check to verify that our TRin framework consistently improves performance across different overlap conditions.

Table 13: Performance comparison on Mental Workload dataset with different overlap rate. Results show mean ± standard deviation over cross-validation folds. The best and performance is highlighted in bold.

| Overlap | Model | ACC($\uparrow$) | F1-macro($\uparrow$) | RMSE($\downarrow$) | PCC($\uparrow$) | FSE($\uparrow$) | HFSE($\downarrow$) |
|---|---|---|---|---|---|---|---|
| 0.95% | EEGNet | $0.7111_{\pm0.1612}$ | $0.6788_{\pm0.1984}$ | $0.4774_{\pm0.1679}$ | $0.4862_{\pm0.3270}$ | $0.3964_{\pm0.2278}$ | $1.0126_{\pm0.4664}$ |
| | + TRin | $\mathbf{0.7348}_{\pm0.1744}$ | $\mathbf{0.6871}_{\pm0.2336}$ | $\mathbf{0.4248}_{\pm0.1713}$ | $\mathbf{0.5865}_{\pm0.3715}$ | $\mathbf{0.6111}_{\pm0.1573}$ | $\mathbf{0.3536}_{\pm0.1377}$ |
| | TSception | $0.6798_{\pm0.1640}$ | $0.6306_{\pm0.2102}$ | $0.5121_{\pm0.1762}$ | $0.4400_{\pm0.3270}$ | $0.6071_{\pm0.1804}$ | $0.3679_{\pm0.2098}$ |
| | + TRin | $\mathbf{0.7390}_{\pm0.1724}$ | $\mathbf{0.7106}_{\pm0.2097}$ | $\mathbf{0.4295}_{\pm0.1729}$ | $\mathbf{0.5598}_{\pm0.3725}$ | $\mathbf{0.6652}_{\pm0.1943}$ | $\mathbf{0.3532}_{\pm0.1484}$ |
| | Deformer | $0.7247_{\pm0.1602}$ | $0.6923_{\pm0.1993}$ | $0.4680_{\pm0.1718}$ | $0.5101_{\pm0.3194}$ | $0.4301_{\pm0.2216}$ | $1.4305_{\pm0.5940}$ |
| | + TRin | $\mathbf{0.7645}_{\pm0.1539}$ | $\mathbf{0.7326}_{\pm0.2031}$ | $\mathbf{0.3983}_{\pm0.1508}$ | $\mathbf{0.6438}_{\pm0.3090}$ | $\mathbf{0.6529}_{\pm0.1568}$ | $\mathbf{0.4064}_{\pm0.1415}$ |
| 0.75% | EEGNet | $0.7063_{\pm0.1630}$ | $0.6740_{\pm0.1995}$ | $0.4801_{\pm0.1668}$ | $0.4903_{\pm0.3221}$ | $0.5033_{\pm0.2301}$ | $0.9360_{\pm0.5120}$ |
| | + TRin | $\mathbf{0.7359}_{\pm0.1767}$ | $\mathbf{0.6885}_{\pm0.2353}$ | $\mathbf{0.4257}_{\pm0.1701}$ | $\mathbf{0.5922}_{\pm0.3672}$ | $\mathbf{0.6420}_{\pm0.1598}$ | $\mathbf{0.3476}_{\pm0.1557}$ |
| | TSception | $0.6793_{\pm0.1620}$ | $0.6297_{\pm0.2092}$ | $0.5108_{\pm0.1774}$ | $0.4439_{\pm0.3235}$ | $0.5567_{\pm0.1823}$ | $0.5689_{\pm0.3380}$ |
| | + TRin | $\mathbf{0.7403}_{\pm0.1729}$ | $\mathbf{0.7117}_{\pm0.2110}$ | $\mathbf{0.4298}_{\pm0.1721}$ | $\mathbf{0.5587}_{\pm0.3749}$ | $\mathbf{0.6215}_{\pm0.1965}$ | $\mathbf{0.5046}_{\pm0.2363}$ |
| | Deformer | $0.7165_{\pm0.1627}$ | $0.6829_{\pm0.2013}$ | $0.4725_{\pm0.1770}$ | $0.4959_{\pm0.3272}$ | $0.4918_{\pm0.2275}$ | $1.0457_{\pm0.4888}$ |
| | + TRin | $\mathbf{0.7614}_{\pm0.1525}$ | $\mathbf{0.7301}_{\pm0.2000}$ | $\mathbf{0.4009}_{\pm0.1498}$ | $\mathbf{0.6425}_{\pm0.2992}$ | $\mathbf{0.6603}_{\pm0.1592}$ | $\mathbf{0.3853}_{\pm0.1617}$ |
| 0.5% | EEGNet | $0.7068_{\pm0.1674}$ | $0.6735_{\pm0.2036}$ | $0.4812_{\pm0.1654}$ | $0.4917_{\pm0.3267}$ | $0.5192_{\pm0.1962}$ | $0.8548_{\pm0.4273}$ |
| | + TRin | $\mathbf{0.7386}_{\pm0.1777}$ | $\mathbf{0.6924}_{\pm0.2356}$ | $\mathbf{0.4250}_{\pm0.1709}$ | $\mathbf{0.5903}_{\pm0.3749}$ | $\mathbf{0.6284}_{\pm0.1617}$ | $\mathbf{0.4008}_{\pm0.1940}$ |
| | TSception | $0.6815_{\pm0.1667}$ | $0.6319_{\pm0.2140}$ | $0.5073_{\pm0.1849}$ | $0.4473_{\pm0.3339}$ | $0.5422_{\pm0.2056}$ | $0.6111_{\pm0.4207}$ |
| | + TRin | $\mathbf{0.7396}_{\pm0.1768}$ | $\mathbf{0.7114}_{\pm0.2142}$ | $\mathbf{0.4300}_{\pm0.1721}$ | $\mathbf{0.5522}_{\pm0.3810}$ | $\mathbf{0.6101}_{\pm0.1957}$ | $\mathbf{0.5697}_{\pm0.2789}$ |
| | Deformer | $0.7183_{\pm0.1614}$ | $0.6841_{\pm0.1995}$ | $0.4673_{\pm0.1781}$ | $0.5074_{\pm0.3189}$ | $0.5167_{\pm0.2381}$ | $0.9761_{\pm0.5158}$ |
| | + TRin | $\mathbf{0.7619}_{\pm0.1548}$ | $\mathbf{0.7293}_{\pm0.2043}$ | $\mathbf{0.3996}_{\pm0.1508}$ | $\mathbf{0.6402}_{\pm0.3149}$ | $\mathbf{0.6527}_{\pm0.1626}$ | $\mathbf{0.4455}_{\pm0.2102}$ |

## K  SUBJECT CONSISTENCY

In this section, we present the consistency of TRin framework improvement across each subject. We calculate the average metrics of TRin framework over four models (EEGNet, TSception, Deformer, and CBraMod) on each subject and compare with the baseline. The results using Mental Workload dataset for Accuracy, FSE, and FDS$_{norm}$ are shown in Figure 6, Figure 7, and Figure 8 respectively.

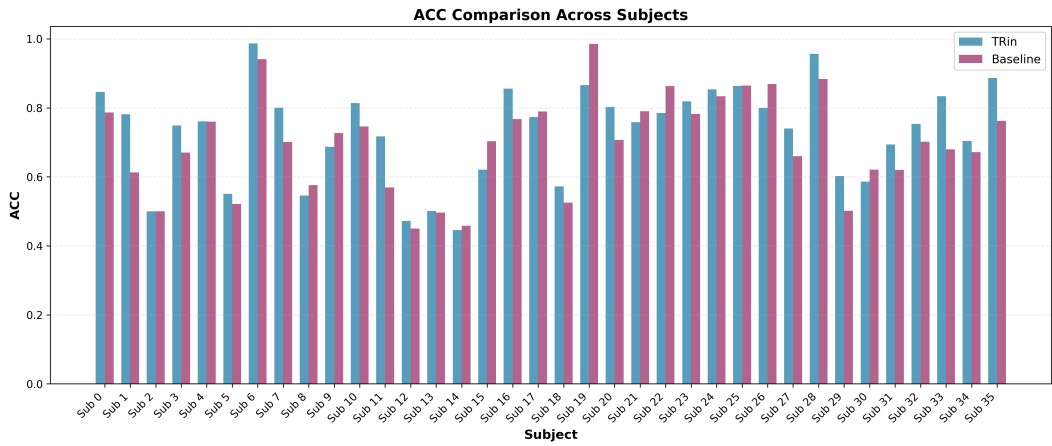

Figure 6: Accuracy comparison between baseline and TRin framework across different subjects using Mental Workload dataset.

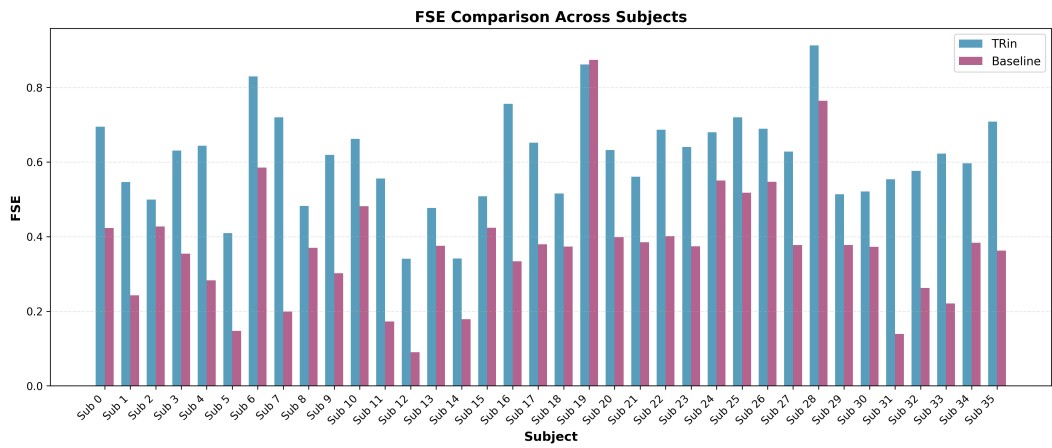

Figure 7: FSE comparison between baseline and TRin framework across different subjects using Mental Workload dataset.

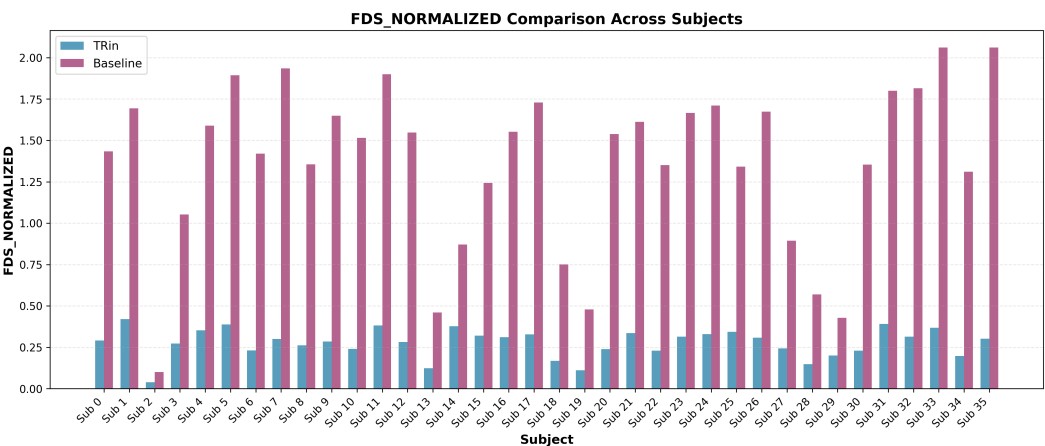

Figure 8: FDS$_{norm}$ comparison between baseline and TRin framework across different subjects using Mental Workload dataset.

## L DISCUSSION

Our findings demonstrate that TRin enhances temporal stability while maintaining or improving classification accuracy. The reduction in output fluctuations directly addresses the temporal instability that hinders the effectiveness of real-time applications, such as neurofeedback training. From a neuroscience perspective, temporal regularization shows potential in increasing model consistency over time, which more accurately reflects the continuous nature of mental states. Furthermore, we propose that instability in current DL-based BCIs may indicate overfitting: models optimized solely for instantaneous accuracy might overlook simple yet essential neuroscientific principles that foster temporal coherence. Thus, we advocate for the BCI community to embrace evaluation criteria that incorporate such foundational neuroscientific principles to evaluate system efficacy more accurately.

Despite the significant improvements achieved by TRin, several limitations should be acknowledged: sensitivity to hyperparameters, requirements for trial lengths, and increased training costs. As detailed in the sensitivity analysis (Appendix G.1), the optimal regularization parameters ($\alpha_s$ and clipRate) differ across models and datasets, necessitating meticulous retuning for new applications. Moreover, sequence-preserving data handling necessitates more temporally sequenced samples, resulting in smaller TRin gains for datasets with shorter trials (e.g., Attention), compared to those like Mental Workload and SEED. The creation of augmented sequence-preserving training data also

increases training costs due to the need for larger training overlaps, although inference remains efficient. Future research will explore algorithmic adjustments to lessen overlap requirements and better accommodate shorter trials.

Our results indicate several promising research avenues, including multi-modal integration, personalized frameworks, and training strategies that align more closely with neuroscience. Future efforts will extend the principles of temporal robustness to adaptive BCIs and multi-modal physiological streams (e.g., combining EEG with peripheral signals or facial expressions), potentially enhancing robustness and contextual awareness. Moreover, existing studies suggest that personalized BCI systems can improve real-time neurofeedback training. Personalized frameworks can dynamically adjust temporal-regularization hyperparameters to align with individual user characteristics and state variability. Additionally, the benefits of temporal regularization in DL-based BCIs imply that integrating further neuroscience principles may also lead to practical gains. We hope this work sets the stage for continued exploration of temporal-regularization and neuroscience-aligned strategies in DL-based BCIs, facilitating their translation into practical, user-centric systems.

## M  LLM USAGE DISCLOSURE

ChatGPT 4o was utilized during the final writing phase, only for identifying typos, grammatical errors, and slightly awkward phrasing in the near-final manuscript draft. LLMs contributed neither to the core research ideas, algorithm development, model design decisions, implementation, experimental analysis, result interpretation, nor to formulating the paper's core contributions and conclusions. Authors conceived the research, conducted the work, and drafted the manuscript's substantive content.

## N  ADDITIONAL VISUALIZATIONS

The results of appendix show the visualizations of predictions for different models on different datasets. For each trial, the visualizations show four methods: Baseline (BL), Baseline with Label Smoothing (BL + LS), TRin replaced by MSE loss (DyM), and full TRin (DyS). The first row shows BL and BL + LS, while the second row shows DyM and DyS.

## N.1 MENTAL WORKLOAD DATASET

### N.1.1 EEGNET ON MENTAL WORKLOAD DATASET

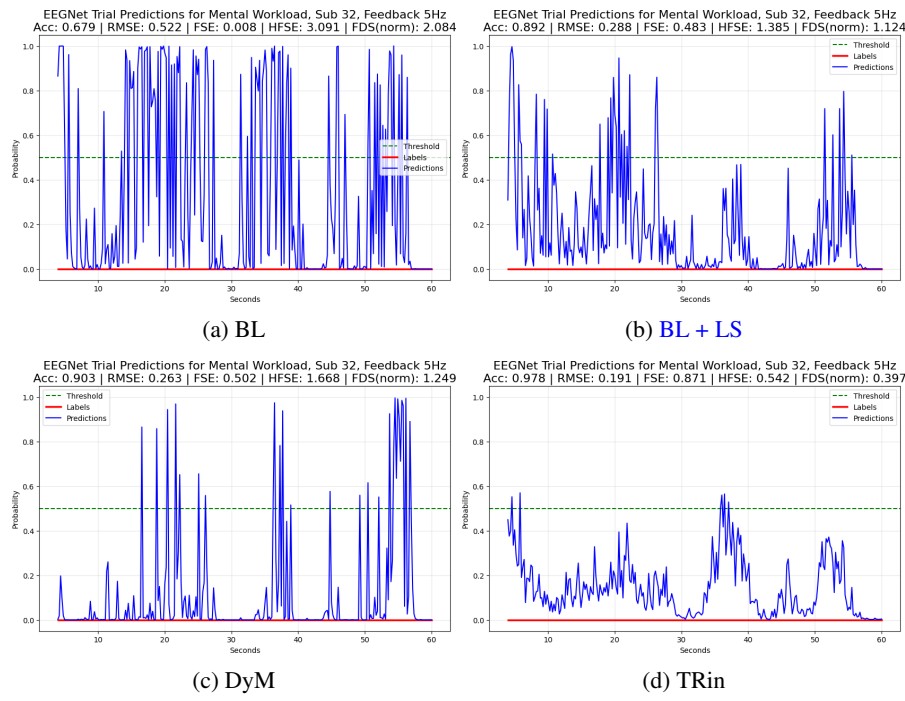

Figure 9: Mental Workload Dataset - Sub32 Trial1 (EEGNet Model)

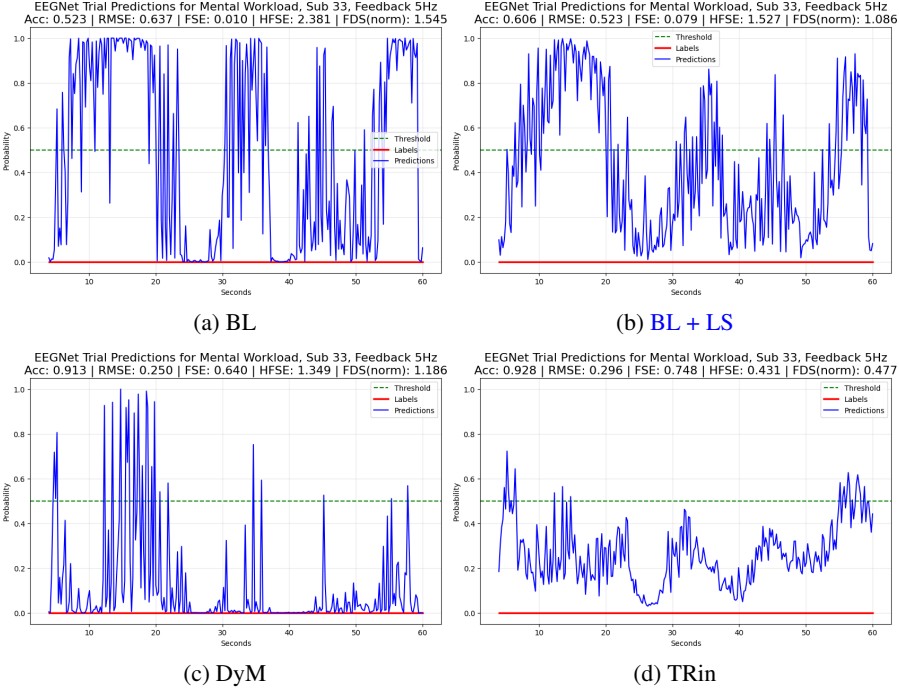

Figure 10: Mental Workload Dataset - Sub33 Trial1 (EEGNet Model)

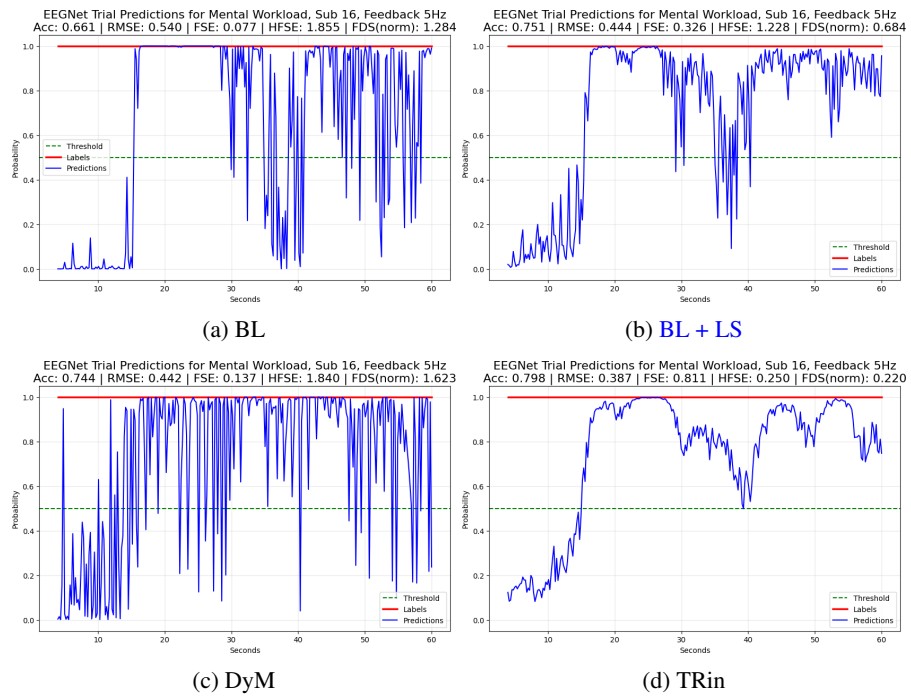

Figure 11: Mental Workload Dataset - Sub16 Trial2 (EEGNet Model)

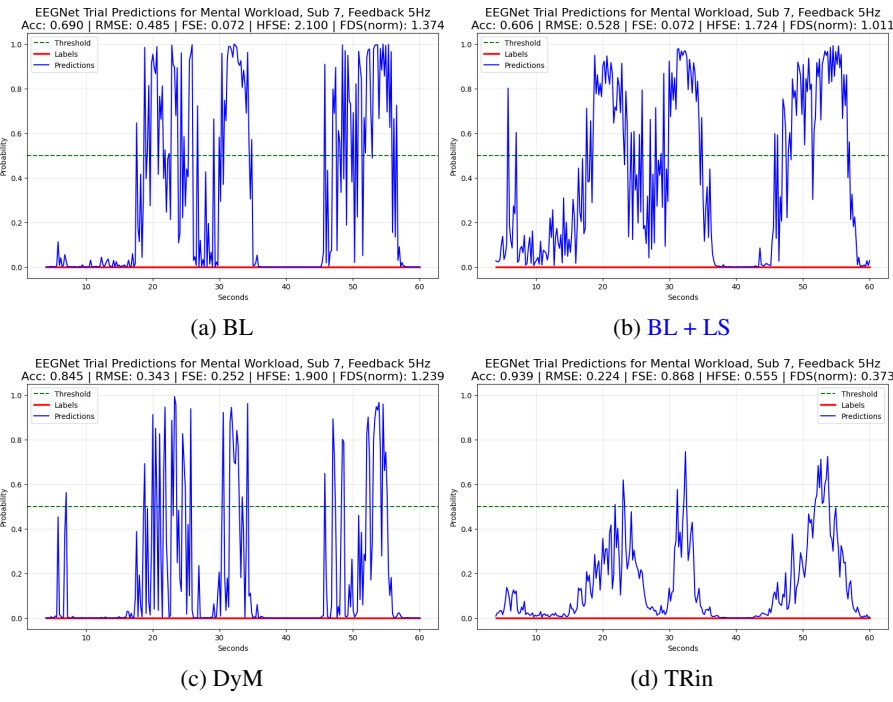

Figure 12: Mental Workload Dataset - Sub7 Trial1 (EEGNet Model)

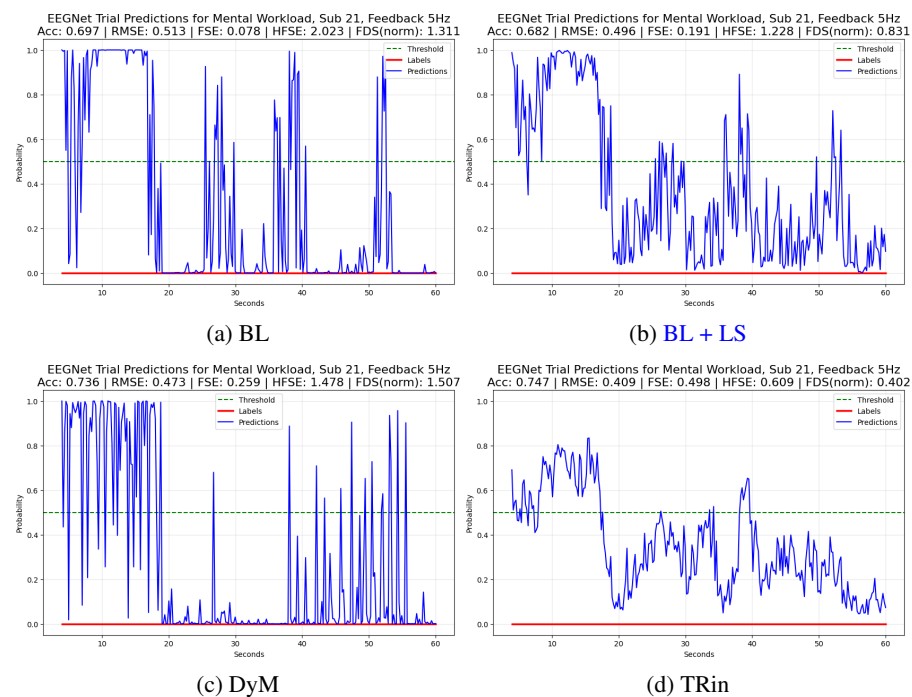

Figure 13: Mental Workload Dataset - Sub21 Trial1 (EEGNet Model)

### N.1.2 TSCEPTION ON MENTAL WORKLOAD DATASET

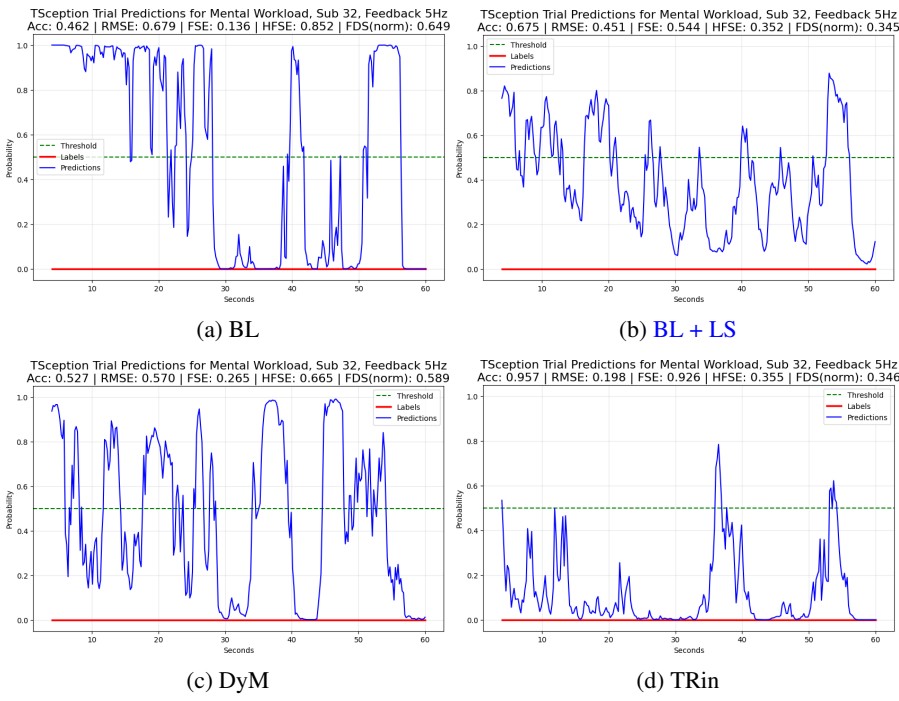

Figure 14: Mental Workload Dataset - Sub32 Trial1 (TSception Model)

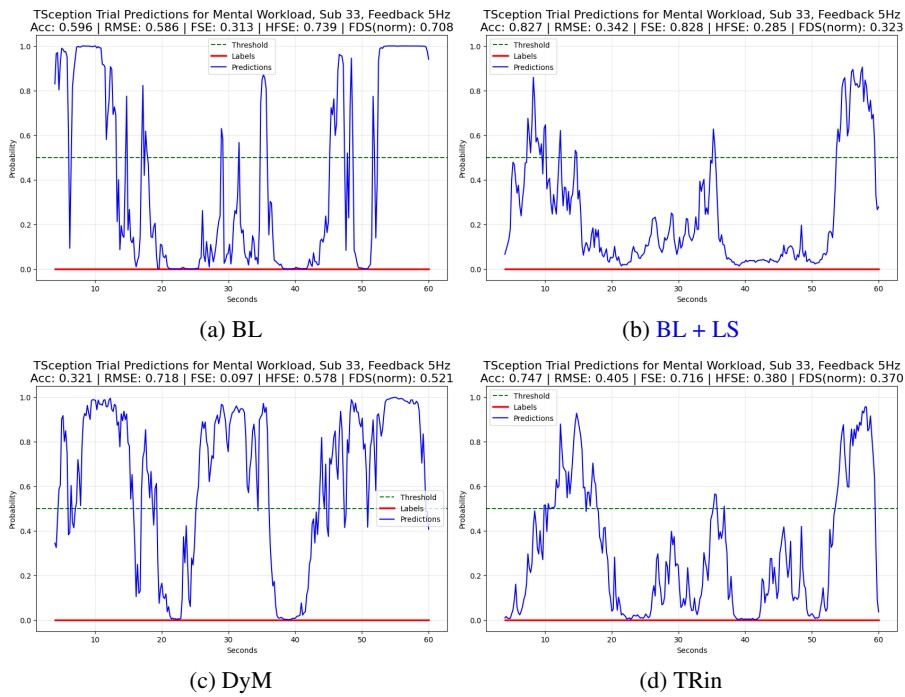

Figure 15: Mental Workload Dataset - Sub33 Trial1 (TSception Model)

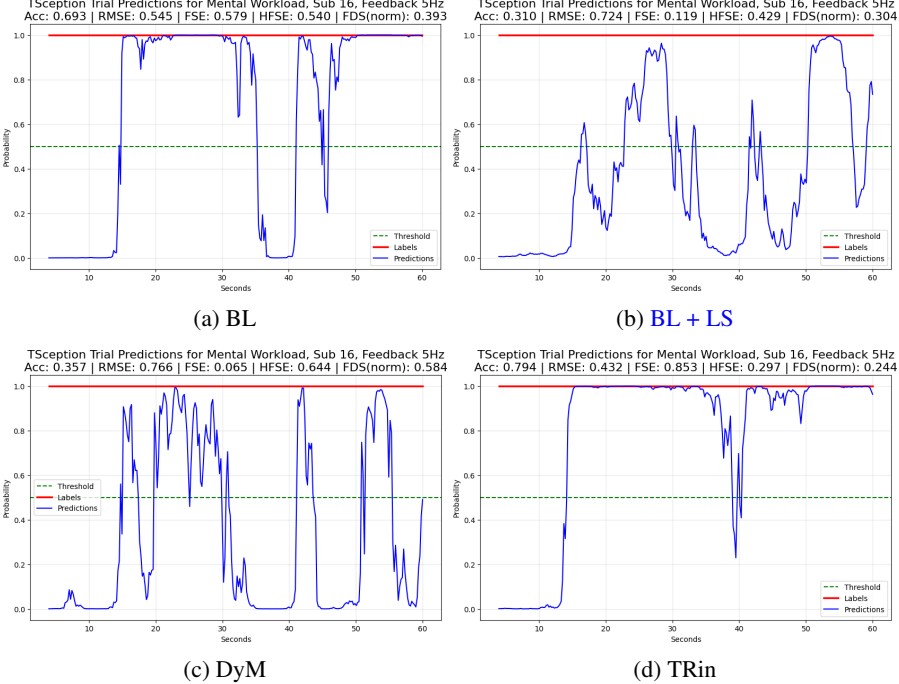

Figure 16: Mental Workload Dataset - Sub16 Trial2 (TSception Model)

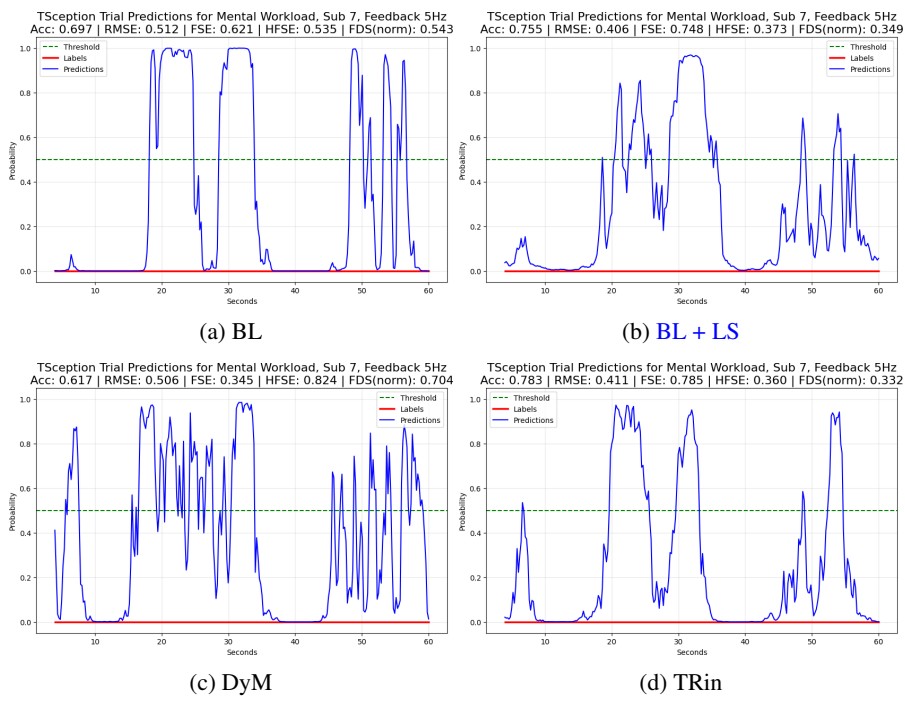

Figure 17: Mental Workload Dataset - Sub7 Trial1 (TSception Model)

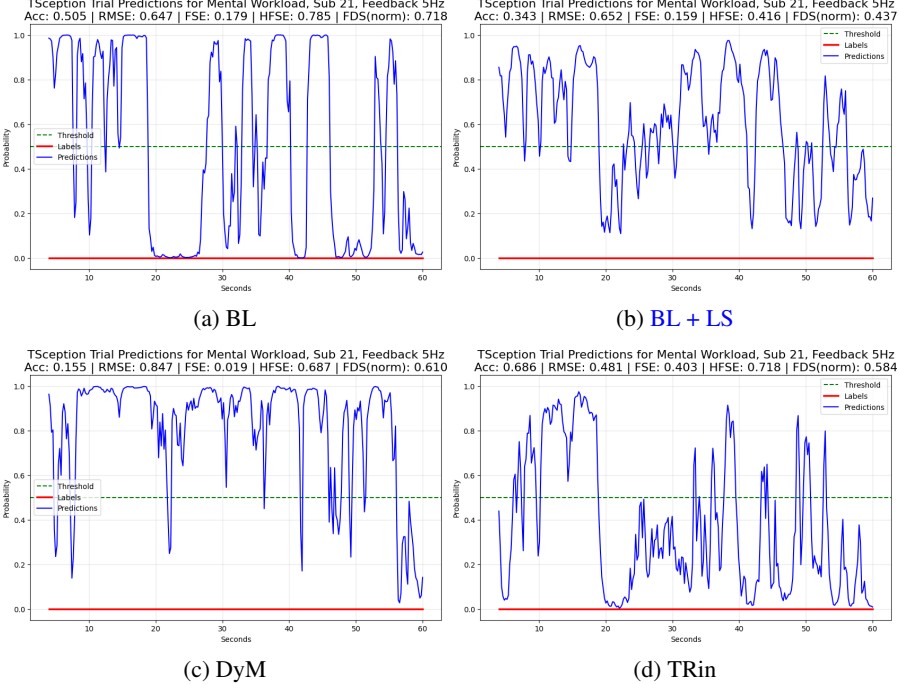

Figure 18: Mental Workload Dataset - Sub21 Trial1 (TSception Model)

### N.1.3 DEFORMER ON MENTAL WORKLOAD DATASET

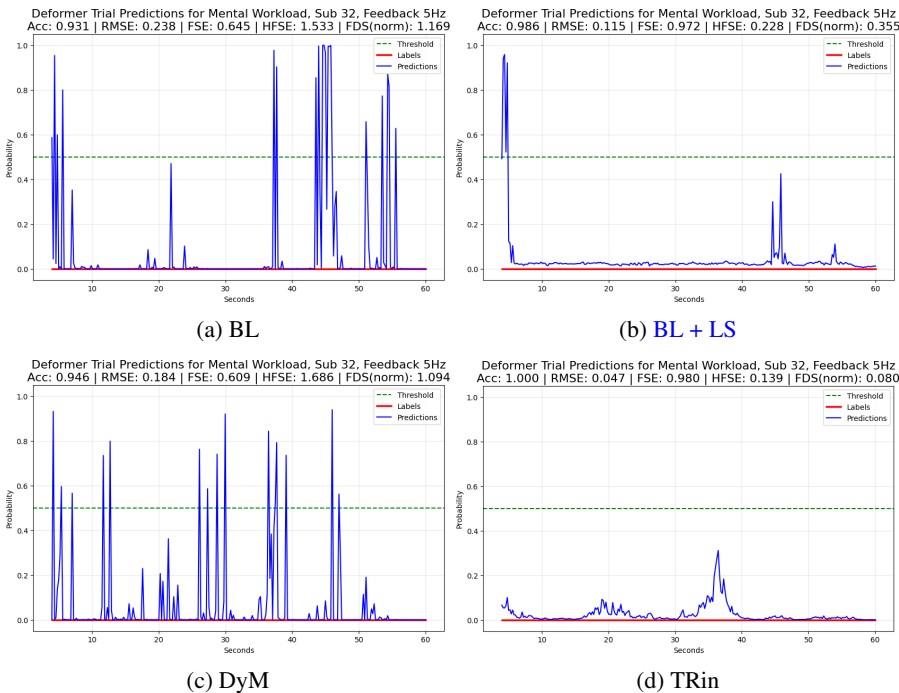

Figure 19: Mental Workload Dataset - Sub32 Trial1 (Deformer Model)

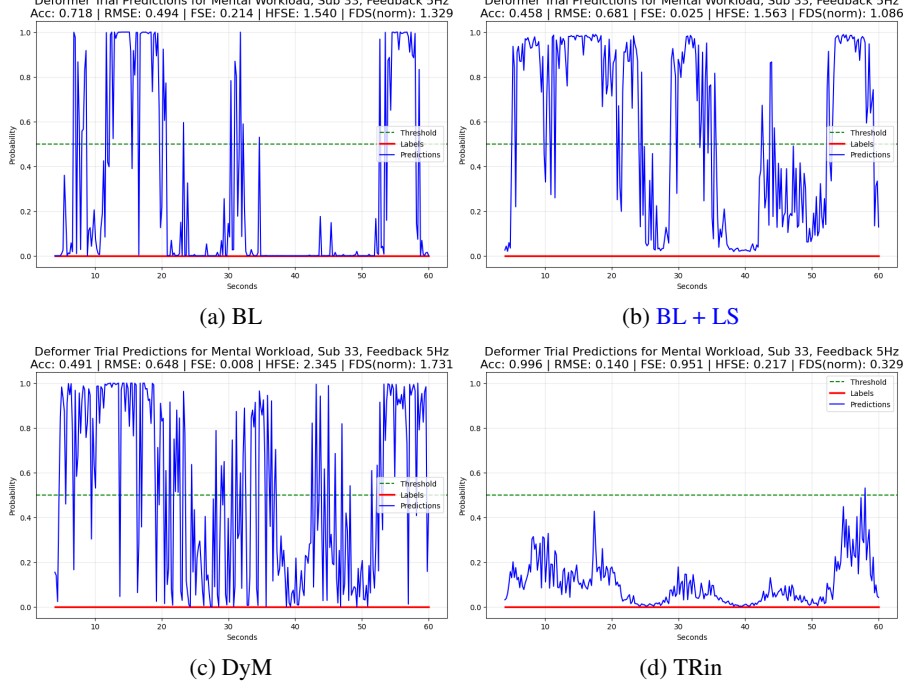

Figure 20: Mental Workload Dataset - Sub33 Trial1 (Deformer Model)

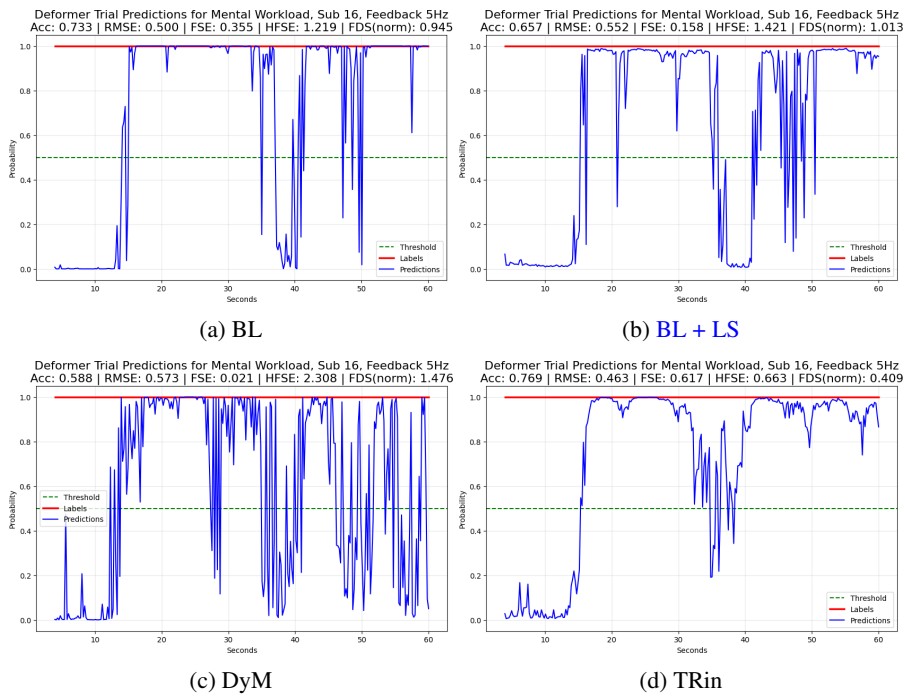

Figure 21: Mental Workload Dataset - Sub16 Trial2 (Deformer Model)

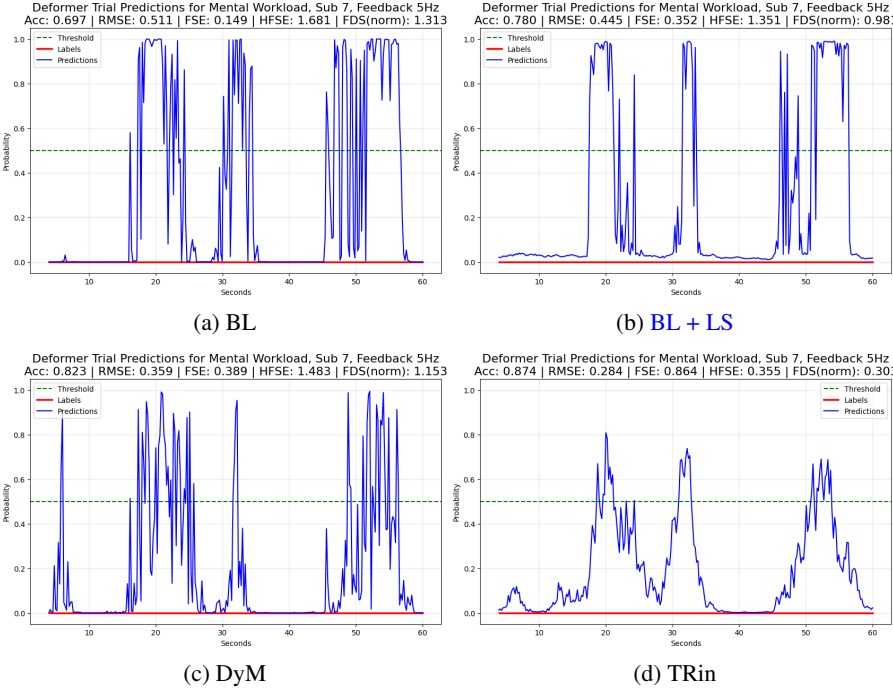

Figure 22: Mental Workload Dataset - Sub7 Trial1 (Deformer Model)

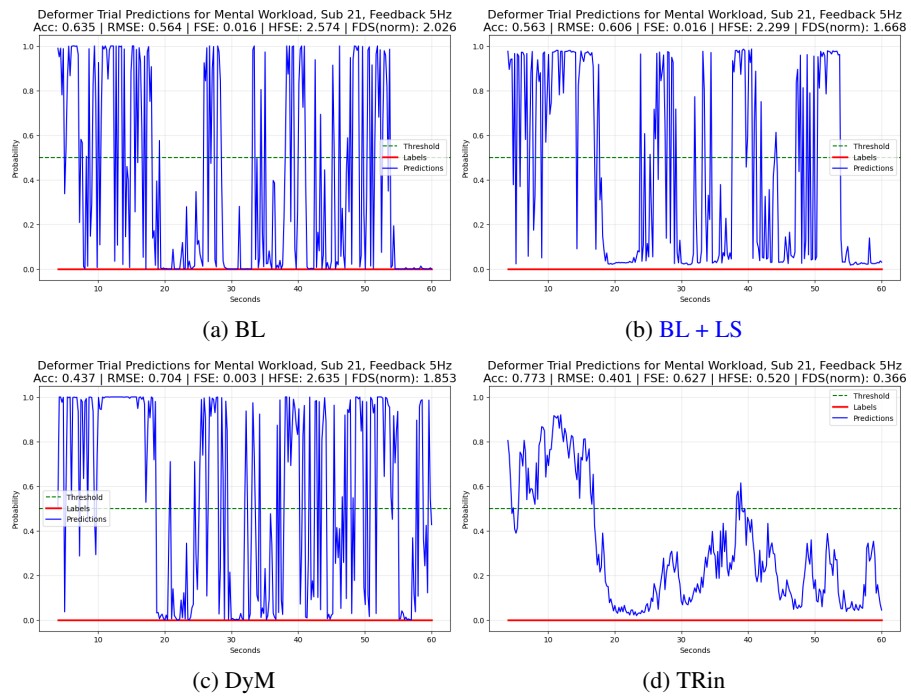

Figure 23: Mental Workload Dataset - Sub21 Trial1 (Deformer Model)

## N.2 SEED DATASET

### N.2.1 EEGNET ON SEED DATASET

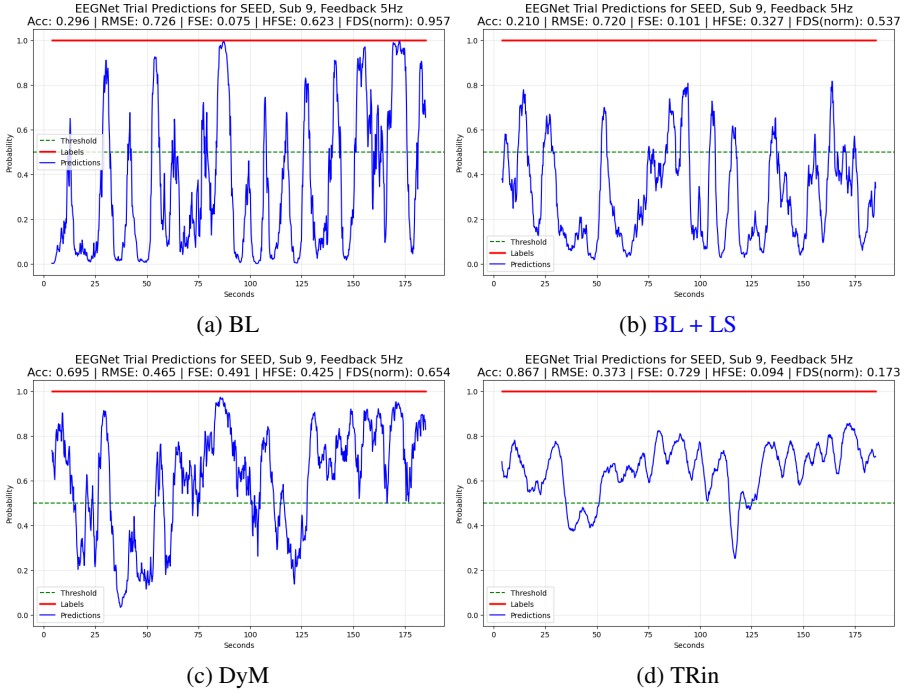

Figure 24: SEED Dataset - Sub9 Trial1 (EEGNet Model)

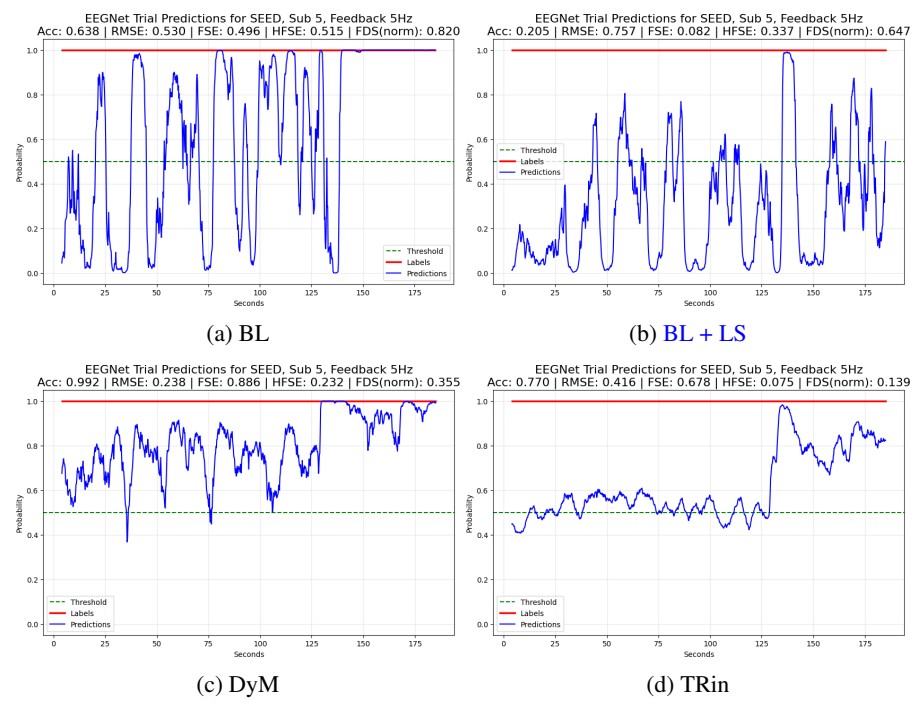

Figure 25: SEED Dataset - Sub5 Trial4 (EEGNet Model)

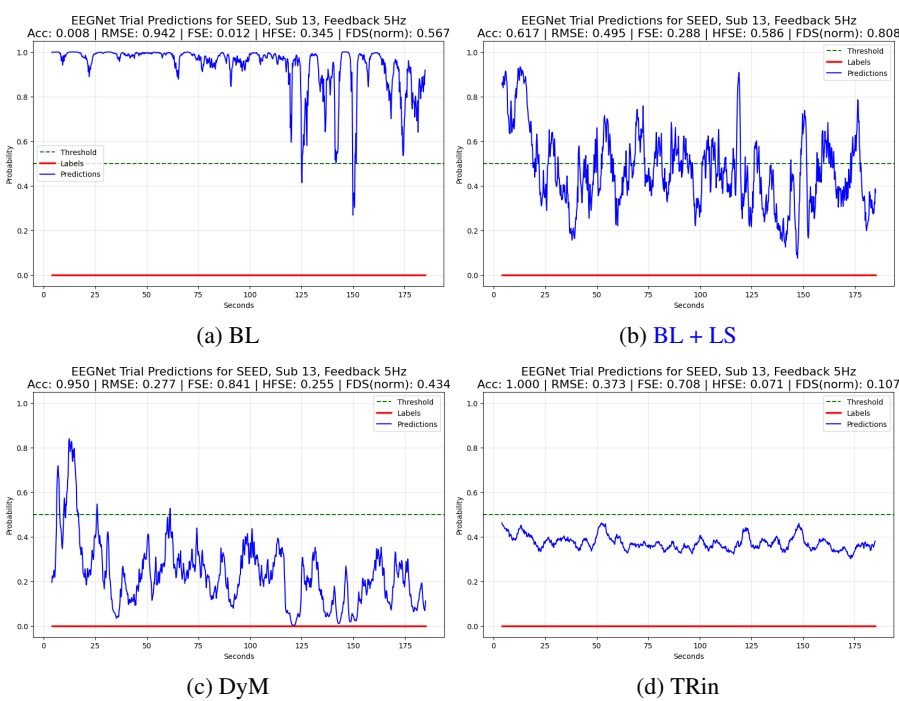

Figure 26: SEED Dataset - Sub13 Trial2 (EEGNet Model)

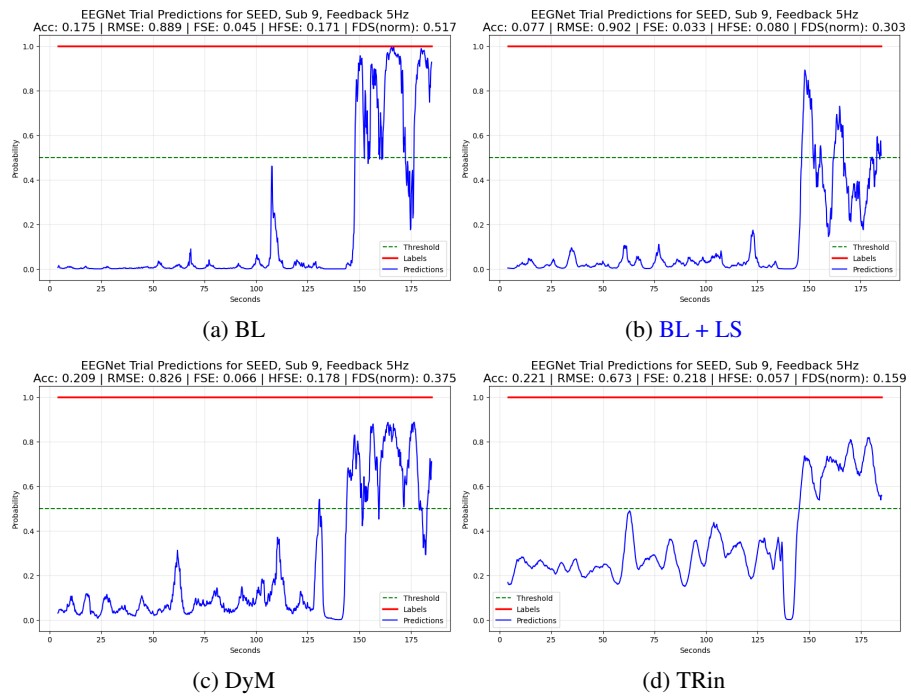

(a) BL

(b) BL + LS

(c) DyM

(d) TRin

Figure 27: SEED Dataset - Sub9 Trial9 (EEGNet Model)

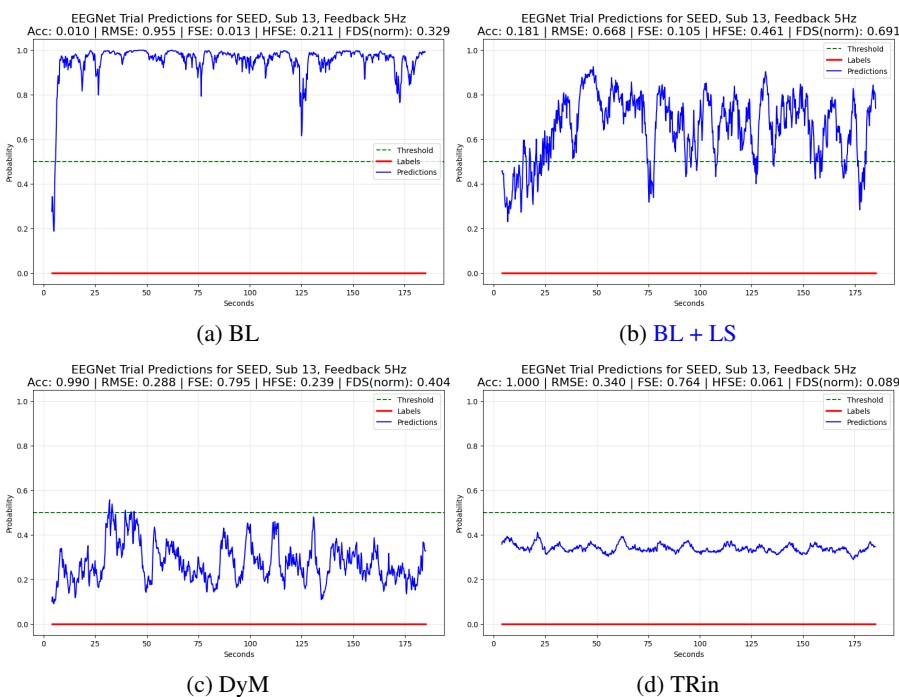

(a) BL

(b) BL + LS

(c) DyM

(d) TRin

Figure 28: SEED Dataset - Sub13 Trial3 (EEGNet Model)

### N.2.2 TSCEPTION ON SEED DATASET

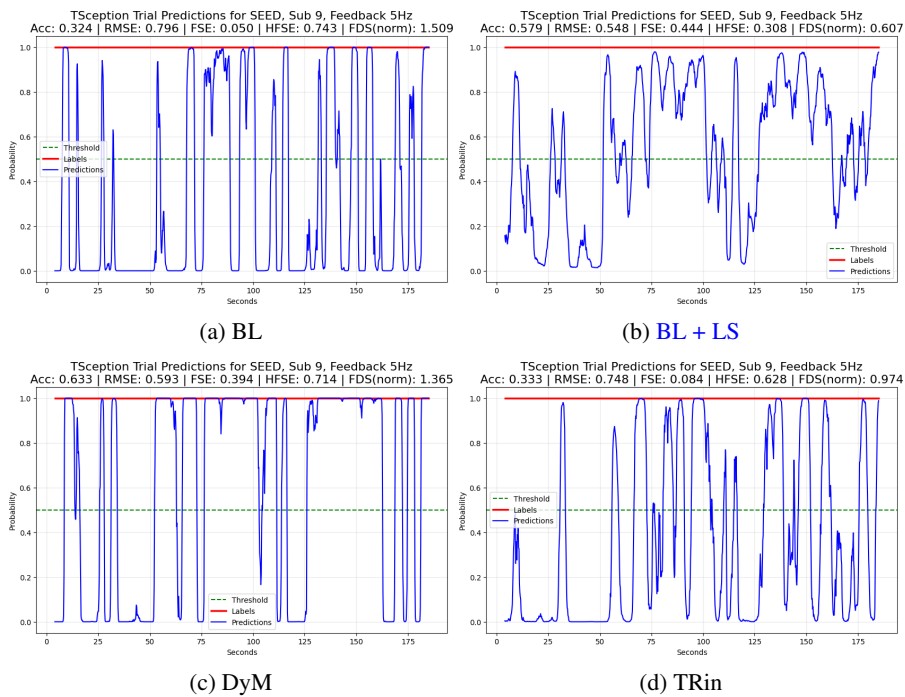

Figure 29: SEED Dataset - Sub9 Trial1 (TSception Model)

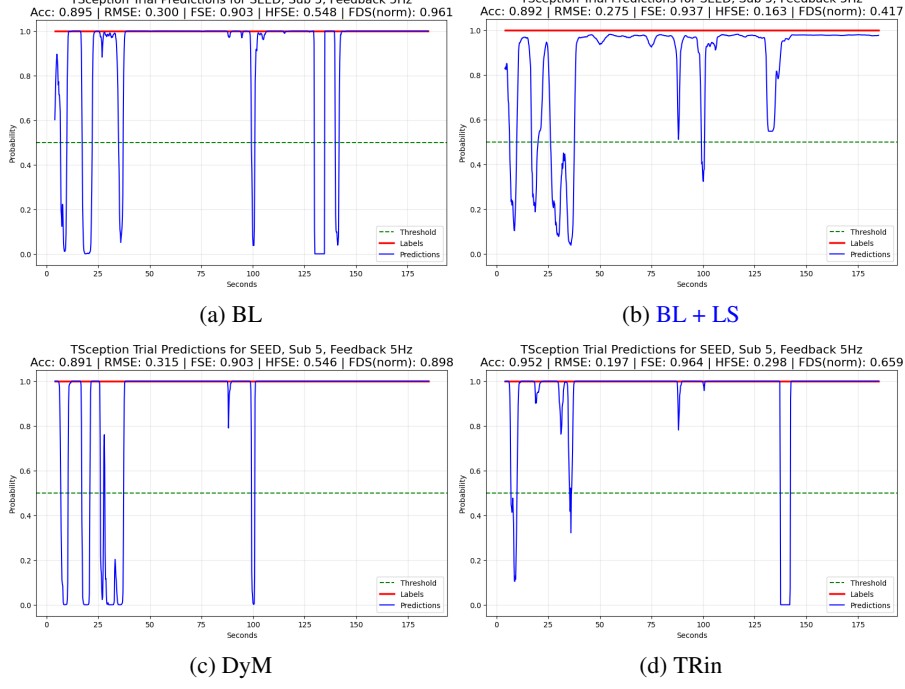

Figure 30: SEED Dataset - Sub5 Trial4 (TSception Model)

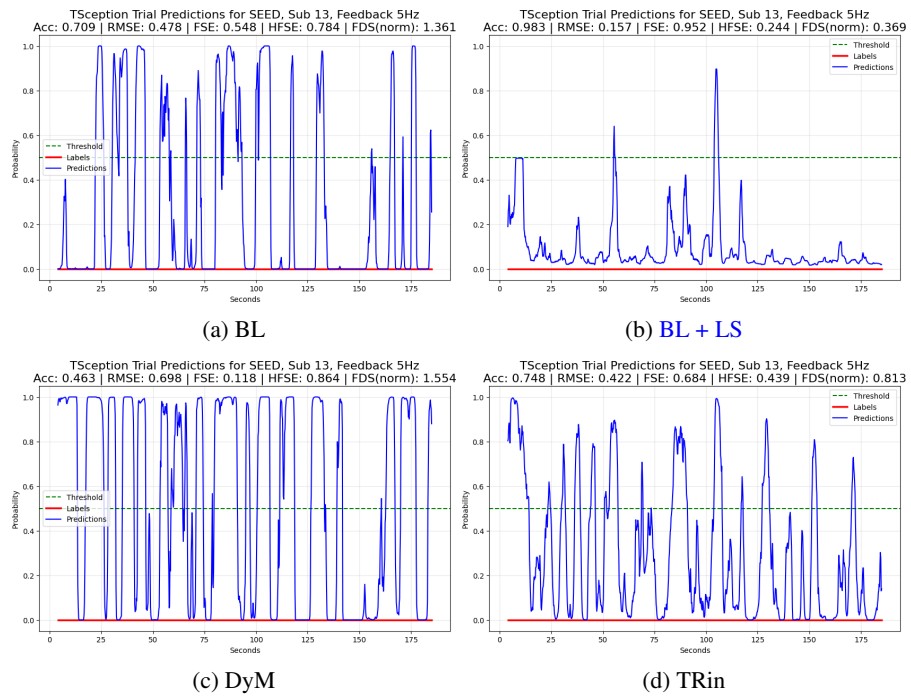

Figure 31: SEED Dataset - Sub13 Trial2 (TSception Model)

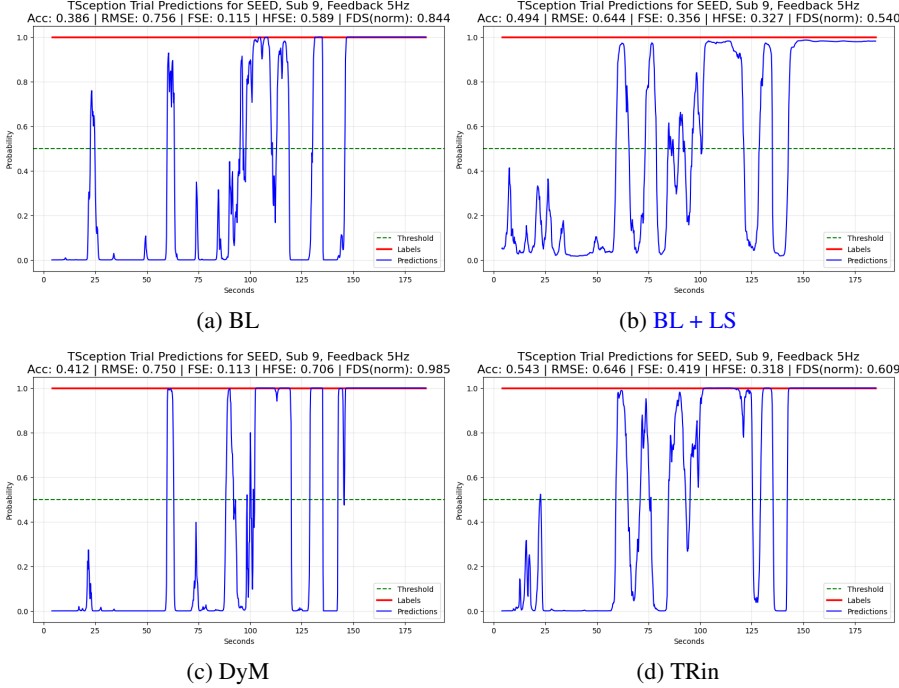

Figure 32: SEED Dataset - Sub9 Trial9 (TSception Model)

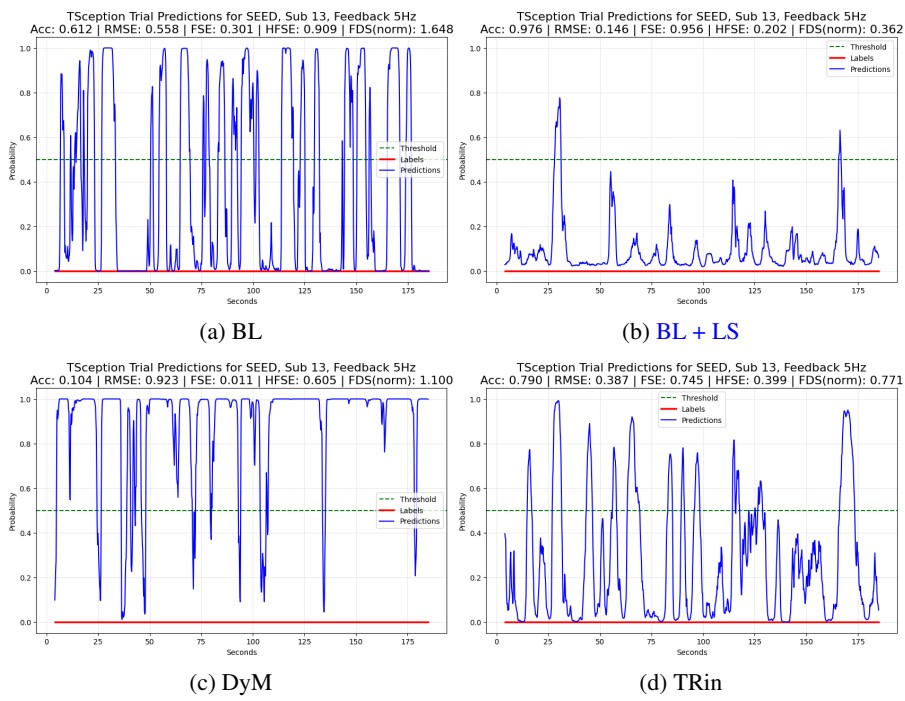

Figure 33: SEED Dataset - Sub13 Trial3 (TSception Model)

### N.2.3 DEFORMER ON SEED DATASET

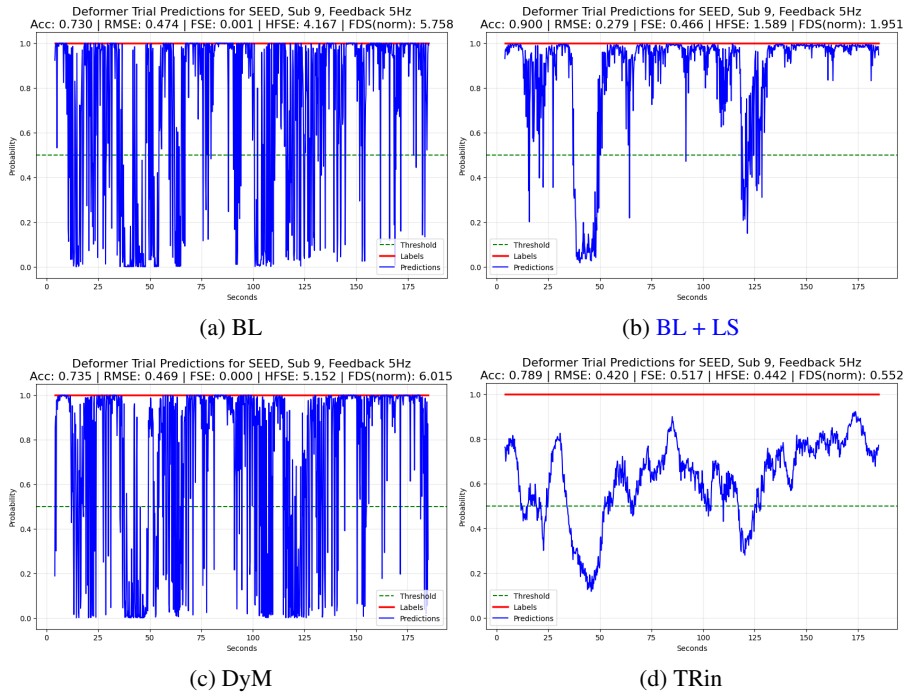

Figure 34: SEED Dataset - Sub9 Trial1 (Deformer Model)

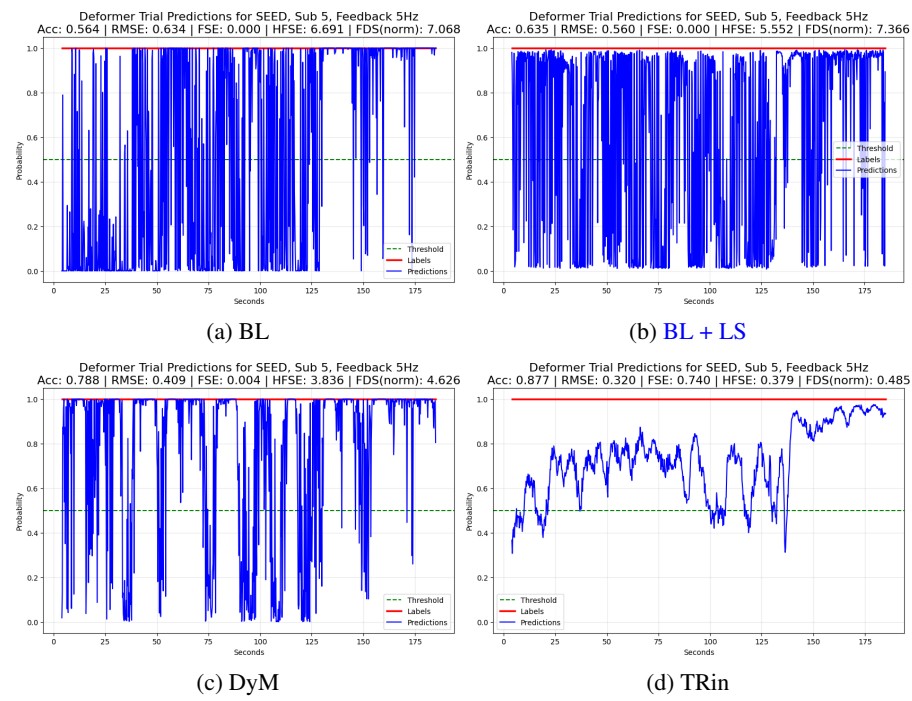

Figure 35: SEED Dataset - Sub5 Trial4 (Deformer Model)

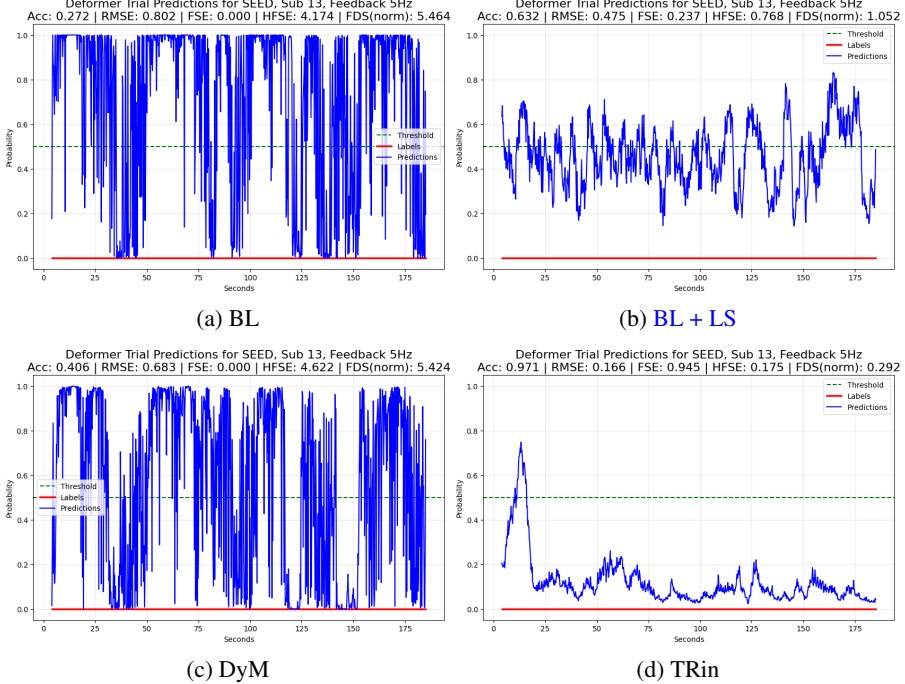

Figure 36: SEED Dataset - Sub13 Trial2 (Deformer Model)

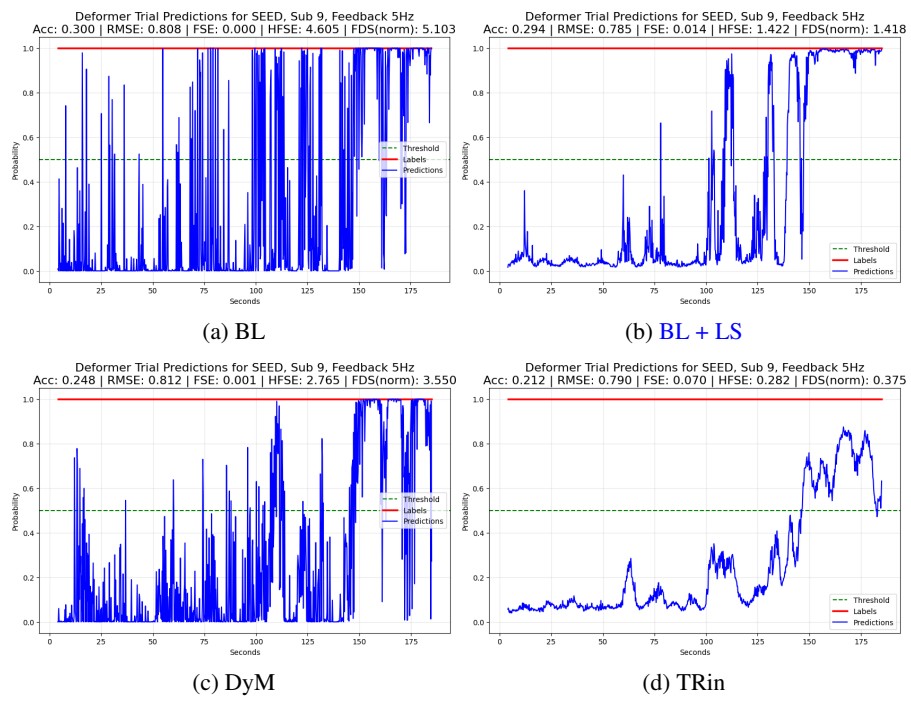

Figure 37: SEED Dataset - Sub9 Trial9 (Deformer Model)

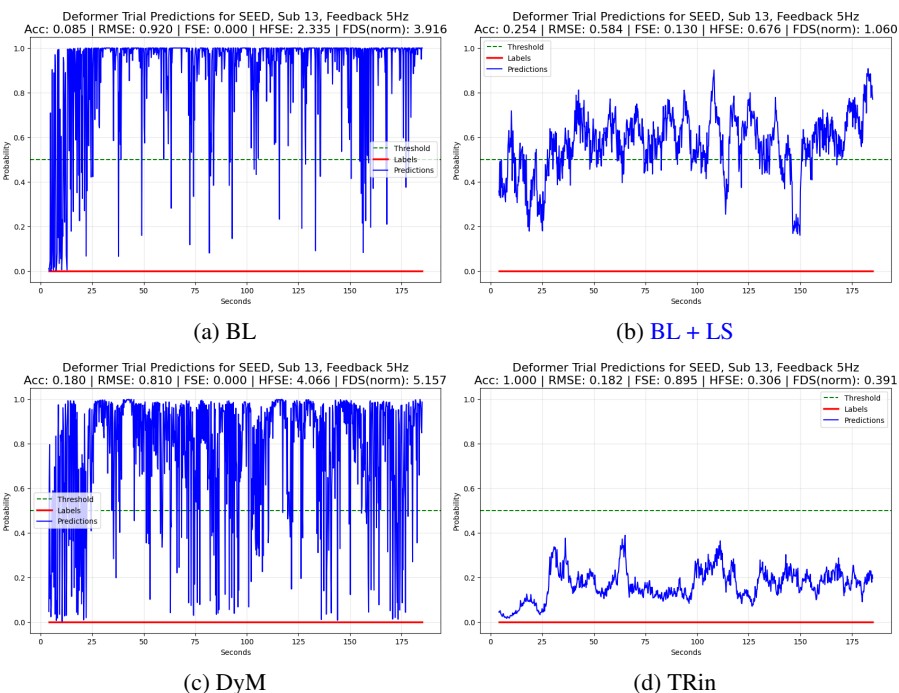

Figure 38: SEED Dataset - Sub13 Trial3 (Deformer Model)

## N.3 ATTENTION DATASET

### N.3.1 EEGNET ON ATTENTION DATASET

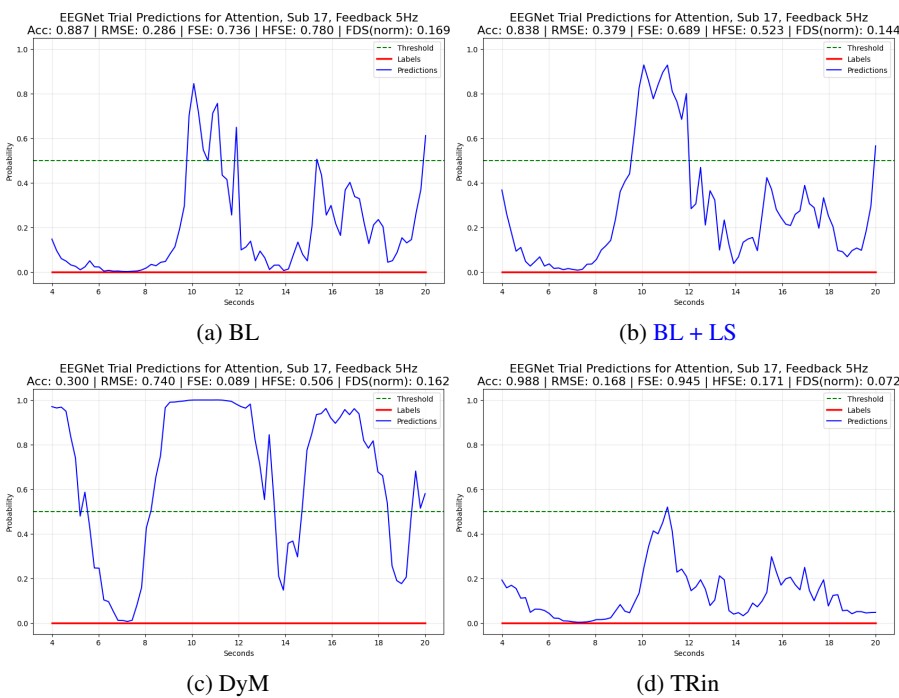

Figure 39: Attention Dataset - Sub17 Trial12 (EEGNet Model)

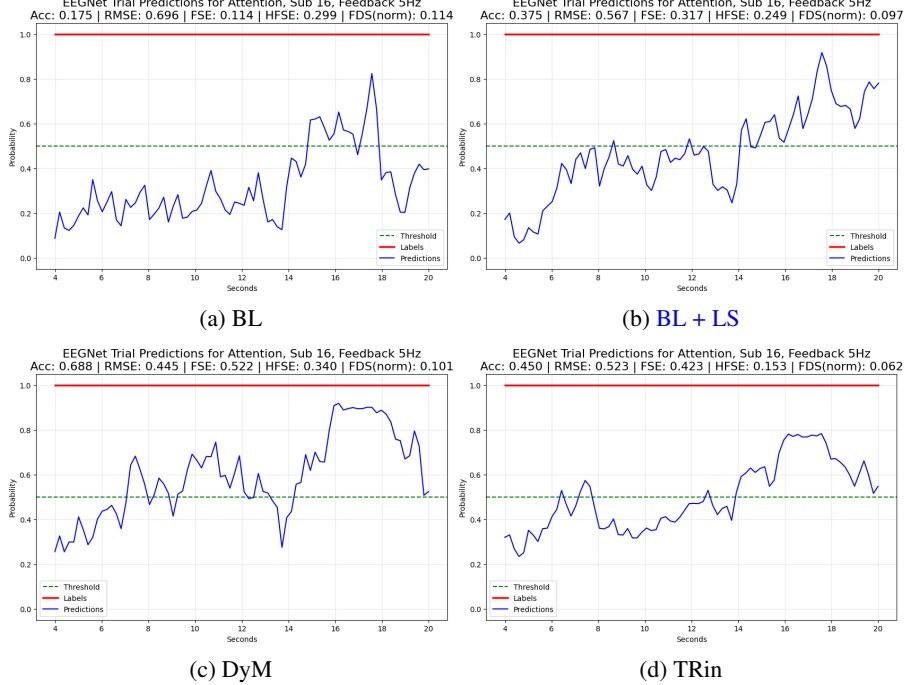

Figure 40: Attention Dataset - Sub16 Trial18 (EEGNet Model)

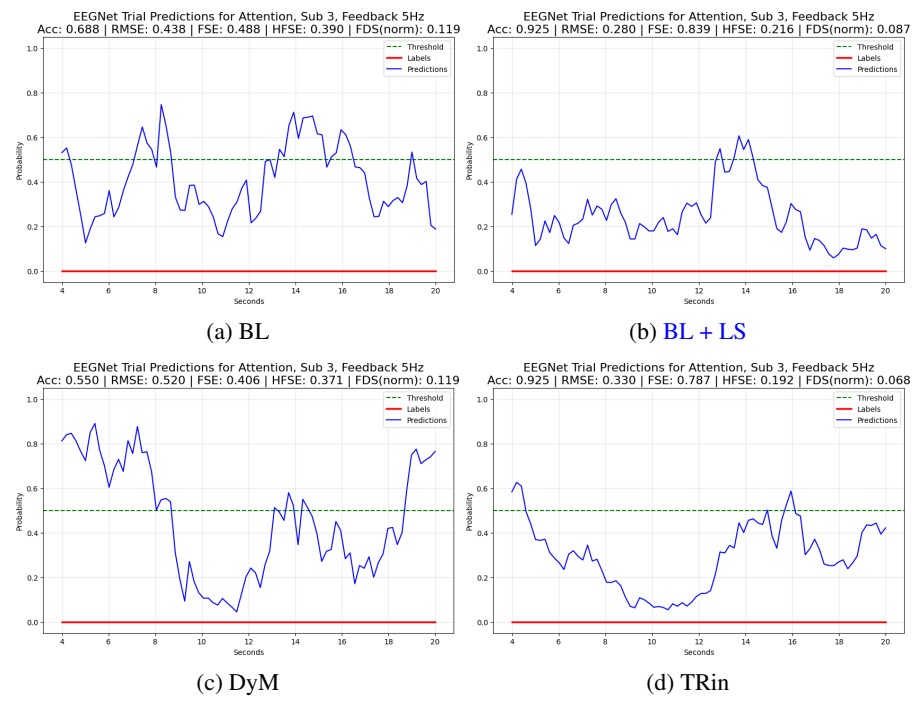

Figure 41: Attention Dataset - Sub3 Trial11 (EEGNet Model)

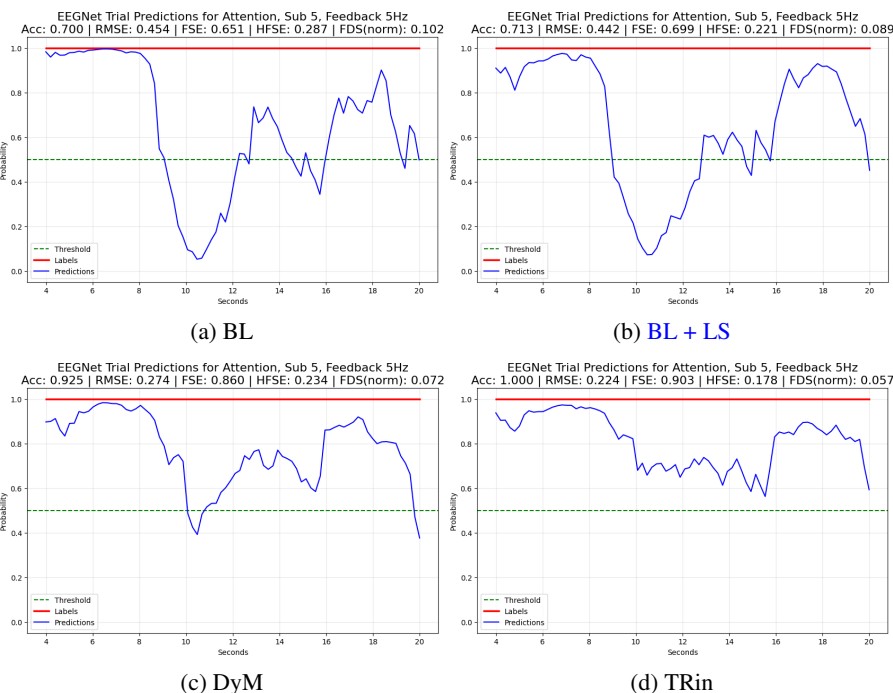

Figure 42: Attention Dataset - Sub5 Trial27 (EEGNet Model)

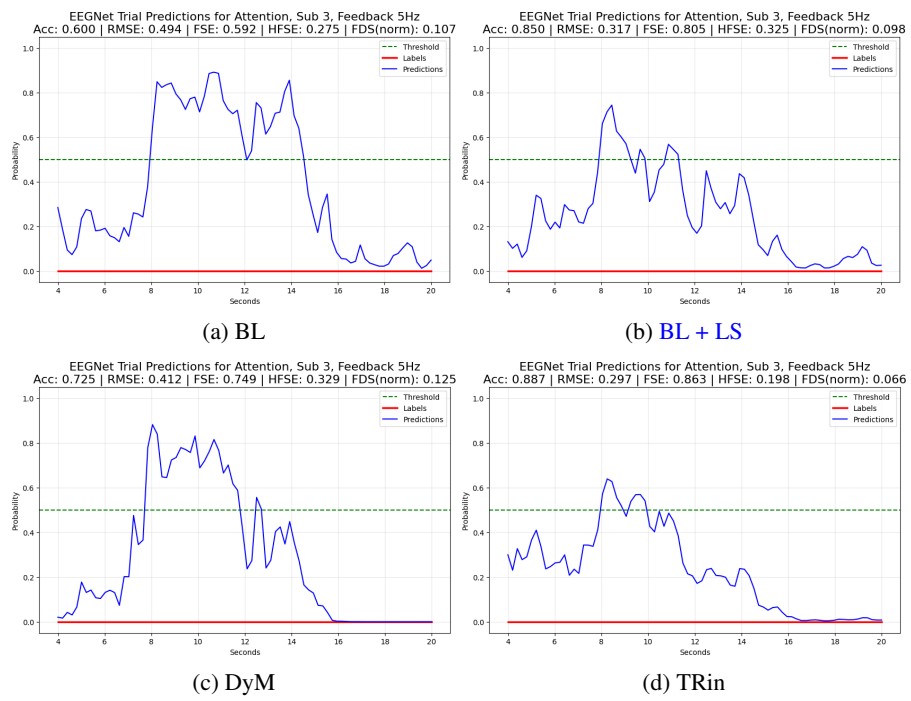

Figure 43: Attention Dataset - Sub3 Trial12 (EEGNet Model)

### N.3.2 TSCEPTION ON ATTENTION DATASET

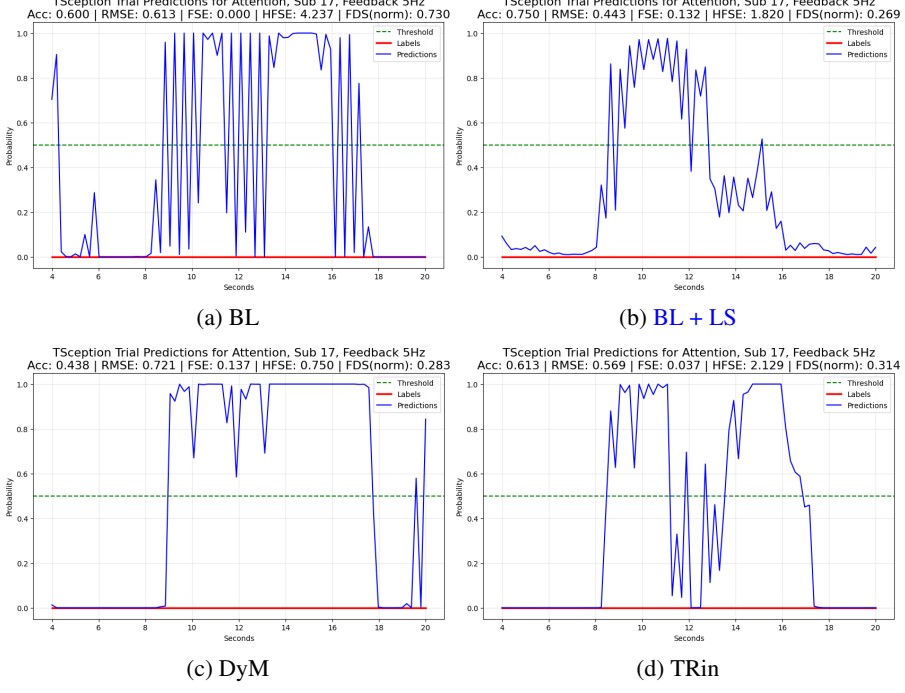

Figure 44: Attention Dataset - Sub17 Trial12 (TSception Model)

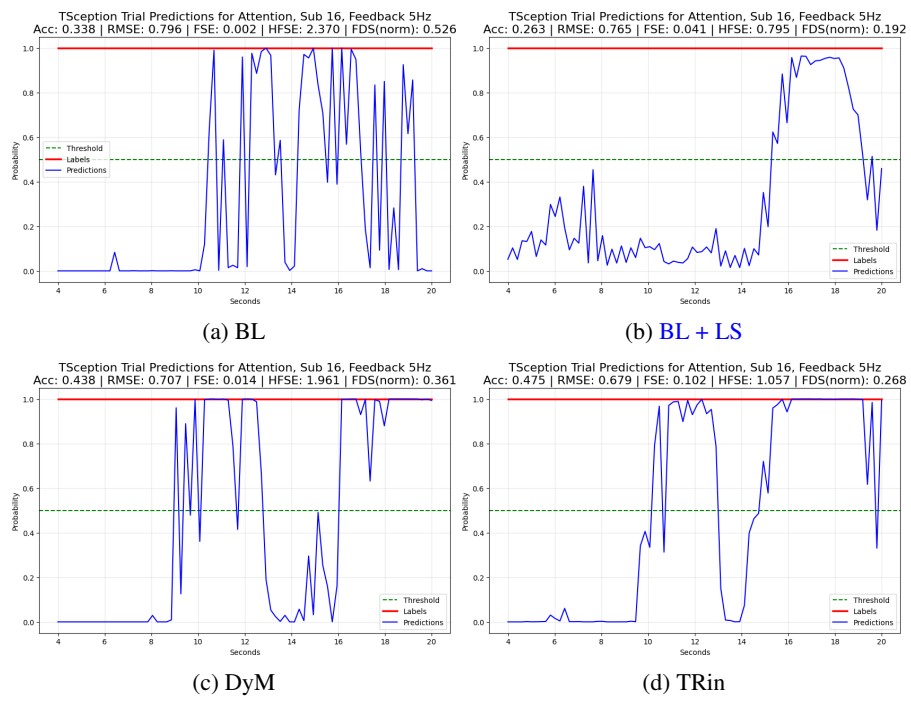

Figure 45: Attention Dataset - Sub16 Trial18 (TSception Model)

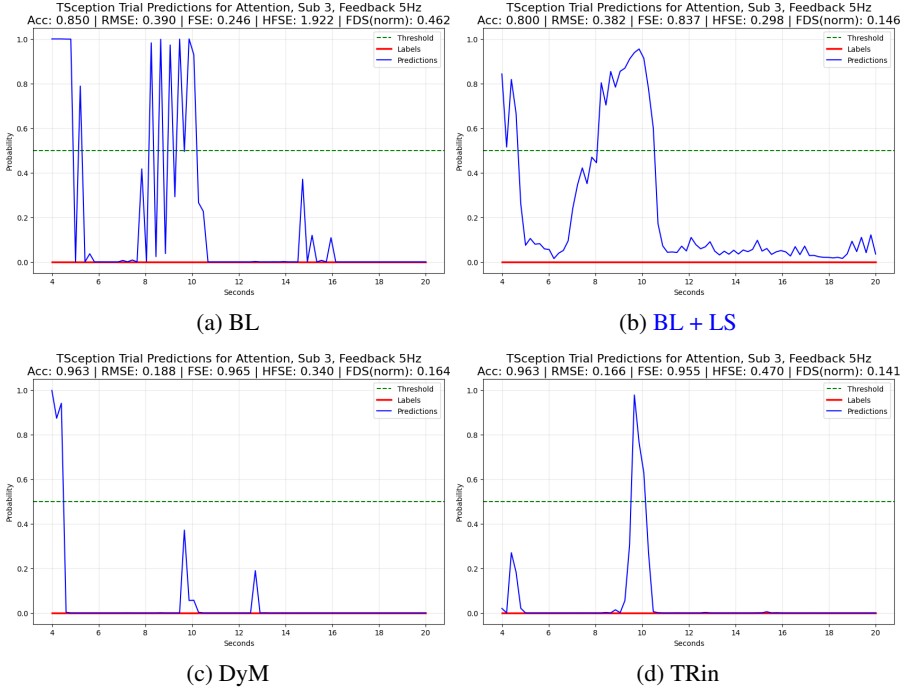

Figure 46: Attention Dataset - Sub3 Trial11 (TSception Model)

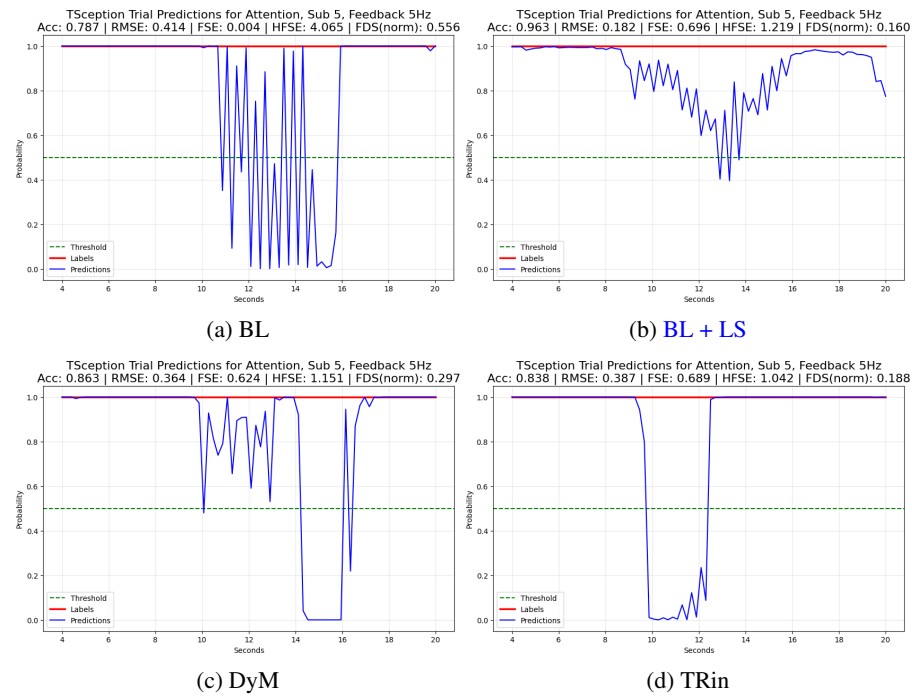

Figure 47: Attention Dataset - Sub5 Trial27 (TSception Model)

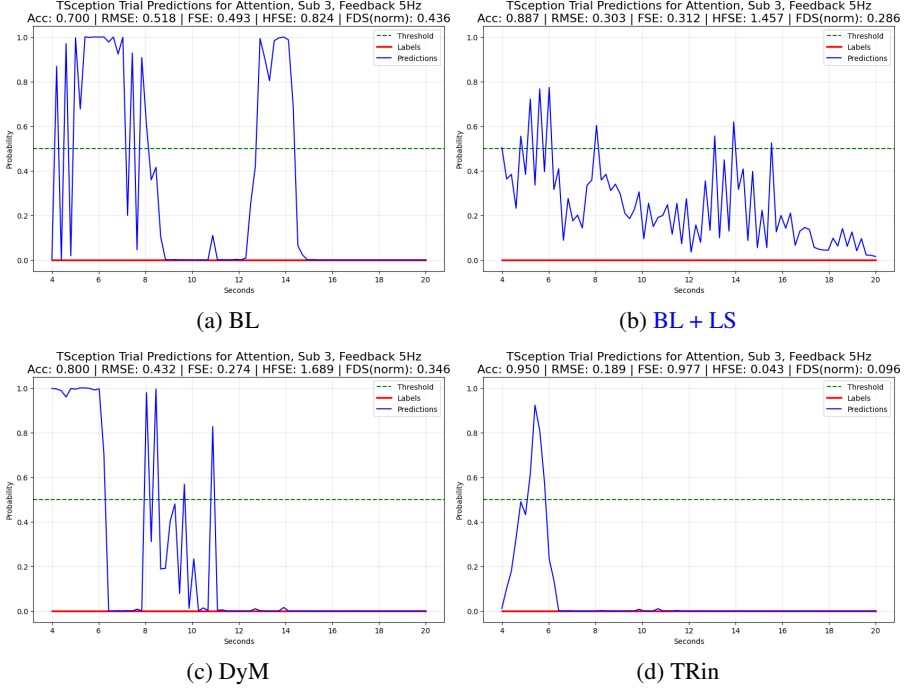

Figure 48: Attention Dataset - Sub3 Trial12 (TSception Model)

### N.3.3 DEFORMER ON ATTENTION DATASET

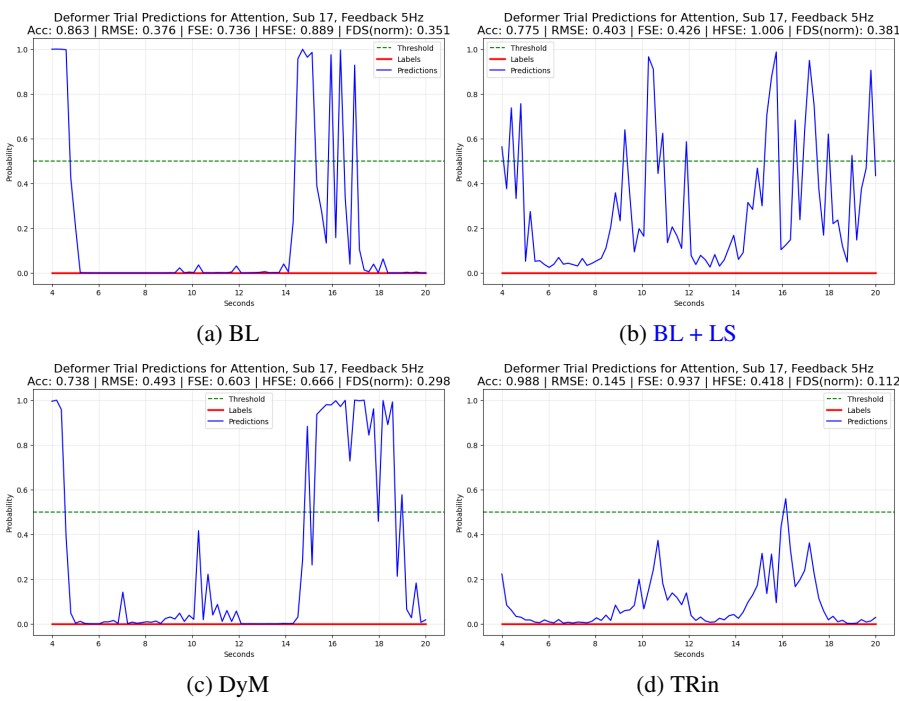

Figure 49: Attention Dataset - Sub17 Trial12 (Deformer Model)

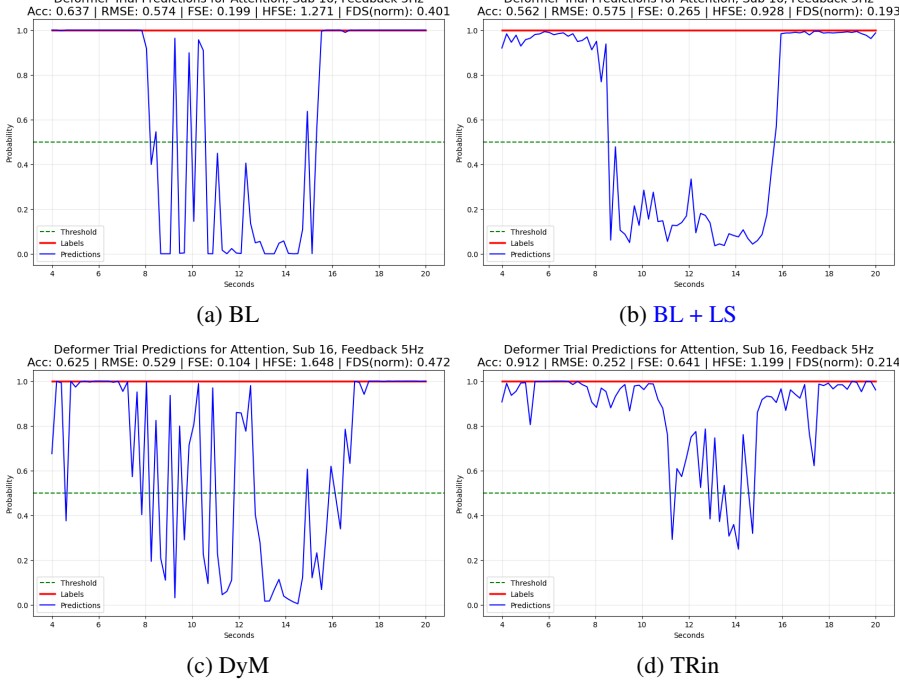

Figure 50: Attention Dataset - Sub16 Trial18 (Deformer Model)

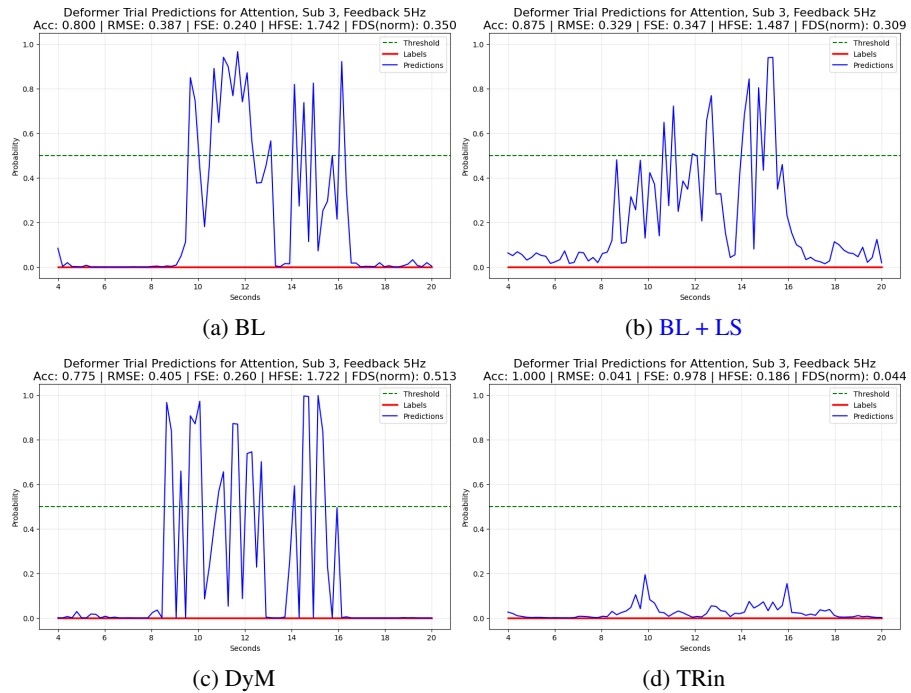

Figure 51: Attention Dataset - Sub3 Trial11 (Deformer Model)

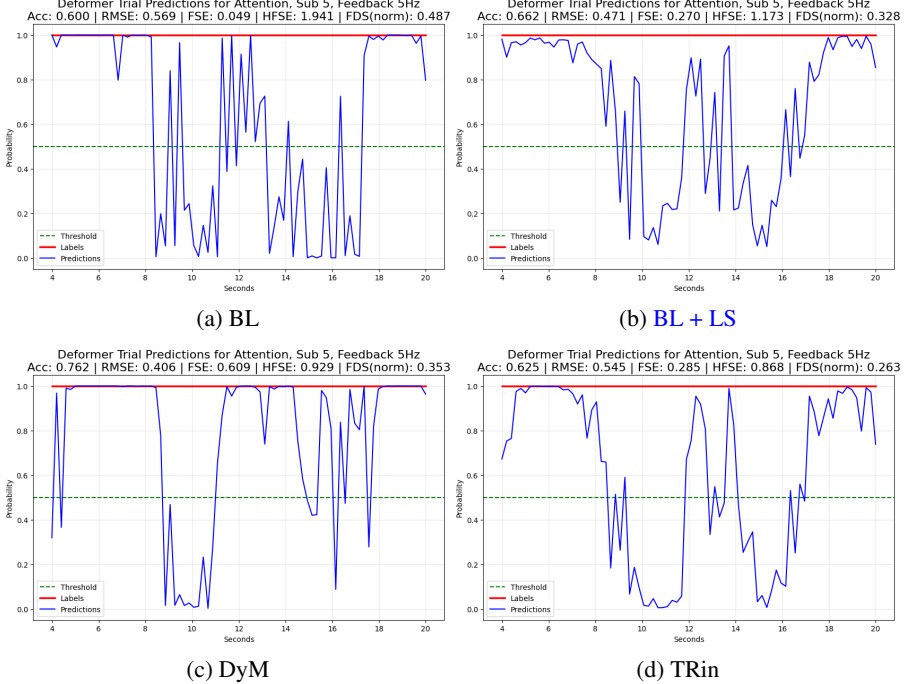

Figure 52: Attention Dataset - Sub5 Trial27 (Deformer Model)

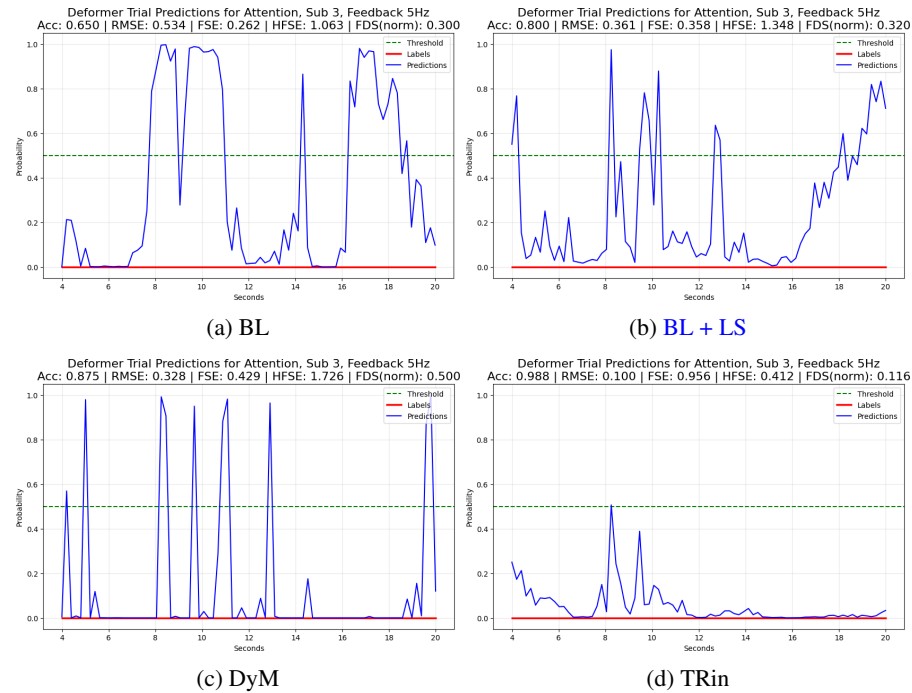

Figure 53: Attention Dataset - Sub3 Trial12 (Deformer Model)

