# OpenReview forum: "Toward Stable Brain-Computer Interfaces: Revealing and Addressing Prediction Fluctuations in EEG-Based BCIs"
_ICLR.cc/2026/Conference — Submitted to ICLR 2026_

### Official Review · Reviewer_nK9J · 2025-10-30

**Soundness:** 3
**Presentation:** 3
**Contribution:** 3
**Rating:** 6
**Confidence:** 4

**Summary:**

This study focuses on the temporal instability of predictions in EEG-based BCIs, where consecutive outputs fluctuate despite high accuracy. It systematically characterizes prediction instability across datasets and models, proposes two new temporal-stability metrics (FDS and FSE), and introduces a new training framework (TRin) that integrates temporal regularization with a dynamic training strategy. Experimental results show that TRin effectively reduces prediction fluctuations and improves classification performance across multiple EEG datasets and architectures.

**Strengths:**

1. The paper squarely targets temporal instability of EEG-BCI predictions—an issue that is rarely foregrounded yet highly consequential for online, closed-loop use. By reframing “prediction stability” as a first-class objective (rather than a post-hoc smoothing concern), the work opens a new axis for evaluating and improving BCI systems.

2. The authors introduce interpretable, quantitative stability metrics that disentangle accuracy from temporal jitter, and they propose a principled training scheme to directly improve stability at the model level.

3. The objectives, metrics, and loss terms are clearly specified, with intuitive explanations and well-chosen ablations. A rigorous setup—cross-subject evaluation, consistent window/stride, multiple architectures, and statistical testing—makes the reported gains credible.

**Weaknesses:**

**High overlap setting.** The temporal overlap is very large (~95% overlap), which induces strong sample correlation, magnifies the apparent advantage of any smoothing-leaning method, and can overstate statistical significance. Part of the reported gains may therefore stem from resampling/overlap rather than the proposed objective.

**Narrow base model coverage.**
The current baselines skew toward classic or mid-capacity models, leaving open whether improvements hold with stronger backbones and standard temporal-consistency approaches. To strengthen the evidence, integrate the regularizer into more advanced architectures—for example, EEG foundation models—and report results under a matched latency budget.

**Questions:**

1. Real-time BCIs need both stable outputs and quick response. Temporal regularization or heavy overlap can add delay and blur change points. This may improve FSE/FDS on slow-changing labels but cause the model to lag or suppress too much at sudden changes, hurting control in closed-loop use. Could you include a step-change experiment to quantify responsiveness under abrupt changes?

2. Without comparisons to standard consistency and simple smoothing methods, it’s hard to tell whether the gains come from your method or from generic smoothing. Could you provide matched-latency, equal-tuning comparisons against standard consistency/smoothing methods (e.g., Temporal Ensembling).

---

> ### Author Response · Authors · 2025-11-21
>
> We thank the reviewer for the positive assessment of our paper! We have uploaded a supplemented paper with changes marked in blue.
>
> ---
> **Q1: [95% overlap overstate temporal stability]** "High overlap setting ... the reported gains may therefore stem from resampling/overlap rather than the proposed objective."
>
> **A1:** Thank you for your critical thoughts. The high overlap (95%) between consecutive windows is to ensure real-time neurofeedback feasibility by 200ms delay [Teo et al., 2021; Jochumsen et al., 2019]. It also raises concerns about whether fluctuation issues are amplified. To address this, we add a new experiment (Appendix J) to evaluate TRin's performance across varying overlaps (95%, 75%, 50%).
>
> The result shows that TRin maintains consistent performance gains across all overlap conditions (Table A), demonstrating that its regularization transcends input overlap. Improvements persist even when segment overlap are reduced, confirming robustness.
>
> **Table A:** Performance comparison on Mental Workload dataset with different overlap rate. Results show mean ± standard deviation over cross-validation folds. The best performance is highlighted in **bold**.
>
> | Overlap | Model | ACC(↑) | F1-macro(↑) | RMSE(↓) | PCC(↑) | FSE(↑) | HFSE(↓) |
> |---------|-------|--------|--------------|---------|--------|--------|---------|
> | 95% | EEGNet | 0.7111±0.1612 | 0.6788±0.1984 | 0.4774±0.1679 | 0.4862±0.3270 | 0.3964±0.2278 | 1.0126±0.4664 |
> | 95% | EEGNet + TRin | **0.7348**±0.1744 | **0.6871**±0.2336 | **0.4248**±0.1713 | **0.5865**±0.3715 | **0.6111**±0.1573 | **0.3536**±0.1377 |
> | 95% | TSception | 0.6798±0.1640 | 0.6306±0.2102 | 0.5121±0.1762 | 0.4400±0.3270 | 0.6071±0.1804 | 0.3679±0.2098 |
> | 95% | TSception + TRin | **0.7390**±0.1724 | **0.7106**±0.2097 | **0.4295**±0.1729 | **0.5598**±0.3725 | **0.6652**±0.1943 | **0.3532**±0.1484 |
> | 95% | Deformer | 0.7247±0.1602 | 0.6923±0.1993 | 0.4680±0.1718 | 0.5101±0.3194 | 0.4301±0.2216 | 1.4305±0.5940 |
> | 95% | Deformer + TRin | **0.7645**±0.1539 | **0.7326**±0.2031 | **0.3983**±0.1508 | **0.6438**±0.3090 | **0.6529**±0.1568 | **0.4064**±0.1415 |
> | 75% | EEGNet | 0.7063±0.1630 | 0.6740±0.1995 | 0.4801±0.1668 | 0.4903±0.3221 | 0.5033±0.2301 | 0.9360±0.5120 |
> | 75% | EEGNet + TRin | **0.7359**±0.1767 | **0.6885**±0.2353 | **0.4257**±0.1701 | **0.5922**±0.3672 | **0.6420**±0.1598 | **0.3476**±0.1557 |
> | 75% | TSception | 0.6793±0.1620 | 0.6297±0.2092 | 0.5108±0.1774 | 0.4439±0.3235 | 0.5567±0.1823 | 0.5689±0.3380 |
> | 75% | TSception + TRin | **0.7403**±0.1729 | **0.7117**±0.2110 | **0.4298**±0.1721 | **0.5587**±0.3749 | **0.6215**±0.1965 | **0.5046**±0.2363 |
> | 75% | Deformer | 0.7165±0.1627 | 0.6829±0.2013 | 0.4725±0.1770 | 0.4959±0.3272 | 0.4918±0.2275 | 1.0457±0.4888 |
> | 75% | Deformer + TRin | **0.7614**±0.1525 | **0.7301**±0.2000 | **0.4009**±0.1498 | **0.6425**±0.2992 | **0.6603**±0.1592 | **0.3853**±0.1617 |
> | 50% | EEGNet | 0.7068±0.1674 | 0.6735±0.2036 | 0.4812±0.1654 | 0.4917±0.3267 | 0.5192±0.1962 | 0.8548±0.4273 |
> | 50% | EEGNet + TRin | **0.7386**±0.1777 | **0.6924**±0.2356 | **0.4250**±0.1709 | **0.5903**±0.3749 | **0.6284**±0.1617 | **0.4008**±0.1940 |
> | 50% | TSception | 0.6815±0.1667 | 0.6319±0.2140 | 0.5073±0.1849 | 0.4473±0.3339 | 0.5422±0.2056 | 0.6111±0.4207 |
> | 50% | TSception + TRin | **0.7396**±0.1768 | **0.7114**±0.2142 | **0.4300**±0.1721 | **0.5522**±0.3810 | **0.6101**±0.1957 | **0.5697**±0.2789 |
> | 50% | Deformer | 0.7183±0.1614 | 0.6841±0.1995 | 0.4673±0.1781 | 0.5074±0.3189 | 0.5167±0.2381 | 0.9761±0.5158 |
> | 50% | Deformer + TRin | **0.7619**±0.1548 | **0.7293**±0.2043 | **0.3996**±0.1508 | **0.6402**±0.3149 | **0.6527**±0.1626 | **0.4455**±0.2102 |
>
> Importantly, both TRin and baseline models operate independently per segment during inference, so reducing overlap is functionally equivalent to downsampling predictions, which increases latency. Belinskaia et al. (2020) suggest that delayed neurofeedback impedes training. Thus, any observed performance improvement at lower overlap rates should not be interpreted as enhanced real-time effectiveness but as a sanity check to verify our TRin's consistency.
>
> ---
> ### References:
>
> - S.-H. J. Teo, X. W. W. Poh, T. S. Lee, C. Guan, Y. B. Cheung, D. S. S. Fung, H. H. Zhang, Z. Y. Chin, C. C. Wang, M. Sung, T. J. Goh, S. J. Weng, X. J. J. Tng, and C. G. Lim. Brain-computer interface based attention and social cognition training programme for children with ASD and co-occurring ADHD: A feasibility trial. *Research in Autism Spectrum Disorders*, 89:101882, 2021.
>
> - M. Jochumsen, M. S. Navid, R. W. Nedergaard, N. Signal, U. Rashid, A. Hassan, H. Haavik, D. Taylor, and I. K. Niazi. Self-paced online vs. cue-based offline brain–computer interfaces for inducing neural plasticity. *Brain Sciences*, 9(6):127, 2019.
>
> - A. Belinskaia, N. Smetanin, M. Lebedev, and A. Ossadtchi. Short-delay neurofeedback facilitates training of the parietal alpha rhythm. *Journal of Neural Engineering*, 17(6):066012, 2020.

---

> ### Author Response · Authors · 2025-11-21
>
> ---
> **Q2: [Model latency analysis]** "Real-time BCIs need both stable outputs and quick response ... include a step-change experiment to quantify responsiveness under abrupt changes?"
>
> **A2:** Thank you for the great question. We apologize for any prior ambiguity and emphasize: TRin introduces no additional computational overhead or temporal dependencies during real-time inference. Unlike moving-average methods or recurrent frameworks, TRin operates identically to conventional models at inference time. It processes each input segment independently without reliance on historical predictions or prior signal segments. Also, TRin introduces no extra runtime cost compared to baseline architectures while still achieving regularization benefits. This key advantage of TRin has been explained in a new section 2.3 for clarity. Thank you.
>
> We add a supplement simulation experiment in Appendix I. The result (Table B) demonstrates that TRin maintains inference speeds comparable to baseline models, consistent with our theoretical expectations. Notably, while TRin preserves real-time efficiency, conventional post hoc sliding window methods introduce additional latency without improving classification accuracy. This highlights TRin's practical advantages for deployment.
>
> For visualization or more details, please refer to Appendix I in the supplemented paper.
>
> **Table B:** Latency analysis with change in ground truth on Mental Workload dataset. Results show mean ± standard deviation over cross-validation folds. The best performance is highlighted in **bold** while the second best performance is highlighted in *italics*. SW represents sliding window post-processing, TRin represents our proposed framework, and TRinS represents TRin with post-processing smoothing. Lt refers to latency and UT refers to the number of unsuccessful transitions.
>
> | Model | ACC(↑) | F1-macro(↑) | RMSE(↓) | FSE(↑) | HFSE(↓) | FDS_norm(↓) | Lt(↓) | UT(↓) |
> |-------|--------|--------------|---------|--------|---------|-------------|-------|-------|
> | EEGNet | 0.7091±0.1584 | 0.6773±0.1951 | 0.4811±0.1625 | 0.1837±0.2179 | 2.4239±0.8844 | 2.3286±0.8305 | *0.8457*±0.6848 | **0.1111**±0.3143 |
> | EEGNet + SW | 0.7167±0.1638 | 0.6834±0.2026 | 0.4555±0.1685 | 0.6147±0.2553 | 1.0076±0.2252 | 0.9627±0.2695 | 1.0055±0.8142 | 0.1944±0.3958 |
> | EEGNet + TRin | *0.7327*±0.1701 | *0.6866*±0.2270 | *0.4270*±0.1663 | *0.6257*±0.2596 | *0.9739*±0.1439 | *0.8757*±0.2165 | **0.7736**±0.6787 | **0.1111**±0.3143 |
> | EEGNet + TRinS | **0.7369**±0.1747 | **0.6897**±0.2329 | **0.4232**±0.1683 | **0.7158**±0.2482 | **0.6831**±0.0617 | **0.5157**±0.0775 | 0.9204±0.8783 | *0.1389*±0.3458 |
> | TSception | 0.6771±0.1607 | 0.6288±0.2058 | 0.5166±0.1680 | 0.5545±0.3682 | 0.9376±0.1940 | 0.8842±0.2638 | **0.6455**±0.8027 | *0.1667*±0.3727 |
> | TSception + SW | 0.6777±0.1612 | 0.6293±0.2062 | 0.5117±0.1708 | 0.6116±0.2623 | *0.7391*±0.0796 | *0.6538*±0.1478 | 0.8084±0.8415 | 0.1667±0.3727 |
> | TSception + TRin | **0.7367**±0.1698 | **0.7084**±0.2071 | *0.4327*±0.1700 | *0.6897*±0.2177 | 0.8531±0.1125 | 0.7815±0.1643 | *0.7596*±0.7819 | **0.1389**±0.3458 |
> | TSception + TRinS | *0.7369*±0.1747 | *0.6897*±0.2329 | **0.4232**±0.1683 | **0.7158**±0.2482 | **0.6831**±0.0617 | **0.5157**±0.0775 | 0.9187±0.9565 | **0.1389**±0.3458 |
> | Deformer | 0.7228±0.1572 | 0.6910±0.1956 | 0.4721±0.1663 | 0.1994±0.2123 | 2.4212±0.8142 | 2.3758±0.8505 | *0.7736*±0.6193 | *0.1111*±0.3143 |
> | Deformer + SW | 0.7299±0.1633 | 0.6967±0.2037 | 0.4438±0.1728 | 0.6263±0.2682 | 0.9803±0.2115 | 0.9633±0.2702 | 0.9441±0.7902 | *0.1111*±0.3143 |
> | Deformer + TRin | *0.7620*±0.1504 | *0.7311*±0.1978 | *0.4018*±0.1460 | *0.6948*±0.2231 | *0.9164*±0.1500 | *0.7961*±0.1737 | **0.7374**±0.8119 | **0.0833**±0.2764 |
> | Deformer + TRinS | **0.7626**±0.1519 | **0.7315**±0.1993 | **0.3984**±0.1474 | **0.7652**±0.2191 | **0.6683**±0.0840 | **0.5065**±0.0749 | 0.9047±0.9916 | *0.1111*±0.3143 |
> | CBraMod | 0.6730±0.1592 | 0.6214±0.2087 | 0.5329±0.1748 | 0.0609±0.1803 | 3.5570±1.2185 | 3.0957±1.2356 | **0.6158**±0.6967 | *0.1389*±0.3458 |
> | CBraMod + SW | 0.6795±0.1668 | 0.6232±0.2200 | 0.4948±0.1870 | 0.3837±0.3400 | 1.1396±0.3929 | 1.1396±0.3929 | 0.8182±0.8824 | 0.1667±0.3727 |
> | CBraMod + TRin | *0.7096*±0.1667 | *0.6602*±0.2148 | *0.4441*±0.1292 | *0.5256*±0.2589 | *1.2520*±0.3902 | *1.0022*±0.3408 | *0.7605*±0.7012 | **0.1111**±0.3143 |
> | CBraMod + TRinS | **0.7149**±0.1681 | **0.6646**±0.2239 | **0.4403**±0.1305 | **0.7098**±0.2152 | **0.9745**±0.9905 | **0.5317**±0.0900 | 0.9745±0.9905 | *0.1389*±0.3458 |

---

> ### Author Response · Authors · 2025-11-21
>
> ---
> **Q3: [Include foundation models as baseline]** "Narrow base model coverage ... EEG foundation models—and report results under a matched latency budget."
>
> **A3:** Thank you for your constructive suggestions. We incorporate CbraMod, a newest EEG foundation model, as a baseline model across all main quantitative experiment analyses. Full results are not listed here due to space limitation, please kindly refer to Tables 1/2/4 in the supplemented paper. These evaluations further demonstrate our proposed TRin framework consistently achieves performance improvements regardless of model architecture scale or complexity. Regarding latency budget, although CbraMod is 10 times larger than regular models such as Deformer, it's computational latency is still negligible compared to latency between sampling of EEG segments (e.g. 200 ms).
>
> ---
> **Q4: [Comparisons to simple smoothing methods]** "Without comparisons to standard consistency and simple smoothing methods ... standard consistency/smoothing methods (e.g., Temporal Ensembling)."
>
> **A4:** Thank you for this excellent comment. Regarding the suggestion to use temporal ensembling, we find the approach somewhat misaligned with our problem setting. Temporal ensembling primarily operates over the training timeline, focusing on stabilizing predictions across epochs, rather than on aligning sequential signal representations within each input sequence. To further examine alternative consistency strategies, we evaluated several standard methods, including (1) cross-entropy (CE) with a temporal smoothing penalty, (2) CE with label smoothing, and (3) post-hoc moving average smoothing.
>
> 1. **CE + temporal smoothing penalty:** We did a supplementary experiment using CE + temporal smoothing penalty and common practices where entire batches remain sequential [Dileo et al., 2023] without TRin's clip-wise shuffling strategy. For detailed results, please refer to Table C or Appendix H.2. The result shows that BL-TL (CE + temporal smoothing penalty only) achieves chance-level performance across all models. A possible reason is that EEG datasets typically have limited trial duration, often resulting in only one batch being sampled per trial. Without proper shuffling (TRin strategy), models tend to overfit the sequence of the data rather than learning generalizable patterns. This underscores the critical role of TRin's clip-wise shuffling, which ensures robust feature learning.
>
> **Table C:** CE + temporal smoothing penalty without clip-wise shuffling strategy on Mental Workload dataset across models. Results show mean ± standard deviation over cross-validation folds. The most optimal value is highlighted in **bold** and secondary optimal value is highlighted in *italics*. BL-TL represents CE + temporal smoothing penalty without TRin's clip-wise shuffling strategy.
>
> | Model | Config | ACC(↑) | F1-macro(↑) | RMSE(↓) | PCC(↑) | FSE(↑) | FDS_norm(↓) |
> |-------|--------|--------|--------------|---------|--------|--------|-------------|
> | EEGNet | BL-CE | *0.7111*±0.1612 | *0.6788*±0.1984 | *0.4774*±0.1679 | *0.4862*±0.3270 | *0.3964*±0.2278 | 1.0126±0.4664 |
> | EEGNet | BL-TL | 0.4988±0.0278 | 0.4894±0.0324 | 0.5297±0.0415 | 0.0027±0.0800 | 0.1576±0.1044 | *0.8915*±0.6609 |
> | EEGNet | TRin | **0.7348**±0.1744 | **0.6871**±0.2336 | **0.4248**±0.1713 | **0.5865**±0.3715 | **0.6111**±0.1573 | **0.3536**±0.1377 |
> | TSception | BL-CE | *0.6798*±0.1640 | *0.6306*±0.2102 | *0.5121*±0.1762 | *0.4400*±0.3270 | *0.6071*±0.1804 | 0.3311±0.1708 |
> | TSception | BL-TL | 0.5356±0.1133 | 0.4897±0.1327 | 0.5596±0.1131 | 0.1580±0.3268 | 0.4128±0.1621 | *0.3293*±0.2863 |
> | TSception | TRin | **0.7390**±0.1724 | **0.7106**±0.2097 | **0.4295**±0.1729 | **0.5598**±0.3725 | **0.6652**±0.1943 | **0.2927**±0.1130 |
> | Deformer | BL-CE | *0.7247*±0.1602 | *0.6923*±0.1993 | *0.4680*±0.1718 | *0.5101*±0.3194 | *0.4301*±0.2216 | 1.0448±0.4584 |
> | Deformer | BL-TL | 0.5024±0.1472 | 0.4482±0.1577 | 0.5154±0.0496 | 0.0237±0.4094 | 0.3898±0.1066 | *0.3256*±0.1161 |
> | Deformer | TRin | **0.7645**±0.1539 | **0.7326**±0.2031 | **0.3983**±0.1508 | **0.6438**±0.3090 | **0.6529**±0.1568 | **0.3075**±0.1065 |
>
>
> ---
> ### References:
>
> - M. Dileo, P. Minervini, M. Zignani, and S. Gaito. Temporal smoothness regularisers for neural link predictors. In *Temporal Graph Learning Workshop at NeurIPS*, New Orleans, USA, 2023.

---

> > ### Author Response · Authors · 2025-11-21
> >
> > (continued from the previous response **A4**)
> >
> >
> > 2. **CE + label smoothing:** We have incorporated label smoothing results in quantitative analyses across datasets. Full results are not listed here due to space limitation, please kindly refer to Tables 1/2/4 in the supplemented paper. Our experiments reveal that while label smoothing serves as a simple yet effective temporal regularizer, it demonstrates less consistent performance than TRin across architectures, with TSception being the only exception.
> >
> >     Notably, label smoothing variants show only marginal classification accuracy improvements, even less than MSE loss variants on SEED. This suggests that although effective for temporal regularization, label smoothing may not sufficiently guide models to learn meaningful or generalizable feature representations to benefit classification performance. At the same time, Visualization in Figure 4 and Appendix N shows that label smoothing variant is an effective temporal regularization, but still less capable of suppressing high frequency fluctuations than TRin.
> >
> > 3. **Post Moving Average:** Our latency analysis (Table B) demonstrates TRin's superiority in accuracy, stability, and latency. Notably, TRin improves both classification accuracy and temporal stability. In contrast, methods such as label smoothing or post-hoc smoothing enhance stability without meaningful accuracy gains. This distinction indicates that simple smoothing does not inherently improve classification. Instead, TRin’s dynamic regularization encourages discriminative feature learning, leading to temporally stable yet correct predictions.

---

### Official Review · Reviewer_ZHwA · 2025-10-31

**Soundness:** 2
**Presentation:** 3
**Contribution:** 2
**Rating:** 4
**Confidence:** 4

**Summary:**

This paper proposes TRin (Temporal Regularization integrated network), a training framework for EEG-based BCIs that aims to improve the temporal stability of decoding outputs. TRin combines a temporal regularization loss and a dynamic curriculum to enforce smooth and consistent predictions over time.
New stability metrics (FSE and FDS) are introduced to quantify prediction consistency, and experiments on multiple EEG datasets show improved decoding accuracy and stability over baselines.

**Strengths:**

- This paper is well-written and intuitively illustrated.
- Experiments across multiple datasets show consistent gains in both accuracy and stability, highlighting the practical potential of the proposed framework.
- The introduction of the FSE and FDS metrics provides a quantitative extension to prior evaluation practices, enabling more systematic analysis of prediction stability in EEG decoding.

**Weaknesses:**

- The ablation study lacks key variants such as temporal loss only or TRin without curriculum, making it unclear which component drives the observed improvement. Moreover, the number of evaluated models is limited, reducing confidence in the generality of the results. Clarification through fine-grained ablation or sensitivity analysis is needed.
- The datasets used in this study have trial-level constant labels that do not capture within-trial dynamics. As a result, variations in model outputs could reflect unlabeled but meaningful state transitions, rather than noise. This mismatch between label granularity and the definition of “stability” weakens the conceptual foundation of the proposed framework.
- Figure 1 illustrates how unstable predictions could lead to inconsistent feedback, but this remains only a conceptual example without empirical validation. No experiments evaluate TRin’s impact on real-time inference latency, adaptation speed, or feedback consistency.

**Questions:**

- Q1. The paper claims that prediction fluctuation undermines user engagement and neurofeedback performance, but provides no supporting evidence or references. Could the authors cite prior work or empirical findings that demonstrate this effect?
- Q2. Could “smooth-but-wrong” predictions be over-rewarded under constant-label conditions, since the stability metrics penalize fluctuation rather than correctness?
- Q3. Were the results and visualizations obtained from all subjects or only selected examples? If only a subset was shown, how consistent are these patterns across subjects?

---

> ### Author Response · Authors · 2025-11-21
>
> Thank you for your critical feedback! We have uploaded a supplemented paper with changes marked in blue.
>
> ---
> **Q1: [Ablation study lacks temporal loss only and TRin without curriculum]** "The ablation study lacks key variants such as temporal loss only or TRin without curriculum ... is needed."
>
> **A1:** Thank you for the suggestion. We did a supplementary experiment using CE + temporal loss and common practices where entire batches remain sequential [Dileo et al., 2023] without TRin's clip-wise shuffling strategy. For detailed results, please refer to Table A or Appendix H.2. The result shows that BL-TL (CE + temporal smoothing penalty only) achieves chance-level performance across all models. A possible reason is that EEG datasets typically have limited trial duration, often resulting in only one batch being sampled per trial. Without proper shuffling (TRin strategy), models tend to overfit the sequence of the data rather than learning generalizable patterns. This underscores the critical role of TRin's clip-wise shuffling, which ensures robust feature learning.
>
>
> **Table A:** Ablation study of temporal regularization loss function alone on Mental Workload dataset across models. Results show mean ± standard deviation over cross-validation folds. The most optimal value is highlighted in **bold** and secondary optimal value is highlighted in *italics*. BL-TL represents CE + temporal smoothing penalty without TRin's clip-wise shuffling strategy.
>
> | Model | Config | ACC(↑) | F1-macro(↑) | RMSE(↓) | PCC(↑) | FSE(↑) | FDS_norm(↓) |
> |-------|--------|--------|--------------|---------|--------|--------|-------------|
> | EEGNet | BL-CE | *0.7111*±0.1612 | *0.6788*±0.1984 | *0.4774*±0.1679 | *0.4862*±0.3270 | *0.3964*±0.2278 | 1.0126±0.4664 |
> | EEGNet | BL-TL | 0.4988±0.0278 | 0.4894±0.0324 | 0.5297±0.0415 | 0.0027±0.0800 | 0.1576±0.1044 | *0.8915*±0.6609 |
> | EEGNet | TRin | **0.7348**±0.1744 | **0.6871**±0.2336 | **0.4248**±0.1713 | **0.5865**±0.3715 | **0.6111**±0.1573 | **0.3536**±0.1377 |
> | TSception | BL-CE | *0.6798*±0.1640 | *0.6306*±0.2102 | *0.5121*±0.1762 | *0.4400*±0.3270 | *0.6071*±0.1804 | 0.3311±0.1708 |
> | TSception | BL-TL | 0.5356±0.1133 | 0.4897±0.1327 | 0.5596±0.1131 | 0.1580±0.3268 | 0.4128±0.1621 | *0.3293*±0.2863 |
> | TSception | TRin | **0.7390**±0.1724 | **0.7106**±0.2097 | **0.4295**±0.1729 | **0.5598**±0.3725 | **0.6652**±0.1943 | **0.2927**±0.1130 |
> | Deformer | BL-CE | *0.7247*±0.1602 | *0.6923*±0.1993 | *0.4680*±0.1718 | *0.5101*±0.3194 | *0.4301*±0.2216 | 1.0448±0.4584 |
> | Deformer | BL-TL | 0.5024±0.1472 | 0.4482±0.1577 | 0.5154±0.0496 | 0.0237±0.4094 | 0.3898±0.1066 | *0.3256*±0.1161 |
> | Deformer | TRin | **0.7645**±0.1539 | **0.7326**±0.2031 | **0.3983**±0.1508 | **0.6438**±0.3090 | **0.6529**±0.1568 | **0.3075**±0.1065 |
>
> For curriculum learning, we apologize for any prior ambiguity in terminology: dynamic training and curriculum learning refer to the same strategy. Our original ablation study (Appendix H.1) on dynamic training implemented curriculum learning. This connection has now been explicitly highlighted in the revised manuscript for clarity.
>
> ---
> **Q2: [Number of evaluated models is limited]** "Moreover, the number of evaluated models is limited, reducing confidence in the generality of the results."
>
> **A2:** We understand your concerns and incorporate CbraMod, a newest EEG foundation model, as a baseline model across all main quantitative experiment analyses. Full results are not listed here due to space limitation, please kindly refer to Tables 1/2/4 in the supplemented paper. These evaluations further demonstrate our proposed TRin framework consistently achieves performance improvements regardless of model architecture scale or complexity.
>
> ---
> ### References:
>
> - M. Dileo, P. Minervini, M. Zignani, and S. Gaito. Temporal smoothness regularisers for neural link predictors. In *Temporal Graph Learning Workshop at NeurIPS*, New Orleans, USA, 2023.

---

> ### Author Response · Authors · 2025-11-21
>
> ---
> **Q3: [Model outputs after regularization may remove unlabeled but meaningful state transitions]** "The datasets used in this study have trial-level constant labels ... the definition of "stability" weakens the conceptual foundation of the proposed framework."
>
> **A3:** Thank you for your critical thoughts. As discussed in the introduction, prior foundational work has established that emotions persist from seconds to hours [Verduyn et al., 2015], while cognitive attentional mechanisms exhibit slow-changing dynamics [Seeburger et al., 2024] and periodically unfold about 14 seconds [Kasten et al., 2024]. Critically, these works establish that mental states are less likely to exhibit abrupt, high-frequency transitions at millisecond timescales. This serves as a principle central to our framework: focus on unrealistic high-frequency fluctuations.
>
> Our evaluation metrics (FDS and FSE) and learning objective (Temporal Regularization loss) were specifically designed to penalize unrealistic high-frequency fluctuations aggressively, while not penalizing explicitly regarding possible meaningful state transitions with low-frequency. Specifically, FDS and Temporal Regularization loss exclusively targets consecutive timestep variations (highest possible frequency); FSE employs the weight function $−ln(1 − f)$ to exponentially amplify penalties for higher frequencies while minimally affecting slower, biologically plausible shifts (e.g., second-scale changes). Intuitively, as evidenced in visualization Figure 4 and Appendix N, TRin only relieves unrealistic high-frequency fluctuations while still capable of preserving plausible variability at lower frequencies.
>
> Besides, we wish to further emphasize the rationale behind our framework through quantitative results. State transitions are often studied through EEG frequency band power analysis [Song et al., 2014, Kasten et al., 2024]. This insight motivated us to leverage established EEG features, such as frequency band power. We incorporated traditional SVMs utilizing these band features, serving as a direct rough indicator of task-related fluctuation range. By comparing performance fluctuations against the SVM baseline, we establish a sanity check. One can determine whether there are neurophysiological violations by checking deviations with SVM baseline. Our evaluations reveal that baseline deep learning models exhibit ~200% greater high-frequency unrealistic fluctuations (measured by HFSE and FDS) compared to the SVM benchmark. Such significant high-frequency instability highlights the necessity of TRin's regularization.
>
> ---
>
> ### References:
>
> - P. Verduyn, P. Delaveau, J.-Y. Rotgé, P. Fossati, and I. Van Mechelen. Determinants of emotion duration and underlying psychological and neural mechanisms. *Emotion Review*, 7(4):330–335, 2015. doi: 10.1177/1754073915590618.
>
> - D. T. Seeburger, N. Xu, M. Ma, S. Larson, C. Godwin, S. D. Keilholz, and E. H. Schumacher. Time-varying functional connectivity predicts fluctuations in sustained attention in a serial tapping task. *Cognitive, Affective, & Behavioral Neuroscience*, 2024. doi: 10.3758/s13415-024-01156-1.
>
> - F. H. Kasten, Q. Busson, and B. Zoefel. Opposing neural processing modes alternate rhythmically during sustained auditory attention. *Communications Biology*, 7(1):1125, 2024. doi: 10.1038/s42003-024-06834-x.
>
> - K. Song, M. Meng, L. Chen, K. Zhou, and H. Luo. Behavioral Oscillations in Attention: Rhythmic α Pulses Mediated through θ Band. *Journal of Neuroscience*, 34(14):4837–4844, 2014. doi: 10.1523/JNEUROSCI.4856-13.2014.

---

> ### Author Response · Authors · 2025-11-21
>
> ---
> **Q4: [Support linking prediction fluctuation undermines neurofeedback performance]** "Figure 1 illustrates how unstable predictions could lead to inconsistent feedback ... could the authors cite prior work or empirical findings that demonstrate this effect?"
>
> **A4:** Thank you for your careful question. To the best of our knowledge, this work is the first to identify and systematically address temporal fluctuation in DL-based BCI decoders. Previous clinical studies have rarely discussed this issue, likely because most employed traditional machine learning models such as SVMs, which do not inherently exhibit prediction fluctuation. Consequently, the problem of prediction instability has been largely overlooked.
>
> Nevertheless, several lines of indirect evidence suggest that such fluctuations may have critical implications.
> 1. Literature emphasizes that the effectiveness of neural feedback is reduced by biased or sham feedback [Kober et al. 2018, Pigott et al. 2021, Sitaram et al. 2017, Mladenovic et al. 2021]. As illustrated in Figure 1, current DL decoders exhibit high-frequency oscillations, revealing inconsistency with actual brain states. This inconsistency plausibly undermines feedback reliability and so does user learning effect.
>
> 2. Besides, feedback transparency affects users' sense of agency so does neural feedback performance [Dussard et al. 2021]. Feedback transparency refers to the intuitive connection between the feedback signal and the task performed by the user. High-frequency fluctuations in feedback signals would decrease feedback transparency. Therefore, it's plausible that prediction fluctuation undermines users' sense of agency and neurofeedback performance.
>
> 3. Furthermore, the frequent use of post-hoc smoothing procedures in DL-based decoders is often implemented solely to reduce fluctuations, despite their large latency that impedes training [Belinskaia et al. 2020]. This highlights the practical necessity of temporal regularization outweighs other drawbacks from post-hoc. Our proposed framework addresses this gap directly. When evaluated against post-hoc label smoothing (Appendix I or Table A in A5), TRin demonstrated superior accuracy, temporal stability, and reduced latency, collectively affirming its practical advantage.
>
> We acknowledge the limitation that direct causal evidence linking prediction fluctuation to impaired neurofeedback performance is still lacking. However, rather than weakening our contribution, we believe our findings establish a new research direction, providing both the conceptual and methodological foundation enabling future neuroscientific investigations into the role of temporal stability.
>
> ---
> ### References:
>
> - S. E. Kober, M. Witte, S. Grinschgl, C. Neuper, and G. Wood. Placebo hampers ability to self-regulate brain activity: A double-blind sham-controlled neurofeedback study. *NeuroImage*, 181, 797–806 (2018). doi: 10.1016/j.neuroimage.2018.07.025
>
> - H. E. Pigott, R. Cannon, and M. Trullinger. The fallacy of sham-controlled neurofeedback trials: A reply to Thibault and colleagues (2018). *Journal of Attention Disorders*, 25(3), 448–457 (2021). doi: 10.1177/1087054718790802
>
> - R. Sitaram, T. Ros, L. Stoeckel, S. Haller, F. Scharnowski, J. Lewis-Peacock, N. Weiskopf, M. L. Blefari, M. Rana, E. Oblak, N. Birbaumer, and J. Sulzer. Closed-loop brain training: the science of neurofeedback. *Nature Reviews Neuroscience* 18, 86–100 (2017). doi: 10.1038/nrn.2016.164
>
> - J. Mladenović, J. Frey, S. Pramij, J. Mattout, and F. Lotte. Towards identifying optimal biased feedback for various user states and traits in motor imagery BCI. *IEEE Transactions on Biomedical Engineering*, 2021.
>
> - C. Dussard, L. Pillette, C. Dumas, E. Pierrieau, L. Hugueville, B. Lau, C. Jeunet-Kelway, and N. George. Influence of feedback transparency on motor imagery neurofeedback performance: the contribution of agency. *Journal of Neural Engineering*, 21(5):056029, 2021. doi: 10.1088/1741-2552/ad5a31
>
> - A. Belinskaia, N. Smetanin, M. Lebedev, and A. Ossadtchi. Short-delay neurofeedback facilitates training of the parietal alpha rhythm. *Journal of Neural Engineering*, 17(6):066012, 2020. doi: 10.1088/1741-2552/abc8d7.

---

> ### Author Response · Authors · 2025-11-21
>
> ---
> **Q5: [Simulation on real time inference]** "No experiments evaluate TRin's impact on real-time inference latency, adaptation speed, or feedback consistency."
>
> **A5:** Thank you for the great question. We apologize for any prior ambiguity and emphasize: TRin introduces no additional computational overhead or temporal dependencies during real-time inference. Unlike moving-average methods or recurrent frameworks, TRin operates identically to conventional models at inference time. It processes each input segment independently without reliance on historical predictions or prior signal segments. Also, TRin introduces no extra runtime cost compared to baseline architectures while still achieving regularization benefits. This key advantage of TRin has been explained in a new section 2.3 for clarity. Thank you.
>
> We add a supplement simulation experiment in Appendix I. The result (Table B) demonstrates that TRin maintains inference speeds comparable to baseline models, consistent with our theoretical expectations. Notably, while TRin preserves real-time efficiency, conventional post hoc sliding window methods introduce additional latency without improving classification accuracy. This highlights TRin's practical advantages for deployment.
>
> For visualization or more details, please refer to Appendix I in the supplemented paper.
>
> **Table B:** Latency analysis with change in ground truth on Mental Workload dataset. Results show mean ± standard deviation over cross-validation folds. The best performance is highlighted in **bold** while the second best performance is highlighted in *italics*. SW represents sliding window post-processing, TRin represents our proposed framework, and TRinS represents TRin with post-processing smoothing. Lt refers to latency and UT refers to the number of unsuccessful transitions.
>
> | Model | ACC(↑) | F1-macro(↑) | RMSE(↓) | FSE(↑) | HFSE(↓) | FDS_norm(↓) | Lt(↓) | UT(↓) |
> |-------|--------|--------------|---------|--------|---------|-------------|-------|-------|
> | EEGNet | 0.7091±0.1584 | 0.6773±0.1951 | 0.4811±0.1625 | 0.1837±0.2179 | 2.4239±0.8844 | 2.3286±0.8305 | *0.8457*±0.6848 | **0.1111**±0.3143 |
> | EEGNet + SW | 0.7167±0.1638 | 0.6834±0.2026 | 0.4555±0.1685 | 0.6147±0.2553 | 1.0076±0.2252 | 0.9627±0.2695 | 1.0055±0.8142 | 0.1944±0.3958 |
> | EEGNet + TRin | *0.7327*±0.1701 | *0.6866*±0.2270 | *0.4270*±0.1663 | *0.6257*±0.2596 | *0.9739*±0.1439 | *0.8757*±0.2165 | **0.7736**±0.6787 | **0.1111**±0.3143 |
> | EEGNet + TRinS | **0.7369**±0.1747 | **0.6897**±0.2329 | **0.4232**±0.1683 | **0.7158**±0.2482 | **0.6831**±0.0617 | **0.5157**±0.0775 | 0.9204±0.8783 | *0.1389*±0.3458 |
> | TSception | 0.6771±0.1607 | 0.6288±0.2058 | 0.5166±0.1680 | 0.5545±0.3682 | 0.9376±0.1940 | 0.8842±0.2638 | **0.6455**±0.8027 | *0.1667*±0.3727 |
> | TSception + SW | 0.6777±0.1612 | 0.6293±0.2062 | 0.5117±0.1708 | 0.6116±0.2623 | *0.7391*±0.0796 | *0.6538*±0.1478 | 0.8084±0.8415 | 0.1667±0.3727 |
> | TSception + TRin | **0.7367**±0.1698 | **0.7084**±0.2071 | *0.4327*±0.1700 | *0.6897*±0.2177 | 0.8531±0.1125 | 0.7815±0.1643 | *0.7596*±0.7819 | **0.1389**±0.3458 |
> | TSception + TRinS | *0.7369*±0.1747 | *0.6897*±0.2329 | **0.4232**±0.1683 | **0.7158**±0.2482 | **0.6831**±0.0617 | **0.5157**±0.0775 | 0.9187±0.9565 | **0.1389**±0.3458 |
> | Deformer | 0.7228±0.1572 | 0.6910±0.1956 | 0.4721±0.1663 | 0.1994±0.2123 | 2.4212±0.8142 | 2.3758±0.8505 | *0.7736*±0.6193 | *0.1111*±0.3143 |
> | Deformer + SW | 0.7299±0.1633 | 0.6967±0.2037 | 0.4438±0.1728 | 0.6263±0.2682 | 0.9803±0.2115 | 0.9633±0.2702 | 0.9441±0.7902 | *0.1111*±0.3143 |
> | Deformer + TRin | *0.7620*±0.1504 | *0.7311*±0.1978 | *0.4018*±0.1460 | *0.6948*±0.2231 | *0.9164*±0.1500 | *0.7961*±0.1737 | **0.7374**±0.8119 | **0.0833**±0.2764 |
> | Deformer + TRinS | **0.7626**±0.1519 | **0.7315**±0.1993 | **0.3984**±0.1474 | **0.7652**±0.2191 | **0.6683**±0.0840 | **0.5065**±0.0749 | 0.9047±0.9916 | *0.1111*±0.3143 |
> | CBraMod | 0.6730±0.1592 | 0.6214±0.2087 | 0.5329±0.1748 | 0.0609±0.1803 | 3.5570±1.2185 | 3.0957±1.2356 | **0.6158**±0.6967 | *0.1389*±0.3458 |
> | CBraMod + SW | 0.6795±0.1668 | 0.6232±0.2200 | 0.4948±0.1870 | 0.3837±0.3400 | 1.1396±0.3929 | 1.1396±0.3929 | 0.8182±0.8824 | 0.1667±0.3727 |
> | CBraMod + TRin | *0.7096*±0.1667 | *0.6602*±0.2148 | *0.4441*±0.1292 | *0.5256*±0.2589 | *1.2520*±0.3902 | *1.0022*±0.3408 | *0.7605*±0.7012 | **0.1111**±0.3143 |
> | CBraMod + TRinS | **0.7149**±0.1681 | **0.6646**±0.2239 | **0.4403**±0.1305 | **0.7098**±0.2152 | **0.9745**±0.9905 | **0.5317**±0.0900 | 0.9745±0.9905 | *0.1389*±0.3458 |

---

> ### Author Response · Authors · 2025-11-21
>
> ---
> **Q6: [Smooth-but-wrong predictions be over-rewarded]** "Could "smooth-but-wrong" predictions be over-rewarded under constant-label conditions, since the stability metrics penalize fluctuation rather than correctness?"
>
> **A6:** Sorry for any misunderstanding to this question, since we cannot determine whether you are referring to training time or inference time. Therefore, we explain all of them separately:
>
> 1. During training, "smooth-but-wrong" predictions are unlikely to be over-rewarded. Unlike the supplementary experiment in Appendix H.2, where entire batches consisted of temporally sequential samples from a single class, the TRin training strategy ensures label diversity within each batch. Each batch contains clips from multiple classes, and the total loss is computed as the average across all clips. Consequently, the model must correctly classify each clip to minimize the objective. This setup encourages both temporal smoothness and accurate class discrimination.
>
> 2. Experiment results during inference further confirm that TRin improves both classification accuracy and temporal stability. In contrast, methods such as label smoothing (Tables 1/2/4 in the paper) or post-hoc smoothing (Table A in Q5, or Appendix I in the paper) enhance stability without meaningful accuracy gains. This distinction indicates that simple smoothing does not inherently improve classification. Instead, TRin's dynamic regularization encourages discriminative feature learning, leading to temporally stable yet correct predictions.
>
> 3. If the question pertains purely to the definition of stability metrics, then yes, but only metrics focusing solely on fluctuation (e.g., FDS and HFSE). However, the FSE metric was specifically designed to balance both fluctuation and classification fidelity. Our sensitivity analysis (Appendix G) is based on FSE for precisely this reason. Importantly, inference experiments show that TRin's optimal configuration improves both accuracy and stability. Finally, it is worth noting that while FDS and HFSE are more fluctuation specific, they remain valuable, particularly for isolating and quantifying temporal consistency enabling detailed scrutiny.
>
> ---
> **Q7: [TRin only benefit a subset of subjects]** "Were the results and visualizations obtained from all subjects or only selected examples? If only a subset was shown, how consistent are these patterns across subjects?"
>
> **A7:** We provide a consistency analysis of the TRin framework across subjects. Accuracy, FSE, and FDS metrics are summarized in Figures 6, 7, and 8, respectively in Appendix K. The results demonstrate that TRin consistently improves performance across subjects, confirming its robustness.

---

### Official Review · Reviewer_Cc2y · 2025-11-01

**Soundness:** 3
**Presentation:** 3
**Contribution:** 3
**Rating:** 4
**Confidence:** 3

**Summary:**

This paper identifies prediction fluctuations in deep learning-based EEG brain-computer interfaces as a critical problem that undermines real-time feedback and user engagement. To address the temporal instability, work proposes metrics: Frequency-weighted Spectral Entropy (FSE) and First-Order Difference Standard Deviation (FDS) as quantitative metrics beyond standard accuracy measures. It further proposes a temporal robustness integrated training framework (TRin) combining a temporal regularisation loss with a dynamic training strategy to jointly improve classification accuracy and reduce erratic temporal fluctuations. Empirical validation across 3 paradigms and datasets with different families of model architectures demonstrates consistent performance gains.

**Strengths:**

The paper addresses a fundamental and underexplored aspect of temporal stability, crucial for real-time neurofeedback and clinically relevant applications.

The novel metrics FSE and FDS provide meaningful evaluation of prediction smoothness and instability, overcoming limitations of conventional metrics that degenerate under constant trial labels.

The temporal regularisation loss is theoretically grounded, representing a Gaussian random walk prior, which aligns with the slow and continuous cognitive dynamics suggested by the neuroscience literature.

The dynamic clip-based data shuffling and hyperparameter scheduling effectively preserve temporal dependencies while maintaining generalisation capacity.

Empirical results show consistent accuracy improvements alongside a significant reduction in prediction fluctuations, supporting the utility of the approach.

**Weaknesses:**

Although the temporal regularization is motivated by slow cognitive state changes, the paper does not provide direct neurophysiological validation that enforced stability corresponds to true neural dynamics rather than smoothed noise or artifacts. This gap raises questions about the interpretability of improvements.

The high overlap(95%) between input windows means that consecutive model inputs share most of the data, potentially inflating stability metrics and masking true rapid neural transitions. This challenges whether the regularisation improves actual neural state modelling or merely smooths highly correlated inputs.

Testing with lower overlaps or variable window lengths would clarify how much of the temporal stability gain is due to the loss function versus inherent redundancy.

The risk of suppressing meaningful transient neuromarkers by over-regularizing predictions is noted but not deeply examined or quantitatively analyzed.

**Questions:**

The following are the questions or recommendations that would help me better interpret the work:

Can the authors provide or integrate direct neurophysiological validation linking temporal stability metrics to known neural state dynamics or markers?

Have experiments been conducted with lower or variable sliding window overlaps to disentangle the effects of input overlap from the temporal regularization itself?

What are the trade-offs between temporal regularisation during training and simpler post-hoc label smoothing in terms of accuracy, stability, latency, and computational complexity? Benchmarking would help understand and quantify the contributions.


How does the model’s temporal stability behave during abrupt neural or cognitive state transitions that may challenge the Gaussian random walk prior assumption? Are there paradigms where this could be verified, such as motor imagery, where the neuromarkers are often ERD/ERS?

---

> ### Author Response · Authors · 2025-11-21
>
> Thank you for your very thoughtful questions! We have uploaded a supplemented paper with changes marked in blue.
>
> ---
> **Q1: [Neurophysiological validation linking temporal stability metrics]** "Can the authors provide or integrate direct neurophysiological validation linking temporal stability metrics to known neural state dynamics or markers?"
>
> **A1:** As discussed in the introduction, prior foundational work has established that emotions persist from seconds to hours [Verduyn et al., 2015], while cognitive attentional mechanisms exhibit slow-changing dynamics [Seeburger et al., 2024] and periodically unfold about 14 seconds [Kasten et al., 2024]. Crucially, these studies confirm that mental states are less likely to exhibit high frequency abrupt transitions in milliseconds. This neurophysiological principle directly informs us of our temporal stability metrics. Both FDS and FSE are designed to penalize unrealistic high-frequency fluctuations most aggressively:
>
> 1. FDS exclusively targets consecutive timestep variations (highest possible frequency).
>
> 2. FSE employs the weight function $−ln(1 − f)$ to exponentially amplify penalties for higher frequencies while minimally affecting slower, biologically plausible shifts (e.g., second-scale changes).
>
> This frequency scaled approach ensures our metrics align with known neural dynamics and avoid penalizing meaningful cognitive/affective variability.
>
> While our metrics operationalize current neurophysiological consensus, the exact duration thresholds for mental states remain undetermined. We hope future studies will quantify these temporal boundaries more precisely, enabling further refinement of stability assessments.
>
> ---
> **Q2: [Neurophysiological validation that enforced stability corresponds to true neural dynamics]** "Although the temporal regularization is motivated by slow cognitive state changes ... interpretability of improvements."
>
> **A2:** We appreciate the reviewer's valuable question. As discussed in the introduction and A1, cognitive/affective states evolve relatively slowly and do not exhibit abrupt transitions on the millisecond scale. In particular, state transitions are well studied through EEG frequency band power analysis [Song et al., 2014, Kasten et al., 2024]. This insight motivated us to leverage established EEG features, such as frequency band power. We incorporated traditional SVMs utilizing these band features, serving as a direct rough indicator of task-related fluctuation range. By comparing performance fluctuations against the SVM baseline, we establish a sanity check. One can determine whether there are neurophysiological violations by checking deviations with SVM baseline. Our evaluations reveal that baseline deep learning models exhibit ~200% greater high-frequency unrealistic fluctuations (measured by HFSE and FDS) compared to the SVM benchmark. Such significant high-frequency instability highlights the necessity of TRin's regularization.
>
> Additional indirect evidence also suggests that TRin learns neurophysiologically generalizable features through temporal regularization. Notably, inference tests reveal that TRin improves not only temporal stability but also classification performance. Unlike methods such as label smoothing (Tables 1/2/4) or post-processing (Appendix I), which enhance stability without meaningful accuracy gains. This contrast implies that merely smoothing predictions does not inherently improve classification. Instead, TRin's regularization appears to guide the model toward more generalizable features, indicating that temporal smoothing may inherently refine neurophysiologically discriminative learning rather than just stabilizing outputs.
>
> ---
>
> ### References:
>
> - P. Verduyn, P. Delaveau, J.-Y. Rotgé, P. Fossati, and I. Van Mechelen. Determinants of emotion duration and underlying psychological and neural mechanisms. *Emotion Review*, 7(4):330–335, 2015. doi: 10.1177/1754073915590618.
>
> - D. T. Seeburger, N. Xu, M. Ma, S. Larson, C. Godwin, S. D. Keilholz, and E. H. Schumacher. Time-varying functional connectivity predicts fluctuations in sustained attention in a serial tapping task. *Cognitive, Affective, & Behavioral Neuroscience*, 2024. doi: 10.3758/s13415-024-01156-1.
>
> - F. H. Kasten, Q. Busson, and B. Zoefel. Opposing neural processing modes alternate rhythmically during sustained auditory attention. *Communications Biology*, 7(1):1125, 2024. doi: 10.1038/s42003-024-06834-x.
>
> - K. Song, M. Meng, L. Chen, K. Zhou, and H. Luo. Behavioral Oscillations in Attention: Rhythmic α Pulses Mediated through θ Band. *Journal of Neuroscience*, 34(14):4837–4844, 2014. doi: 10.1523/JNEUROSCI.4856-13.2014.

---

> > ### Author Response · Authors · 2025-11-21
> >
> > ---
> > **Q3: [Potential over smoothing predictions]** "The risk of suppressing meaningful transient neuromarkers by over-regularizing predictions is noted but not deeply examined or quantitatively analyzed."
> >
> > **A3:** In A2, we discussed why TRin's regularization is neurophysiologically aligned and generalizable. Two additional lines of evidence further demonstrate that the optimized TRin framework effectively avoids over smoothing:
> >
> > 1. As shown sensitivity analysis in Appendix G (Tables 7/8), we determine the optimal regularization strength by fine tuning hyperparameters via FSE on the validation set to balance classification and temporal stability. Notably, these gains persist during inference, reinforcing that TRin's regularization is generalizable rather than over restrictive.
> >
> > 2. Consistent with the discussion in the introduction and A1, cognitive/affective states evolve on a second-scale timespan (not milliseconds). Visualizations in Figure 4 and Appendix N confirm TRin preserves this natural variation, while only suppressing unrealistic high-frequency fluctuations. These imply the model effectively aligns with neurophysiological constraints rather than enforcing excessive smoothness.
> >
> > ---
> > **Q4: [Model performance under abrupt state transitions]** "How does the model's temporal stability behave during abrupt neural or cognitive state transitions ... such as motor imagery, where the neuromarkers are often ERD/ERS?"
> >
> > **A4:** Thanks for the great question! Unfortunately, our TRin framework may not be directly applicable to event-based paradigms such as motor imagery, as TRin relies on sustained or long-duration stimuli within each state to achieve effective temporal regularization during training. Moreover, the intrinsic nature of motor imagery task fundamentally differs from that of emotion or attention paradigms. The former typically involves abrupt, discrete events lasting 1-5 seconds [Jeon et al., 2011], whereas the latter can exhibit continuous sustained states and can last for minutes [Zyma et al., 2019; Zheng & Lu, 2015]. Consequently, enforcing temporal consistency in motor imagery tasks might be conceptually inappropriate and offer limited benefit. For these reasons, we did not apply TRin to motor imagery datasets.
> >
> > Instead, we did a supplement simulation experiment in Appendix I. Due to the absence of fine-grained ground truth labels and abrupt mental state transitions in current BCI datasets, we simulate abrupt state transition by concatenating trials from the same subject with differing labels. The result (Table A) demonstrates that TRin maintains inference speeds comparable to baseline models, consistent with our theoretical expectations (clarified in Section 2.3 and A6). Notably, while TRin preserves real-time efficiency, conventional post hoc sliding window methods introduce additional latency without improving classification accuracy. This highlights TRin's practical advantages for deployment.
> >
> > ---
> > ### References:
> >
> > - Y. Jeon, C. S. Nam, Y.-J. Kim, and M. C. Whang. Event-related (De)synchronization (ERD/ERS) during motor imagery tasks: Implications for brainecomputer interfaces. *International Journal of Industrial Ergonomics*, 41:428–436, 2011. doi: 10.1016/j.ergon.2011.03.005.
> >
> > - I. Zyma, S. Tukaev, I. Seleznov, K. Kiyono, A. Popov, M. Chernykh, and O. Shpenkov. Electroencephalograms during Mental Arithmetic Task Performance. *Data*, 4(1):14, 2019. doi: 10.3390/data4010014.
> >
> > - W.-L. Zheng and B.-L. Lu. Investigating Critical Frequency Bands and Channels for EEG-based Emotion Recognition with Deep Neural Networks. *IEEE Transactions on Autonomous Mental Development*, 7(3):162–175, 2015. doi: 10.1109/TAMD.2015.2431497.

---

> > > ### Author Response · Authors · 2025-11-21
> > >
> > > (continued from the previous response **A4**)
> > >
> > > For visualization or more details, please refer to Appendix I in the supplemented paper.
> > >
> > > **Table A:** Latency analysis with change in ground truth on Mental Workload dataset. Results show mean ± standard deviation over cross-validation folds. The best performance is highlighted in **bold** while the second best performance is highlighted in *italics*. SW represents sliding window post-processing, TRin represents our proposed framework, and TRinS represents TRin with post-processing smoothing. Lt refers to latency and UT refers to the number of unsuccessful transitions.
> > >
> > > | Model | ACC(↑) | F1-macro(↑) | RMSE(↓) | FSE(↑) | HFSE(↓) | FDS_norm(↓) | Lt(↓) | UT(↓) |
> > > |-------|--------|--------------|---------|--------|---------|-------------|-------|-------|
> > > | EEGNet | 0.7091±0.1584 | 0.6773±0.1951 | 0.4811±0.1625 | 0.1837±0.2179 | 2.4239±0.8844 | 2.3286±0.8305 | *0.8457*±0.6848 | **0.1111**±0.3143 |
> > > | EEGNet + SW | 0.7167±0.1638 | 0.6834±0.2026 | 0.4555±0.1685 | 0.6147±0.2553 | 1.0076±0.2252 | 0.9627±0.2695 | 1.0055±0.8142 | 0.1944±0.3958 |
> > > | EEGNet + TRin | *0.7327*±0.1701 | *0.6866*±0.2270 | *0.4270*±0.1663 | *0.6257*±0.2596 | *0.9739*±0.1439 | *0.8757*±0.2165 | **0.7736**±0.6787 | **0.1111**±0.3143 |
> > > | EEGNet + TRinS | **0.7369**±0.1747 | **0.6897**±0.2329 | **0.4232**±0.1683 | **0.7158**±0.2482 | **0.6831**±0.0617 | **0.5157**±0.0775 | 0.9204±0.8783 | *0.1389*±0.3458 |
> > > | TSception | 0.6771±0.1607 | 0.6288±0.2058 | 0.5166±0.1680 | 0.5545±0.3682 | 0.9376±0.1940 | 0.8842±0.2638 | **0.6455**±0.8027 | *0.1667*±0.3727 |
> > > | TSception + SW | 0.6777±0.1612 | 0.6293±0.2062 | 0.5117±0.1708 | 0.6116±0.2623 | *0.7391*±0.0796 | *0.6538*±0.1478 | 0.8084±0.8415 | 0.1667±0.3727 |
> > > | TSception + TRin | **0.7367**±0.1698 | **0.7084**±0.2071 | *0.4327*±0.1700 | *0.6897*±0.2177 | 0.8531±0.1125 | 0.7815±0.1643 | *0.7596*±0.7819 | **0.1389**±0.3458 |
> > > | TSception + TRinS | *0.7369*±0.1747 | *0.6897*±0.2329 | **0.4232**±0.1683 | **0.7158**±0.2482 | **0.6831**±0.0617 | **0.5157**±0.0775 | 0.9187±0.9565 | **0.1389**±0.3458 |
> > > | Deformer | 0.7228±0.1572 | 0.6910±0.1956 | 0.4721±0.1663 | 0.1994±0.2123 | 2.4212±0.8142 | 2.3758±0.8505 | *0.7736*±0.6193 | *0.1111*±0.3143 |
> > > | Deformer + SW | 0.7299±0.1633 | 0.6967±0.2037 | 0.4438±0.1728 | 0.6263±0.2682 | 0.9803±0.2115 | 0.9633±0.2702 | 0.9441±0.7902 | *0.1111*±0.3143 |
> > > | Deformer + TRin | *0.7620*±0.1504 | *0.7311*±0.1978 | *0.4018*±0.1460 | *0.6948*±0.2231 | *0.9164*±0.1500 | *0.7961*±0.1737 | **0.7374**±0.8119 | **0.0833**±0.2764 |
> > > | Deformer + TRinS | **0.7626**±0.1519 | **0.7315**±0.1993 | **0.3984**±0.1474 | **0.7652**±0.2191 | **0.6683**±0.0840 | **0.5065**±0.0749 | 0.9047±0.9916 | *0.1111*±0.3143 |
> > > | CBraMod | 0.6730±0.1592 | 0.6214±0.2087 | 0.5329±0.1748 | 0.0609±0.1803 | 3.5570±1.2185 | 3.0957±1.2356 | **0.6158**±0.6967 | *0.1389*±0.3458 |
> > > | CBraMod + SW | 0.6795±0.1668 | 0.6232±0.2200 | 0.4948±0.1870 | 0.3837±0.3400 | 1.1396±0.3929 | 1.1396±0.3929 | 0.8182±0.8824 | 0.1667±0.3727 |
> > > | CBraMod + TRin | *0.7096*±0.1667 | *0.6602*±0.2148 | *0.4441*±0.1292 | *0.5256*±0.2589 | *1.2520*±0.3902 | *1.0022*±0.3408 | *0.7605*±0.7012 | **0.1111**±0.3143 |
> > > | CBraMod + TRinS | **0.7149**±0.1681 | **0.6646**±0.2239 | **0.4403**±0.1305 | **0.7098**±0.2152 | **0.9745**±0.9905 | **0.5317**±0.0900 | 0.9745±0.9905 | *0.1389*±0.3458 |
> > >
> > >
> > > ---
> > > **Q5: [95% overlap inflating temporal stability metrics and whether TRin's regularization merely encode highly correlated segments]** "The high overlap(95%) between input windows ... merely smooths highly correlated inputs." "Testing with lower overlaps or variable window lengths ... due to the loss function versus inherent redundancy." "Have experiments ... effects of input overlap from the temporal regularization itself?""
> > >
> > > **A5:** Thank you for your critical thoughts. The high overlap (95%) between consecutive windows is to ensure real-time neurofeedback feasibility by 200ms delay [Teo et al., 2021; Jochumsen et al., 2019]. It also raises concerns about whether fluctuation issues are amplified, and whether temporal stability improvements are merely encoding correlated inputs. To address this, we add a new experiment (Appendix J) to evaluate TRin's performance across varying overlaps (95%, 75%, 50%).
> > >
> > > ---
> > > ### References:
> > >
> > > - S.-H. J. Teo, X. W. W. Poh, T. S. Lee, C. Guan, Y. B. Cheung, D. S. S. Fung, H. H. Zhang, Z. Y. Chin, C. C. Wang, M. Sung, T. J. Goh, S. J. Weng, X. J. J. Tng, and C. G. Lim. Brain-computer interface based attention and social cognition training programme for children with ASD and co-occurring ADHD: A feasibility trial. *Research in Autism Spectrum Disorders*, 89:101882, 2021. doi: 10.1016/j.rasd.2021.101882.
> > >
> > > - M. Jochumsen, M. S. Navid, R. W. Nedergaard, N. Signal, U. Rashid, A. Hassan, H. Haavik, D. Taylor, and I. K. Niazi. Self-paced online vs. cue-based offline brain–computer interfaces for inducing neural plasticity. *Brain Sciences*, 9(6):127, 2019. doi: 10.3390/brainsci9060127.

---

> > > > ### Author Response · Authors · 2025-11-21
> > > >
> > > > (continued from the previous response **A5**)
> > > >
> > > >
> > > > The result shows that TRin maintains consistent performance gains across all overlap conditions (Table B), demonstrating that its regularization transcends input overlap. Improvements persist even when segment correlations are reduced (75%/50% overlap), confirming robustness.
> > > >
> > > > **Table B:** Performance comparison on Mental Workload dataset with different overlap rate. Results show mean ± standard deviation over cross-validation folds. The best performance is highlighted in **bold**.
> > > >
> > > > | Overlap | Model | ACC(↑) | F1-macro(↑) | RMSE(↓) | PCC(↑) | FSE(↑) | HFSE(↓) |
> > > > |---------|-------|--------|--------------|---------|--------|--------|---------|
> > > > | 95% | EEGNet | 0.7111±0.1612 | 0.6788±0.1984 | 0.4774±0.1679 | 0.4862±0.3270 | 0.3964±0.2278 | 1.0126±0.4664 |
> > > > | 95% | EEGNet + TRin | **0.7348**±0.1744 | **0.6871**±0.2336 | **0.4248**±0.1713 | **0.5865**±0.3715 | **0.6111**±0.1573 | **0.3536**±0.1377 |
> > > > | 95% | TSception | 0.6798±0.1640 | 0.6306±0.2102 | 0.5121±0.1762 | 0.4400±0.3270 | 0.6071±0.1804 | 0.3679±0.2098 |
> > > > | 95% | TSception + TRin | **0.7390**±0.1724 | **0.7106**±0.2097 | **0.4295**±0.1729 | **0.5598**±0.3725 | **0.6652**±0.1943 | **0.3532**±0.1484 |
> > > > | 95% | Deformer | 0.7247±0.1602 | 0.6923±0.1993 | 0.4680±0.1718 | 0.5101±0.3194 | 0.4301±0.2216 | 1.4305±0.5940 |
> > > > | 95% | Deformer + TRin | **0.7645**±0.1539 | **0.7326**±0.2031 | **0.3983**±0.1508 | **0.6438**±0.3090 | **0.6529**±0.1568 | **0.4064**±0.1415 |
> > > > | 75% | EEGNet | 0.7063±0.1630 | 0.6740±0.1995 | 0.4801±0.1668 | 0.4903±0.3221 | 0.5033±0.2301 | 0.9360±0.5120 |
> > > > | 75% | EEGNet + TRin | **0.7359**±0.1767 | **0.6885**±0.2353 | **0.4257**±0.1701 | **0.5922**±0.3672 | **0.6420**±0.1598 | **0.3476**±0.1557 |
> > > > | 75% | TSception | 0.6793±0.1620 | 0.6297±0.2092 | 0.5108±0.1774 | 0.4439±0.3235 | 0.5567±0.1823 | 0.5689±0.3380 |
> > > > | 75% | TSception + TRin | **0.7403**±0.1729 | **0.7117**±0.2110 | **0.4298**±0.1721 | **0.5587**±0.3749 | **0.6215**±0.1965 | **0.5046**±0.2363 |
> > > > | 75% | Deformer | 0.7165±0.1627 | 0.6829±0.2013 | 0.4725±0.1770 | 0.4959±0.3272 | 0.4918±0.2275 | 1.0457±0.4888 |
> > > > | 75% | Deformer + TRin | **0.7614**±0.1525 | **0.7301**±0.2000 | **0.4009**±0.1498 | **0.6425**±0.2992 | **0.6603**±0.1592 | **0.3853**±0.1617 |
> > > > | 50% | EEGNet | 0.7068±0.1674 | 0.6735±0.2036 | 0.4812±0.1654 | 0.4917±0.3267 | 0.5192±0.1962 | 0.8548±0.4273 |
> > > > | 50% | EEGNet + TRin | **0.7386**±0.1777 | **0.6924**±0.2356 | **0.4250**±0.1709 | **0.5903**±0.3749 | **0.6284**±0.1617 | **0.4008**±0.1940 |
> > > > | 50% | TSception | 0.6815±0.1667 | 0.6319±0.2140 | 0.5073±0.1849 | 0.4473±0.3339 | 0.5422±0.2056 | 0.6111±0.4207 |
> > > > | 50% | TSception + TRin | **0.7396**±0.1768 | **0.7114**±0.2142 | **0.4300**±0.1721 | **0.5522**±0.3810 | **0.6101**±0.1957 | **0.5697**±0.2789 |
> > > > | 50% | Deformer | 0.7183±0.1614 | 0.6841±0.1995 | 0.4673±0.1781 | 0.5074±0.3189 | 0.5167±0.2381 | 0.9761±0.5158 |
> > > > | 50% | Deformer + TRin | **0.7619**±0.1548 | **0.7293**±0.2043 | **0.3996**±0.1508 | **0.6402**±0.3149 | **0.6527**±0.1626 | **0.4455**±0.2102 |
> > > >
> > > > Importantly, both TRin and baseline models operate independently per segment during inference, so reducing overlap is functionally equivalent to downsampling predictions, which increases latency. Belinskaia et al. (2020) suggest that delayed neurofeedback impedes training. Thus, any observed performance improvement at lower overlap rates should not be interpreted as enhanced real-time effectiveness but as a sanity check to verify our TRin framework consistency.
> > > >
> > > > ---
> > > > **Q6: [Comparison with post-hoc]** "What are the trade-offs between temporal regularisation during training and simpler post-hoc label smoothing in terms of accuracy, stability, latency, and computational complexity? Benchmarking would help understand and quantify the contributions."
> > > >
> > > > **A6:** Thank you for your constructive question. As detailed in A4, we evaluated TRin against post-hoc label smoothing (Table A or Appendix I), demonstrating its superiority in accuracy, stability, and latency. We apologize for any prior ambiguity and emphasize: TRin introduces no additional computational overhead or temporal dependencies during real-time inference. Unlike moving-average methods or recurrent frameworks, TRin operates identically to conventional models at inference time. It processes each input segment independently without reliance on historical predictions or prior signal segments. Also, TRin introduces no extra runtime cost compared to baseline architectures while still achieving regularization benefits. This key advantage of TRin has been explained in a new section 2.3 for clarity. Thank you.
> > > >
> > > > ### References:
> > > >
> > > > - A. Belinskaia, N. Smetanin, M. Lebedev, and A. Ossadtchi. Short-delay neurofeedback facilitates training of the parietal alpha rhythm. *Journal of Neural Engineering*, 17(6):066012, 2020. doi: 10.1088/1741-2552/abc8d7.

---

### Official Review · Reviewer_FH2K · 2025-11-02

**Soundness:** 3
**Presentation:** 3
**Contribution:** 2
**Rating:** 2
**Confidence:** 3

**Summary:**

The paper aims to address the gap in the BCI field, which is that all metrics focus on classification outcomes but not on the stability of classifications temporally. Authors mention 3 contributions: 1) a systematic study, 2) two new stability metrics, and 3) the TRin framework. Experiments are reported on three public datasets. The paper is clear and readable with sufficient motivation for the gap in the BCI field.

**Strengths:**

The paper provides sufficient motivation and is well-explained. It is also well written (except for minor issues with the style of introducing acronyms and the clarity of Figure 2), with explicit definitions for the stability metrics. The experimental setup covers multiple datasets and model families and helps interpret how temporal regularization leads to improved model performances.

**Weaknesses:**

The core ideas of (a) penalizing drastic (unexpected) output probability changes, (b) curriculum over time sequence, and (c) stability metrics are reasonable but not clearly distinguished from known practices (e.g., temporal smoothness penalties on logits, clip training, frequency-domain regularization). The paper would benefit from a crisper novelty outline: what is new versus standard temporal smoothing, what prior “temporal-consistency” losses do not cover.
The main comparative analysis to TRin is MSE loss, even though the task is classification, and the paper’s own baseline uses cross-entropy. MSE on logits is an unstable objective. The more relevant baselines are (i) CE + output smoothing, (ii) CE + temporal smoothing penalty on logits, and (iii) label-smoothing under CE. In particular, CE + a simple moving average of posteriors is a strong, practical baseline for real-time systems; Fig. 4’s qualitative results suggest that such post-processing might close much of the perceived gap without changing training.
The paper’s own metrics (FDS/FSE) emphasize continuity. They will reward any approach that smooths predictions. That’s fine, but to demonstrate broader utility, I would also expect analyses that: (i) hold latency constant when comparing against moving-average post-processing, (ii) evaluate under abrupt ground-truth changes, and (iii) report false-transition lag. The current results lean toward “smoother is better” without stress-testing responsiveness.

**Questions:**

Please map TRin’s pieces (loss form, curriculum learning, metrics) to prior temporal-consistency/smoothing literature, and clarify what is new in formulation or guarantees beyond standard smoothness penalties.

Can you share why MSE is used as a comparison to TRin and not CE? Annecdotally, how would the results change, and how would you compare your framework's performance with other metrics beyond "smoother is better"?

---

> ### Author Response · Authors · 2025-11-21
>
> Thank you for your very insightful comments! We have uploaded a supplemented paper with changes marked in blue.
>
> ---
> **Q1: [Map TRin to prior work]** "The core ideas ... standard temporal smoothing, what prior "temporal-consistency" losses do not cover." "Please map TRin's pieces ... beyond standard smoothness penalties."
>
> **A1:** Thank you for the great suggestion! Before discussing prior works, we apologize for any potential confusion regarding the term "clip." Our clip-based training refers to using short, ordered EEG segments as the smallest shuffling unit during training, which is unrelated to contrastive learning (CLIP). Unlike common practices (e.g., Dileo et al., 2023), where entire batches remain sequential, our approach shuffles clips for better generalization using EEG. Please refer to A3.1 below or Appendix H.2 for experimental details.
>
> We supplemented Section 2.4 and Appendix A to discuss prior work and TRin's relationship to it. To our knowledge, TRin is the first application of temporal regularization in BCI models. Although Phan et al. (2023) have proposed a temporal regularization in sleep staging, they aim to force the consecutive epochs' prediction loss (cross entropy) to be as close as possible. Thus, their word 'temporal' denotes training progress and is totally different from what we are discussing in this paper. In this case, we summarize typical uses of temporal smoothness penalties in other fields below and why they are less suitable in BCI than TRin.
>
> 1. **Loss term:** While prior works define temporal losses per batch (e.g., [Dileo et al., 2023], [Xu et al., 2021], and [Varghese et al., 2021]), TRin's Temporal Regularization Loss operates per clip. Batch-level loss in TRin is the average of clip losses. Though clip loss is similar in form to existing losses, our approach ensures compatibility with EEG dynamics. For justification of its necessity, please refer to A3.1 or Appendix H.2.
>
> 2. **Curriculum Learning:** In general, curriculum learning is a multi-stage training approach, such as first train with cross-entropy loss before introducing temporal penalties. We adopt this idea to gradually increase α in our dynamic strategy. However, unlike prior work [Dileo et al., 2023] that increases temporal coverage (inapplicable to BCI trials with limited trial durations), we uniquely address over-smoothing by introducing adaptive clip sizes as an additional curriculum. This novel clip-based approach demonstrates effectiveness in our ablation studies (Appendix H.1).
>
> 3. **Metrics:** In other fields like computer vision, some metrics include flow-based or perceptual consistency to rate temporal coherence of video predictions [Zhang et al., 2022]. They are inspiring, but still not applicable in EEG-based tasks given their fundamental differences in ground-truth and output. EEG-based classification lacks standardized stability metrics. Although regression tasks in EEG adopt RMSE, PCC, and CCC [Ding et al., 2025], these measures fail when ground truth labels are constant (Appendix F.2). Consequently, our work introduces novel stability-specific metrics tailored for EEG classification tasks.
>
>
> ---
> ### References:
>
> - H. Phan, E. Heremans, O. Y. Chén, P. Koch, A. Mertins, and M. De Vos. Improving automatic sleep staging via temporal smoothness regularization. *2023 IEEE International Conference on Acoustics, Speech and Signal Processing (ICASSP)*, pp. 1–5. IEEE, 2023. doi: 10.1109/ICASSP49357.2023.10095805.
>
> - M. Dileo, P. Minervini, M. Zignani, and S. Gaito. Temporal smoothness regularisers for neural link predictors. In *Temporal Graph Learning Workshop at NeurIPS*, New Orleans, USA, 2023.
>
> - Y. Xu, S. Sun, H. Zhang, C. Yi, Y. Miao, D. Yang, X. Meng, Y. Hu, K. Wang, H. Min, H. Song, and C. Miao. Time-aware graph embedding: A temporal smoothness and task-oriented approach. *ACM Transactions on Knowledge Discovery from Data*, 16(3):56, 2021. doi: 10.1145/3480243.
>
> - S. Varghese, S. Gujamagadi, M. Klingner, N. Kapoor, A. Bar, J. D. Schneider, K. Maag, P. Schlicht, F. Huger, and T. Fingscheidt. An unsupervised temporal consistency (tc) loss to improve the performance of semantic segmentation networks. In *Proceedings of the IEEE/CVF Conference on Computer Vision and Pattern Recognition (CVPR) Workshops*, pp. 12–20, 2021.
>
> - Y. Zhang, S. Borse, H. Cai, Y. Wang, N. Bi, X. Jiang, and F. Porikli. Perceptual consistency in video segmentation. In *Proceedings of the IEEE/CVF Winter Conference on Applications of Computer Vision (WACV)*, pp. 2623–2632, 2022. doi: 10.1109/WACV51458.2022.00266.
>
> - Y. Ding, C. Tong, S. Zhang, M. Jiang, Y. Li, K. J. L. Lim, and C. Guan. Emt: A novel transformer for generalized cross-subject eeg emotion recognition. *IEEE Transactions on Neural Networks and Learning Systems*, 36(6):10381–10394, 2025. doi: 10.1109/TNNLS.2025.3552603.

---

> > ### Author Response · Authors · 2025-11-21
> >
> > ---
> > **Q2: [Why do baselines include MSE]** "The main comparative analysis ... MSE on logits is an unstable objective." "Can you share why MSE is used as a comparison to TRin and not CE?"
> >
> > **A2:** Thank you for your careful question. We include MSE as one of our baselines because **(1)** Theoretically, MSE loss's term naturally punishes the output variance and promotes smoothness (Equation 1 or Appendix D.2). **(2)** Our experiment on SEED dataset shows that MSE with TRin training strategy can benefit temporal stability and classification.
> >
> > MSE is indeed an unstable objective. Our test on new baseline foundational model CBraMod finds difficulty in convergence, probably because it's too large and deep. Nonetheless, for the other smaller models (EEGNet, TSception, and Deformer), MSE does not suffer stability issues and can achieve even better results than CE on SEED.
> >
> > The bias-variance decomposition of MSE loss is shown in Equation 1:
> >
> > $$
> > \mathcal{L} _ {MSE} = \mathbb{E}[(\hat{y} - y)^2] = \underbrace{(\mathbb{E}[\hat{y}] - y)^2} _ {\text{Bias}^2} + \underbrace{\mathbb{E}[(\hat{y} - \mathbb{E}[\hat{y}])^2]} _ {\text{Variance}} \tag{1}
> > $$
> >
> > ---
> > **Q3: [Include more relevant baselines]** "The more relevant baselines are (i) CE + output smoothing, (ii) CE + temporal smoothing penalty on logits, and (iii) label-smoothing under CE ... lean toward “smoother is better” without stress-testing responsiveness." "Annecdotally, how would the results change ... other metrics beyond "smoother is better"?"
> >
> >
> > **A3:** Thank you for your constructive feedback! For each variant, we explain our supplemented experiments one by one:
> >
> > 1. **CE + temporal smoothing penalty:** We did a supplementary experiment using CE + temporal smoothing penalty and common practices where entire batches remain sequential [Dileo et al., 2023] without TRin's clip-wise shuffling strategy. For detailed results, please refer to Table A or Appendix H.2. The result shows that BL-TL (CE + temporal smoothing penalty only) achieves chance-level performance across all models. A possible reason is that EEG datasets typically have limited trial duration, often resulting in only one batch being sampled per trial. Without proper shuffling (TRin strategy), models tend to overfit the sequence of the data rather than learning generalizable patterns. This underscores the critical role of TRin's clip-wise shuffling, which ensures robust feature learning.
> >
> > **Table A:** CE + temporal smoothing penalty without clip-wise shuffling strategy on Mental Workload dataset across models. Results show mean ± standard deviation over cross-validation folds. The most optimal value is highlighted in **bold** and secondary optimal value is marked with *asterisk*. BL-TL represents CE + temporal smoothing penalty without TRin's clip-wise shuffling strategy.
> >
> > | Model | Config | ACC(↑) | F1-macro(↑) | RMSE(↓) | PCC(↑) | FSE(↑) | FDS_norm(↓) |
> > |-------|--------|--------|--------------|---------|--------|--------|-------------|
> > | EEGNet | BL-CE | *0.7111*±0.1612 | *0.6788*±0.1984 | *0.4774*±0.1679 | *0.4862*±0.3270 | *0.3964*±0.2278 | 1.0126±0.4664 |
> > | EEGNet | BL-TL | 0.4988±0.0278 | 0.4894±0.0324 | 0.5297±0.0415 | 0.0027±0.0800 | 0.1576±0.1044 | *0.8915*±0.6609 |
> > | EEGNet | TRin | **0.7348**±0.1744 | **0.6871**±0.2336 | **0.4248**±0.1713 | **0.5865**±0.3715 | **0.6111**±0.1573 | **0.3536**±0.1377 |
> > | TSception | BL-CE | *0.6798*±0.1640 | *0.6306*±0.2102 | *0.5121*±0.1762 | *0.4400*±0.3270 | *0.6071*±0.1804 | 0.3311±0.1708 |
> > | TSception | BL-TL | 0.5356±0.1133 | 0.4897±0.1327 | 0.5596±0.1131 | 0.1580±0.3268 | 0.4128±0.1621 | *0.3293*±0.2863 |
> > | TSception | TRin | **0.7390**±0.1724 | **0.7106**±0.2097 | **0.4295**±0.1729 | **0.5598**±0.3725 | **0.6652**±0.1943 | **0.2927**±0.1130 |
> > | Deformer | BL-CE | *0.7247*±0.1602 | *0.6923*±0.1993 | *0.4680*±0.1718 | *0.5101*±0.3194 | *0.4301*±0.2216 | 1.0448±0.4584 |
> > | Deformer | BL-TL | 0.5024±0.1472 | 0.4482±0.1577 | 0.5154±0.0496 | 0.0237±0.4094 | 0.3898±0.1066 | *0.3256*±0.1161 |
> > | Deformer | TRin | **0.7645**±0.1539 | **0.7326**±0.2031 | **0.3983**±0.1508 | **0.6438**±0.3090 | **0.6529**±0.1568 | **0.3075**±0.1065 |

---

> > > ### Author Response · Authors · 2025-11-21
> > >
> > > (continued from the previous response **A3**)
> > >
> > > 2. **CE + label smoothing:** Thank you for this insightful suggestion! Our supplementary experiments show label smoothing is simple and effective. We have incorporated label smoothing results in quantitative analyses across datasets. Full results are not listed here due to space limitation, please kindly refer to Tables 1/2/4 in the supplemented paper. Our experiments reveal that while label smoothing serves as a simple yet effective temporal regularizer, it demonstrates less consistent performance than TRin across architectures, with TSception being the only exception.
> > >
> > >     Notably, label smoothing variants show only marginal classification accuracy improvements, even less than MSE loss variants on SEED. This suggests that although effective for temporal regularization, label smoothing may not sufficiently guide models to learn meaningful or generalizable feature representations to benefit classification performance. At the same time, Visualization in Figure 4 and Appendix N shows that label smoothing variant is an effective temporal regularization, but still less capable of suppressing high frequency fluctuations than TRin.
> > >
> > > 3. **Post Moving Average and Latency:** Sorry for any ambiguity previously, we first wish to clarify that TRin does not introduce any additional computational overhead or temporal dependencies during real-time inference. The framework operates identically to traditional approaches at inference time, requiring no previous signal segments or historical predictions (unlike moving average methods and recurrent architecture). This key advantage of TRin has been explained in a new section 2.3 for clarity.
> > >
> > >     Due to the absence of fine-grained ground truth labels and abrupt mental state transitions in current BCI datasets, we simulate label shifts by concatenating trials from the same subject with differing labels. Our latency analysis (Table B in the following response) demonstrates that TRin maintains inference speeds comparable to baseline models, consistent with our theoretical expectations. Notably, while TRin preserves real-time efficiency, conventional post hoc sliding window methods introduce additional latency without improving classification accuracy. This highlights TRin's practical advantages for deployment. For visualization or detailed latency definition, please refer to Appendix I in the supplemented paper.

---

> > > > ### Author Response · Authors · 2025-11-21
> > > >
> > > > (continued from the previous response **A3**)
> > > >
> > > > **Table B:** Latency analysis with change in ground truth on Mental Workload dataset. Results show mean ± standard deviation over cross-validation folds. The best performance is highlighted in **bold** while the second best performance is highlighted in *italics*. SW represents sliding window post-processing, TRin represents our proposed framework, and TRinS represents TRin with post-processing smoothing. Lt refers to latency and UT refers to the number of unsuccessful transitions.
> > > >
> > > > | Model | ACC(↑) | F1-macro(↑) | RMSE(↓) | FSE(↑) | HFSE(↓) | FDS_norm(↓) | Lt(↓) | UT(↓) |
> > > > |-------|--------|--------------|---------|--------|---------|-------------|-------|-------|
> > > > | EEGNet | 0.7091±0.1584 | 0.6773±0.1951 | 0.4811±0.1625 | 0.1837±0.2179 | 2.4239±0.8844 | 2.3286±0.8305 | *0.8457*±0.6848 | **0.1111**±0.3143 |
> > > > | EEGNet + SW | 0.7167±0.1638 | 0.6834±0.2026 | 0.4555±0.1685 | 0.6147±0.2553 | 1.0076±0.2252 | 0.9627±0.2695 | 1.0055±0.8142 | 0.1944±0.3958 |
> > > > | EEGNet + TRin | *0.7327*±0.1701 | *0.6866*±0.2270 | *0.4270*±0.1663 | *0.6257*±0.2596 | *0.9739*±0.1439 | *0.8757*±0.2165 | **0.7736**±0.6787 | **0.1111**±0.3143 |
> > > > | EEGNet + TRinS | **0.7369**±0.1747 | **0.6897**±0.2329 | **0.4232**±0.1683 | **0.7158**±0.2482 | **0.6831**±0.0617 | **0.5157**±0.0775 | 0.9204±0.8783 | *0.1389*±0.3458 |
> > > > | TSception | 0.6771±0.1607 | 0.6288±0.2058 | 0.5166±0.1680 | 0.5545±0.3682 | 0.9376±0.1940 | 0.8842±0.2638 | **0.6455**±0.8027 | *0.1667*±0.3727 |
> > > > | TSception + SW | 0.6777±0.1612 | 0.6293±0.2062 | 0.5117±0.1708 | 0.6116±0.2623 | *0.7391*±0.0796 | *0.6538*±0.1478 | 0.8084±0.8415 | 0.1667±0.3727 |
> > > > | TSception + TRin | **0.7367**±0.1698 | **0.7084**±0.2071 | *0.4327*±0.1700 | *0.6897*±0.2177 | 0.8531±0.1125 | 0.7815±0.1643 | *0.7596*±0.7819 | **0.1389**±0.3458 |
> > > > | TSception + TRinS | *0.7369*±0.1747 | *0.6897*±0.2329 | **0.4232**±0.1683 | **0.7158**±0.2482 | **0.6831**±0.0617 | **0.5157**±0.0775 | 0.9187±0.9565 | **0.1389**±0.3458 |
> > > > | Deformer | 0.7228±0.1572 | 0.6910±0.1956 | 0.4721±0.1663 | 0.1994±0.2123 | 2.4212±0.8142 | 2.3758±0.8505 | *0.7736*±0.6193 | *0.1111*±0.3143 |
> > > > | Deformer + SW | 0.7299±0.1633 | 0.6967±0.2037 | 0.4438±0.1728 | 0.6263±0.2682 | 0.9803±0.2115 | 0.9633±0.2702 | 0.9441±0.7902 | *0.1111*±0.3143 |
> > > > | Deformer + TRin | *0.7620*±0.1504 | *0.7311*±0.1978 | *0.4018*±0.1460 | *0.6948*±0.2231 | *0.9164*±0.1500 | *0.7961*±0.1737 | **0.7374**±0.8119 | **0.0833**±0.2764 |
> > > > | Deformer + TRinS | **0.7626**±0.1519 | **0.7315**±0.1993 | **0.3984**±0.1474 | **0.7652**±0.2191 | **0.6683**±0.0840 | **0.5065**±0.0749 | 0.9047±0.9916 | *0.1111*±0.3143 |
> > > > | CBraMod | 0.6730±0.1592 | 0.6214±0.2087 | 0.5329±0.1748 | 0.0609±0.1803 | 3.5570±1.2185 | 3.0957±1.2356 | **0.6158**±0.6967 | *0.1389*±0.3458 |
> > > > | CBraMod + SW | 0.6795±0.1668 | 0.6232±0.2200 | 0.4948±0.1870 | 0.3837±0.3400 | 1.1396±0.3929 | 1.1396±0.3929 | 0.8182±0.8824 | 0.1667±0.3727 |
> > > > | CBraMod + TRin | *0.7096*±0.1667 | *0.6602*±0.2148 | *0.4441*±0.1292 | *0.5256*±0.2589 | *1.2520*±0.3902 | *1.0022*±0.3408 | *0.7605*±0.7012 | **0.1111**±0.3143 |
> > > > | CBraMod + TRinS | **0.7149**±0.1681 | **0.6646**±0.2239 | **0.4403**±0.1305 | **0.7098**±0.2152 | **0.9745**±0.9905 | **0.5317**±0.0900 | 0.9745±0.9905 | *0.1389*±0.3458 |
> > > >
> > > >
> > > > Lastly, we wish to clarify that our objective is not to pursue "smoother is better" as an absolute criterion. In fact, the TRin framework can achieve arbitrary low smoothness levels by adjusting hyperparameter α (Appendix G.2, Table 8). Instead, we tune our hyperparameter by FSE on validation set to balance classification accuracy and temporal stability. Critically, inference tests demonstrate that TRin's optimal configuration improves both classification performance and temporal stability. This contrasts with methods like label smoothing (Tables 1/2/4) or post-processing, which enhance stability but yield negligible accuracy gains. This hints TRin's regularization not only smoothens predictions but also guide models toward neurophysiologically generalizable features that inherently benefit classification.

---

### Author Response · Authors · 2025-11-27
**Summary and Further Discussion**

We thank all reviewers for their constructive feedback! Below, we summarize our main contributions and common concerns in the reviews.

---
### Key Contributions

**Problem Formalization:** First systematic study of prediction fluctuations in EEG-based BCIs, highlighting slow-evolving nature.

**Stability Metrics:** Propose FSE and FDS to quantify temporal instability beyond traditional accuracy metrics.

**Framework (TRin):** A stability-aware training framework that jointly improves classification accuracy and temporal stability.

**Real-Time Viability:** Readily applicable to real-time BCIs, with no additional inference latency or computational overhead.

---
### Common Concerns

**Baselines:** Extended experiments show TRin consistently outperforms new baselines (e.g., label smoothing, post-hoc sliding window). Results on new foundational model architecture confirm its robustness.

**Neurophysiological Alignment:** Stability metrics and loss functions target unrealistic high-frequency fluctuations, preserving biologically plausible changes.

**Overlap Sensitivity:** Experiments with 50–95% window overlaps verify that TRin's improvements are not due to input redundancy.

**Responsiveness:** Under abrupt transitions, TRin maintains superior real-time responsiveness compared with sliding-window baselines.

---
We thank the reviewers again for their insightful feedback and welcome any further discussion!

---

### Meta-Review · Area_Chair_mKKV · 2025-12-27

**Summary:**

This paper targets improving issues with temporal stability in sequence models conditional on EEG data. Authors claim that even state-of-the-art models in terms of accuracy behave sharply, which may hinder downstream application. To counter that, authors propose both a regularisation strategy that biases learning algorithms towards smoother candidate models, and further proposed modifications in training data pre-processing so as to better preserve temporal correlation between chunks of data presented to models. Evaluation metrics are also introduced to enable comparing models in terms of how smooth they are along the time axis. Results do show improvements in terms of average performance, both along the propose metrics and accuracy.

**Reviewer Concerns:**

Main concerns brought up by reviewers revolve around the novelty of the proposal, as temporal smoothing approaches for sequence models do exist in the literature, and the paper doesn't fully motivate the proposal of the specific approach as an alternative to existing smoothing methods. There were also comments around the evaluation and the choice of baselines. I would also add concerns related to the motivation of the paper itself. The paper makes strong claims involving issues with existing state-of-the-art model, and so I would expect empirical analysis to demonstrate the issue, and more importantly, characterize it. Why is it that existing models exhibit this behavior? Is it a matter of approximation error? Sequence models do match data temporally in other domains, and the underlying reasons for the behaviour in this specific type of data are not made clear in the paper. Regarding the empirical evaluation, I would also mention that significancy of differences in performance is not clear, although average performance shifts were observed, and it's unclear to what extent variations in results are due to the data pre-processing changes or the proposed regularization approach. It's also not very clear why improvements in accuracy were observed, as the proposed regularization is orthogonal to the training maximum likelihood objective. Reviewers also mentioned other concerns such as the smoothing induced by the regularization possibly affecting the accuracy of models under natural abrupt changes. There were also notes regrading the effect of the overlapping windows confounding the results, and authors did present extra experiments with varying overlaps to show that not to be an issue.

**Reviewer Scores:**

3/4 reviewers leaned negatively. Authors did respond to concerns meaningfully, though I would say some of the raised concerns require major updates in the manuscript. More importantly in my opinion is fully understanding the reason why existing models behave so sharply, as additional regularization is but one candidate solution to that issue. The evaluation also requires expansion. All in all, I would expect only minor changes in scores, and while the proposal is relevant and results point in the right direction, the paper requires improvements and clarifications prior to publication.

---

### Decision · Program_Chairs · 2026-01-26

Reject